# A Reassessment of Sarcopenia from a Redox Perspective as a Basis for Preventive and Therapeutic Interventions

**DOI:** 10.3390/ijms26167787

**Published:** 2025-08-12

**Authors:** Alessia Arcaro, Alessio Lepore, Giovanni Paolo Cetrangolo, Gianluca Paventi, Paul Richard Julian Ames, Fabrizio Gentile

**Affiliations:** 1Department of Medicine and Health Sciences ‘V. Tiberio’, University of Molise, 86100 Campobasso, Italy; alessia.arcaro@unimol.it (A.A.); alessio.lepore@unimol.it (A.L.); gianpaolo.cet@tiscali.it (G.P.C.); 2Department of Agricultural, Environmental and Food Sciences, University of Molise, 86100 Campobasso, Italy; paventi@unimol.it; 3Immune Response & Vascular Disease Unit, CEDOC, Nova University Lisbon, 1169-056 Lisbon, Portugal; paxmes@aol.com; 4Department of Haematology, Dumfries Royal Infirmary, Cargenbridge, Dumfries DG2 8RX, UK

**Keywords:** sarcopenia, oxidative stress, reactive oxygen species (ROS), reactive nitrogen species (RNS), advanced glycation/glycoxidation end products (AGEs), advanced lipid peroxidation end products (ALEs), antioxidants

## Abstract

The use of a wide variety of antioxidants has been advocated as a means to prevent, delay the progression of, or counteract the adverse consequences of sarcopenia, such as loss of muscle strength, muscle quantity/quality, and physical performance. However, these proposals do not always appear to be supported in the literature by a thorough understanding of the contribution of redox perturbations to the pathogenesis of sarcopenia, nor of the biochemical properties, mechanism of action, pharmacokinetics, and pharmacodynamics of different antioxidants. This review discusses these aspects, aiming to provide a rationale for the selection and use of antioxidants in sarcopenia. After providing a definition of sarcopenia in the context of frailty, we distinguish between oxidative eustress as a physiological response of muscle cells to mild stimulation, such as moderate exercise, mediating their capacity for adaptation and regeneration, and oxidative distress as a pathophysiological response to muscle cell damage and death. The role of oxidative damage to biological macromolecules, both direct and mediated by advanced lipid peroxidation end products and advanced glycation/glycoxidation end products, is examined in detail. Next, we discuss antioxidant defense mechanisms, both enzymatic and non-enzymatic, including redox-sensitive gene regulatory events presided over by nuclear factor erythroid 2-related factor 2, the master regulator of enzymatic antioxidants. The review then discusses criteria for a rational classification of non-enzymatic antioxidants. This is followed by a review of some of the main radical-trapping antioxidants, both phenolic and non-phenolic, whose characteristics are compared.

## 1. Introduction

### 1.1. The Evolving Clinical Definition of Sarcopenia

Sarcopenia was originally defined as “a syndrome characterized by progressive and generalized loss of skeletal muscle mass and strength with a risk of adverse outcomes such as physical disability, poor quality of life and death”, occurring mostly in the elderly population [1]. Its prevalence varies from 10 to 20% in people over 65 years of age and reaches 50% in people over 80 years of age [2]. In 2008 the European Working Group on Sarcopenia in Older People (EWGSOP) defined sarcopenia as low muscle mass plus low grip strength or low gait speed [3]. The EWGSOP2 updated their definition of sarcopenia in 2018, stating that it can no longer be considered a condition mainly associated with ageing, rather should be seen as a pathological condition beginning earlier in life due to multiple causes in addition to ageing [4]. In fact, it is now regarded as a muscle disease (muscle failure) in which low muscle strength overtakes the role of low muscle mass and favors the development of adverse outcomes, such as falls, fractures, physical disability, and mortality [5]. Indeed, low muscle strength, low muscle quantity/quality, and low physical performance are the main diagnostic parameters of sarcopenia. Low muscle strength alone is sufficient to suspect sarcopenia, but the presence of both low muscle strength and low muscle quantity or quality confirms the diagnosis. The simultaneous presence of all three parameters defines a condition of severe sarcopenia. To diagnose sarcopenia, the EWGSOP2 recommends the Find–Assess–Confirm–Severity (F-A-C-S) pathway:

*Find cases*: The SARC-F questionnaire must be used to detect sarcopenia-associated symptoms in individuals at risk.

*Assess*: A grip strength test or a chair stand measure, with specific cut-off points for each test, are required to assess sarcopenia (see Table 1).

*Confirm*: A confirmatory diagnosis of sarcopenia requires the determination of low muscle quantity and quality by Dual-energy X-ray Absorptiometry (DXA).

*Severity*: The severity of sarcopenia can be evaluated by performing such tests as (1) gait speed measurement; (2) the Short Physical Performance Battery (SPPB); (3) the Timed Up and Go test (TUG); (4) a 400 m walk test (see Table 1) [5]. The International Working Group on Sarcopenia (IWGS), the Asian Working Group on Sarcopenia (AWGS), the American Foundation for the National Institutes of Health (FNIH), and the Sarcopenia Definition and Outcomes Consortium (SDOC) have their own definitions of sarcopenia and recommendations for diagnostic procedures, assays, and cut-off points [6].

### 1.2. Sarcopenia in the Context of Ageing and Frailty

The physical frailty (PF) phenotype proposed by Fried et al. is defined as a complex syndrome characterized by decreased physiological reserves and resistance to acute stress, resulting from cumulative declines across multiple physiological systems and causing vulnerability to adverse outcomes [1]. Sarcopenia and frailty are associated with ageing and with each other. Joint associations with frailty and ageing extend beyond sarcopenia to include other pathological conditions, with which sarcopenia itself may be associated, participating not only in clinical manifestations, but also in pathogenesis, with a role of cause or effect. For example, frailty index and biological age have been jointly associated with chronic obstructive pulmonary disease (COPD), of which respiratory muscle fatigue and sarcopenia may be not only an associated clinical manifestation, but also an important contributing factor [7]. In patients undergoing dialysis for chronic kidney disease (CKD), serum levels of uremic toxins, such as advanced glycation/glycoxidation end products (AGEs, see Section 3.5 below), increased with frailty status and were inversely associated with physical performance and activity. Furthermore, a DNA aptamer produced against AGEs was able to inhibit nephrectomy-induced gastrocnemius muscle atrophy in mice [8]. Sarcopenia also increases the clinical burden in elderly patients with mineral and bone disorders associated with CKD and frailty [9]. However, whether frailty is an outcome of sarcopenia or sarcopenia is a clinical manifestation of frailty is a matter of debate resembling the “egg and chicken” dilemma [10].

Sarcopenia may be viewed both as a unified manifestation of multiple etiological factors, constituting the biological substrate of frailty, and as the pathway mediating the negative health outcomes of frailty [11]. Frailty has multiple causes and contributing factors, which can be physical, psychological, social, or a combination of these. It is characterized by a spectrum of functional impairments and vulnerabilities that can include loss of muscle mass and strength, reduced energy and exercise tolerance, cognitive impairment, and limitations in mobility and the ability to care for oneself and perform basic and instrumental activities of daily living. Risk factors include advanced age, poverty, isolation, poor nutrition and weight loss, and medical and psychiatric comorbidities, including diabetes, dyslipidemia, hypertension, chronic obstructive pulmonary and kidney disease, cardiovascular and neurological diseases, osteoporosis, musculoskeletal disorders, anxiety, depression, and polypharmacy [12].

Although PF encompasses only part of the frailty spectrum, the five-item instrument proposed for its assessment (unintentional weight loss, self-reported exhaustion, weakened grip strength, slow walking speed, low physical activity) is useful to screen for the functional deficits associated with PF by the Short Physical Performance Battery (SPPB) and for the design of preventive interventions.

PF predisposes to and is predictive of major adverse health outcomes, such as motor and daily living disability, falls, independence loss, institutionalization, and death. This has been the subject of numerous reports. For example, significant associations have been demonstrated between PF and the risk of suicide attempts in a cohort of US veterans aged 65 and older [13]. Furthermore, PF has proven to be a valuable tool useful for preoperative risk stratification and targeted pre-habilitation to surgery in the elderly [14]; predicting the risk of adverse postoperative outcomes in patients undergoing oncology surgery [15]; and identifying cohorts on the Alzheimer’s disease spectrum with a lower quality of life [16]. Furthermore, PF has been associated with a nearly five-fold increase in the odds ratio of patient discharge from the trauma department for further inpatient care [17] and has proven to be the most consistent and significant predictor of hospital readmission or death in patients with heart failure [18].

The recognition of sarcopenia as a major component of PF implies that interventions targeting skeletal muscle decay may provide useful preventive and therapeutic advantages against frailty and its clinical correlates [6]. Indeed, the China Health and Retirement Longitudinal Study (CHARLS) highlights the independent and significant association of sarcopenia with frailty and pre-frailty among older adults in China. In fact, early detection and targeted interventions in sarcopenia are crucial to mitigating frailty and its adverse health outcomes in ageing populations, emphasizing the need for tailored healthcare strategies to promote healthy ageing [19]. From the above, an integrated and multivariate intervention for the early assessment of sarcopenia should represent the primary task of any preventive approach for healthy ageing.

## 2. Sarcopenia: A Redox Mechanistic Perspective

### 2.1. The Continuum of Reactive Oxygen/Nitrogen Species (ROS/RNS)-Incited Responses in Skeletal Muscle

Muscle contraction stimulates the production of superoxide radical anion O_2_•^−^ and H_2_O_2_ through the one-electron or two-electron reduction of O_2_ mediated by NADPH oxidases (NOXs) located in the sarcolemma, sarcoplasmic reticulum, and inner mitochondrial membrane [20,21], endothelial xanthine oxidase (XO) [22,23], respiratory complexes of the mitochondrial electron transport chain (ETC), and several mitochondrial enzymes [24,25]. The fraction of O_2_ from which O_2_•^−^ is generated by one-electron reduction varies from 0.1% to 2%, depending on the mitochondrial stage of respiration. O_2_•^−^ generation is higher in resting muscles compared to muscles undergoing mild and intense aerobic exercise. Electron leakage from complexes I and III of the ETC plays a special role in the generation of O_2_•^−^ and other ROS/RNS [24,25,26]. Mitochondrial H_2_O_2_ production is dramatically decreased with acute and chronic eccentric exercise in rat skeletal muscle [27]. Hence, mitochondria can produce ROS/RNS at multiple locations, but there is increasing agreement that the increases in ROS/RNS production brought on by muscle contraction do not originate in the mitochondria and that NOXs are major sources of ROS/RNS production in contracting muscles [28]. Enzymes of the NOX family have been shown to play important physiological functions in skeletal muscle, being crucial in the excitation–contraction coupling and the regulation of signaling pathways involved in the adaptation to exercise training [29]. NOX2 and 4, which in transverse striatum muscle cells are found in the subsarcolemmal region and the sarcoplasmic reticulum, respectively, are the main sources of ROS/RNS, controlling Ca^2+^ release from the sarcoplasmic reticulum during muscle contraction via oxidative modifications of the ryanodine receptor 1 (Ryr1) [30] (see Section 2.2 below). NOX4 provides the synthesis of H_2_O_2_, whereas NOX1 and 2 produce O_2_•^−^. Both syntheses are enhanced by high-intensity [31] and moderate-intensity exercise [28] (see below for definitions of physical activity regimens). XO is known to contribute to O_2_•^−^ generation in the extracellular space following muscle contraction [32]. The deamination of AMP to inosine monophosphate (IMP) during muscle contraction is linked to a faster conversion of ATP into ADP. O_2_•^−^ is produced when XO converts the hypoxanthine produced from IMP into xanthine [33]. O_2_•^−^ released into the extracellular space by the action of sarcolemmal NOX2 and endothelial XO is converted to H_2_O_2_ by superoxide dismutase 3 (SOD3) [34] and enters muscle cells both by passive diffusion across the lipid bilayer and by transport facilitated by aquaporin 4 (AQP4) (Figure 1) [35]. Instead, H_2_O_2_ released by NOX4 into the mitochondrial matrix (mt-matrix) is transported across the inner and outer mitochondrial membranes by AQP8 [36] and the voltage-dependent anion channel (VDAC), respectively [37]. The H_2_O_2_ pool in the mt-matrix is also fueled by the dismutation of ETC-derived O_2_•^−^ operated by SOD2. H_2_O_2_ produced in the sarcoplasmic reticulum by NOX4 exits into the cytosol via AQP11 [38].

Skeletal muscles constitutively express neuronal and endothelial nitric oxide synthase (nNOS and eNOS). •NO release increases eight-fold during active contraction in mouse diaphragmatic muscle [39]. Neuronal NOS (nNOS, NOS I) is expressed in type II fast-twitch muscle fibers, mostly near the sarcolemma [40], while inducible NOS (iNOS, NOS II) is a source of •NO in inflammatory and immune responses [41]. Instead, the physiological role of endothelial NOS (eNOS, NOS III) in skeletal muscle needs clarification, since •NO release rate and muscle contraction are not affected in NOS III-deficient mice [39]. Reaction of •NO with O_2_•^−^ produces the peroxynitrite anion (ONOO^−^), a potent oxidizing and nitrating agent.

Electron leakage from the mitochondrial ETC is maintained within physiological limits when cellular antioxidant defenses are fully operational, under which conditions H_2_O_2_ and O_2_•^−^ act as second messengers that influence the expression of numerous genes related to antioxidant defense, metabolic control, and stress tolerance (Figure 1). In muscle cells, skeletal muscle satellite cells (SMSCs) and peripheral nervous cells innervating skeletal muscles, ROS/RNS activate redox-sensitive signaling pathways that regulate phasic contraction for movement and respiration, tonic postural contraction, and thermogenesis [42]. Furthermore, ROS/RNS control intracellular signal transduction involved in the metabolic adaptation of muscle cells to increased intensity and duration of workloads [33,43,44,45]. Low levels of oxidant species produced in resting muscle cells ensure the maintenance of proper muscle tone, while increased ROS/RNS production during aerobic exercise helps muscles withstand stress without suffering damage [46].

Below are some definitions of the most common exercise regimens.

*High-intensity exercise (or vigorous, strenuous exercise)* is physical activity that increases heart rate and breathing to high levels and requires high levels of effort and energy expenditure. *Exhaustive exercise* is an extreme form of high-intensity exercise that involves maximal effort, pushing the body to its limits, and resulting in extreme fatigue and potentially exhaustion. *High-intensity interval training (HIIT)* consists of short bouts of vigorous exercise interspersed with short recovery periods. The former can last from 10 s to a few minutes, with exercise intensities ranging from 80 to 150% of maximum oxygen consumption (VO_2_ max). VO_2_ max is an indicator of aerobic and cardiovascular fitness, referring to the maximum amount of O_2_ an individual can utilize during exercise [28]. Adaptation to HIIT involves replenishing glycogen stores in skeletal muscle and increasing the number of cellular mitochondria and the effectiveness of oxidative damage repair systems. Trained individuals produce lower levels of ROS at a given exercise intensity than untrained individuals [47].

*Endurance training (also named aerobic or cardio training)* is continuous exercise performed for extended periods of time at low to moderate intensity, ranging from 50 to 75% of VO_2_ max. Adaptation to endurance training involves increased mitochondrial respiratory capacity and cardiovascular efficiency [28].

*Resistance training* involves performing muscular work against a resisting force, such as lifting weights. It improves muscle strength through hypertrophy. Athletes who practice it have muscle fibers characterized by high oxidative metabolism and protein synthesis capacity and a higher content of myosin heavy chain type I (MyHC I) [48].

*Balance training* involves physical exercises aimed at improving the ability to maintain one’s center of gravity and control posture, both while standing still and in motion. It is particularly useful for preventing falls in the elderly [49].

The process by which cells subjected to a mild stress become resistant to a more severe stress is called hormesis or mitohormesis, because mitochondrial metabolism and dynamics are central in the maintenance of redox homeostasis, energy balance, and the equilibrium between the production and degradation of contractile proteins and between apoptosis and regeneration in muscle cells [44,45,50,51,52,53,54]. Hormesis may underlie the health-improving effects of genes encoding proteins in the insulin signaling pathway, the histone deacetylases called sirtuins, and certain environmental factors, such as dietary energy restriction (controlled caloric restriction and intermittent fasting) and physical exercise [53,55]. However, oxidative stress is a “double-edged sword”, as ROS/RNS such as •NO, O_2_•^−^, and H_2_O_2_ can affect muscle cell viability and performance in a dual fashion, described by a bell-shaped dose–response curve, where ROS/RNS levels typical of low-level activity facilitate adaptation and those accompanying excessive exercise can be detrimental. Indeed, moderate exercise appears to increase tolerable ROS/RNS levels through increased activity of antioxidant enzymes, thus improving physiological functioning [53]. Unlike regular intermittent exercise, where exercise-induced ROS/RNS production is overcompensated by an upregulated antioxidant defense, excessively prolonged or intense exercise is associated with ROS/RNS production that exceeds limits, where antioxidant defenses can be overcome and mitochondrial dysfunction and muscle fiber damage can occur, triggering inflammatory responses that fuel a vicious cycle of oxidative damage [47,56,57,58].

Some authors have attempted to provide quantitative estimates of H_2_O_2_ concentrations that can lead to different cellular outcomes and therefore be associated with different qualities of oxidative stress. In healthy cells, physiological steady-state H_2_O_2_ fluxes cause reversible oxidative modifications of target proteins to levels that allow the regulation of various cellular processes through physiological levels of redox signaling, a condition termed “oxidative eustress”. Conversely, supraphysiological levels of ROS/RNS lead to widespread oxidative damage to proteins, lipids, and nucleic acids, followed by disruption of normal cellular functions, a condition that has instead been termed “oxidative distress” [43,59,60,61,62,63,64]. It has been reported that concentrations of H_2_O_2_ in the range from 0.01 to 0.1 mM are essential for maintaining physiological quiescent metabolism, typical of cell proliferation, differentiation, migration, and angiogenesis [63]. Oxidative eustress is associated with ROS levels that promote optimal adaptive responses to exercise, in which cellular stress sensors, such as AMP-activated protein kinase (AMPK), peroxisome proliferator-activated receptor-γ (PPAR-γ) coactivator 1-α (PGC-1α), and sirtuins (SIRTs), are activated to initiate transcriptional regulatory events (see Section 2.2 below) without causing cell damage [59,63,65]. During muscle contraction, intracellular H_2_O_2_ can reach concentrations of 0.1–0.2 mM [66]. The transition from adaptive eustress to distress can occur at H_2_O_2_ levels ranging from 0.1 to >1 mM, which is difficult to predict in any given situation. Higher levels of H_2_O_2_ generated during exercise can cause cellular damage, muscle injury, and inflammation [59,63]. Furthermore, there is evidence that neuronal growth is essential in muscle remodeling and depends on the production of H_2_O_2_ by NOXs. H_2_O_2_ levels between 1 nM and 10 nM may favor physiological neuronal growth. However, H_2_O_2_ levels above 10 nM cause oxidative damage in neurons, while any further increase above 100 nM leads to neuronal degeneration [63]. Although, to date, few studies have examined the contribution of ROS to muscle neuronal growth, the mechanism may be similar to that observed in muscle. However, the authors who provided the above data cautioned that these can only be considered rough estimates, as real-time measurements of H_2_O_2_ production stimulated by contraction in muscle cells are technically difficult, and cellular H_2_O_2_ levels can differ among cell types and subcellular locations [33,43,63]. Intracellular H_2_O_2_ in single isolated murine skeletal muscle fibers has been measured by real time fluorescence microscopy with 5- (and 6-) chloromethyl-2′,7′-dichlorodihydrofluorescein diacetate (CM-DCFH DA) [67]. Guidelines for measuring ROS and oxidative damage in cells and in vivo have been proposed [68].

Currently, however, there is no unambiguous protocol or marker capable of distinguishing moderate from excessive physical exercise and predicting its beneficial or harmful effects, i.e., improvement or reduction in muscle health and performance [43]. From a practical perspective, exercise programs shown to improve muscle mass, strength, and performance in sarcopenic older adults were multimodal or mixed. They included (a) resistance exercises performed 1 to 5 times per week, at an intensity of 20 to 80% of the one-repetition maximum (RM) or 6 to 14 points on the Ratings of Perceived Exertion (RPE), and lasting 20 to 75 min; (b) aerobic exercises performed 2 to 5 times per week, at an intensity of 50 to 70% of the maximum heart rate or 7 to 17 points on the RPE, and lasting 6 to 30 min; (c) balance exercises performed 2 to 3 times per week, at an effort level of 3 on a scale of 10, and lasting 5 to 30 min [69].

**Figure 1 ijms-26-07787-f001:**
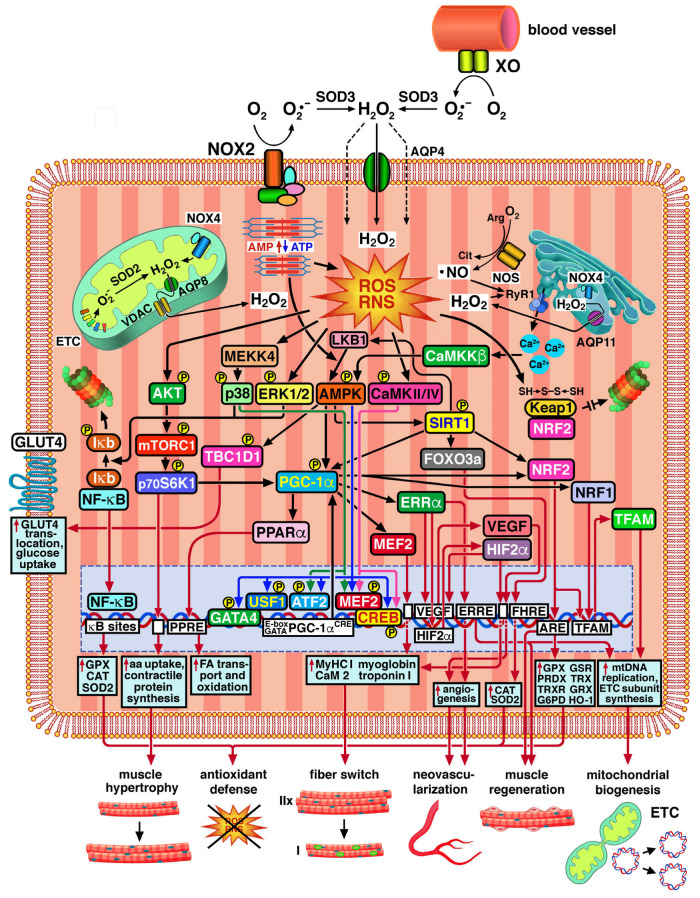
Mechanisms of ROS/RNS-mediated adaptation to exercise in skeletal muscle cells. At the top, the main sources of ROS/RNS in muscle are illustrated, including NOXs at various subcellular sites, mitochondrial ETC, endothelial XO, and NOS. Muscle contraction stimulates ROS/RNS production, resulting in phosphorylation of AKT, p38 MAPK, ERK1/2, AMPK, and CaMKII/IV. The mediating role of MEKK4, LKB1, and CaMKKβ, is illustrated, in conjunction with the contraction-induced changes in the AMP/ATP ratio, and the ROS/RNS-dependent opening of the RyR1 Ca^2+^ release channels. NF-kB pathway activation by p38 MAPK and ERK1/2 signaling and the subsequent upregulation of antioxidant responses are shown. Transcriptional upregulation of PGC-1α mediated by the transcription factors MEF2, CREB, ATF2, USF1, and GATA4 activated by p38 MAPK (green arrows), AMPK (blue arrows), and CaMKII/IV (purple arrows) is illustrated. The figure also shows synergistic interactions of PGC-1α with PPARα, ERRα, MEF2, NRF1, and NRF2, resulting in the upregulation of genes whose products positively influence the metabolic capacity and the tolerance to exercise of muscle cells. Other gene regulatory events are symbolized by red arrows pointing to gene promoters and DNA response sequences, such as κB sites, CREs, E-boxes, GATA sequences, AREs, PPREs, ERREs, and FHREs, which are symbolized by white square boxes. The activating effects of SIRT1 on LKB1, PGC-1α, FOXO3a, and NRF2 are also illustrated. Cellular responses, such as NF-kB-, FOXO3a-, and NRF2-dependent antioxidant responses; AKT/mTORC1/p70S6K1-dependent increases in amino acid uptake and contractile protein synthesis; PPARα-dependent increases in fatty acid transport and oxidation; ERRα- and VEGF-dependent angiogenesis; MEF2- and ERRα/HIF2α-dependent increases in MyHC I synthesis; ERRα- and NRF1/NRF2/TFAM-dependent mtDNA replication and ETC subunit synthesis; and TBC1D1-promoted translocation of GLUT4 to the sarcolemma, which increases glucose uptake, are highlighted by red arrows pointing to light blue square boxes. The resulting muscle adaptations, such as antioxidant defense, muscle hypertrophy, fiber type switching, neovascularization, SMSC regeneration, and mitochondrial biogenesis, are illustrated at the bottom. See the text for details. Legend: aa: amino acid; AMPK: AMP-activated protein kinase; AQP 4, 8, 11: aquaporin 4, 8, 11; ARE, ERRE, FHRE, and PPRE: antioxidant, ERR-α, FOXO3a, and PPAR-α response elements, respectively; ATF2: activating transcription factor 2; CaMKII/IV: Ca^2+^/calmodulin-dependent protein kinase II and IV; CaMKKβ: Ca^2+^/calmodulin-dependent protein kinase kinase β; CAT: catalase; CREB: cAMP-responsive element (CRE)-binding protein; ERR-α: estrogen-related receptor α; ETC: electron transport chain; FA: fatty acid; FOXO3a: activating forkhead box O3a; G6PD: glucose-6-phosphate dehydrogenase; GATA4: GATA binding protein 4; GLUT4: glucose transporter 4; GPX: glutathione peroxidase; GRX: glutaredoxin; GSR: glutathione reductase; HIF2α: hypoxia-induced factor 2α; **HO**-1: heme oxygenase-1; Keap1: Kelch-like erythroid-derived cap’n’collar homolog (ECH)-associated protein 1; LKB1: liver kinase B1; MEF2: myocyte enhancer factor 2; mTORC1: mammalian target of rapamycin complex 1; MyHC I: myosin heavy chain type I; NF-kB: nuclear factor κB; NRF1: nuclear respiratory factor 1; NRF2: nuclear factor erythroid 2-related factor 2; p70S6K1: p70 ribosomal protein subunit 6 kinase 1; PGC1-*α*: peroxisome proliferator-activated receptor-γ (PPAR-γ) coactivator 1-α; PRDX: peroxiredoxin; RyR1: ryanodine receptor 1; SIRT1: silent information regulator 1; SOD2 and 3: superoxide dismutase 2 and 3; TBC1D1: TBC1 domain family member 1; TFAM: transcription factor A, mitochondrial; TRX, thioredoxin; TRXR: thioredoxin reductase; USF1: upstream stimulator factor 1; VDAC: voltage-dependent anion channel; VEGF: vascular endothelial growth factor; XO: xanthine oxidase. Graphic elements for muscle fibers drawn by brgfx/Freepik (http://www.freepik.com).

### 2.2. ROS/RNS-Mediated Adaptation of Skeletal Muscle to Exercise

ROS/RNS generated during muscle contraction mediate the adaptation of skeletal muscle cells to exercise through the activation of various signaling pathways (Figure 1) [20,21,28,33,43]. As an instance, exercise stimulates phosphorylation of p38 and ERK1/2 mitogen-activated protein kinases (MAPKs), which activate the nuclear factor κB (NF-κB) pathway in rats and humans both during moderate exercise [70] and exhaustive exercise [71]. Various NF-κB heterodimers positively regulate several transcriptional targets, including enzymatic pro-oxidants and antioxidants. The latter include glutathione peroxidases (GPXs), catalase, and SOD2 [72], whose NF-κB-mediated expression has been observed also in endurance training [43]. A single bout of anaerobic exercise in rats also induces PGC-1α expression [73]. The expression of both p38 MAPK and PGC-1α was inhibited by allopurinol, which indicates the primary regulatory role of ROS produced by XO [70,73]. Activation of p38 signaling in human HEK 293T cells exposed to 100 μM H_2_O_2_ is mediated by MAPK kinase kinase MEKK4 through the transient formation of conjugates with peroxiredoxin-2 (PRX2) [74]. In skeletal muscle cells, the biogenesis of mitochondria is positively regulated in response to exercise through PGC-1α, improving exercise tolerance and oxidative capacity [75]. PGC-1α expression and activity are markedly enhanced in response to exercise through the coordinated activation of p38 MAPK, AMPK, and Ca^2+^/calmodulin-dependent protein kinases CaMKII and CaMKIV [23], which is dependent on the production of ROS/RNS stimulated by muscle contraction [76]. Treatment of C2C12 muscle cells with 300 μM exogenous H_2_O_2_ induced PGC-1α expression via AMPK activation, while N-acetylcysteine hindered it [75]. Stimulation of AMPK activity in HEK 293 cells exposed to 100–250 μM H_2_O_2_ is mediated by S-glutathionylation, even in the absence of an increased AMP/ATP ratio, which is also a trigger of AMPK activation [77]. Indeed, AMPK acts as a cellular energy sensor, whose conformational changes upon AMP binding allow phosphorylation/activation by liver kinase B1 (LKB1) [78].

AMPK-dependent expression of PGC1α is mediated by the phosphorylation and binding to the PGC1α promoter of several transcription factors. These include myocyte enhancer factor 2 (MEF2), as well as cAMP response element (CRE)-binding protein (CREB), activating transcription factor 2 (ATF2), upstream stimulator factor 1 (USF1), and GATA binding protein 4 (GATA4), which bind to cAMP response elements (CREs), E-box, and GATA sequences, respectively [75,79]. In addition to these indirect ways, AMPK also shows the ability to directly phosphorylate and activate PGC1α. Among the other kinases that participate in the activation of PGC1α expression, p38 MAPK also does so by phosphorylating and activating both MEF2, and ATF2, while CaMKs exert similar effects through the activation of MEF2 and CREB [79]. In turn, PGC-1α activates and/or synergistically interacts with several transcription factors that upregulate genes whose products positively affect the oxidative capacity and the exercise tolerance of skeletal muscle cells under endurance training [80]. Some of these belong to the nuclear receptor superfamily, including peroxisome proliferator-activated α (PPARα), which regulates fatty acid transport and oxidation, and estrogen-related receptor α (ERRα), which is on the pathway that leads to the activation of mitochondrial biogenesis [81], angiogenesis, and muscle regeneration [80]. Other transcription factors involved include myocyte enhancer factor 2 (MEF2), which is implicated in type I muscle fiber determination, as well as nuclear respiratory factor 1 (NRF1) and nuclear factor erythroid 2-related factor 2 (NRF2), which also activate mitochondrial biogenesis by inducing the expression of transcription factor A, mitochondrial (TFAM). The latter enhances mtDNA replication and transcription and the expression of mtDNA-encoded ETC subunits [82,83,84]. In resting muscle fibers, NRF2 is sequestered in the cytoplasm by the regulatory protein Kelch-like erythroid-derived cap’n’collar homolog (ECH)-associated protein 1 (Keap1), which is an oxidative stress sensor that functions as an adaptor for the cullin 3-based ubiquitin E3 ligase, which in turn marks NRF2 for proteasomal degradation. In response to the increase in ROS production induced by muscular contraction, a complex mechanism based upon redox-sensitive cysteines, such as Cys226, Cys613, Cys622, and Cys624, in Keap1 allows NRF2 to escape ubiquitination and proteasomal degradation in response to oxidizing agents. While this is known to result from a conformational change in the E3 ligase complex, it is not certain that it also involves the dissociation of NRF2 from Keap1. According to a model, “suicidal” occupation of NRF2-binding sites in oxidized Keap1 by NRF2 allows newly translated NRF2 to freely translocate to the nucleus [85]. In endurance exercise, contraction-induced increases in ROS production promote NRF2 translocation into the nucleus, where it binds to antioxidant response elements (AREs), thus inducing the expression of genes encoding peroxiredoxins, thioredoxins, glutaredoxins, and other enzymes responsible for GSH production and regeneration, ROS detoxification, and NADPH regeneration, which protect muscle cells from oxidative damage [86,87] (see also Section 4.3 below).

PGC-1α can also be activated through deacetylation by silent information regulator 1 (SIRT1), a NAD^+^-dependent deacetylase. SIRT1 is indirectly activated when AMPK is activated, either as a consequence of the increased cellular AMP/ATP ratio during muscle contraction or of the direct action of ROS, as already noted above. The resulting upregulation of nicotinamide phosphoribosyltransferase (NAMPT) determines increases in NAD^+^ availability, which in turn activate SIRT1. In turn, SIRT1 can contribute to AMPK activation by activating LKB1 through deacetylation [78]. During high-intensity exercise, AMPK phosphorylation and activation by Ca^2+^/calmodulin-dependent protein kinase kinase β (CaMKKβ) may also result from the increased cytosolic Ca^2+^ concentrations caused by oxidative modifications of ryanodine Ca^2+^ release channels in sarcoplasmic reticulum (see below) [88]. By activating PGC-1α, SIRT1 participates in mitochondrial biogenesis and muscle fiber remodeling toward the slow, oxidative type. In addition, SIRT1 alleviates oxidative stress in different ways: (1) by inhibiting NF-κB through deacetylation of the RelA/p65 subunit; (2) by activating forkhead box O3a (FOXO3a), which induces the transcription of several nuclear and mitochondrial genes encoding enzymatic antioxidants, including catalase and SOD2; (3) by preventing NRF2 ubiquitination and degradation [89].

Insulin sensitivity and glucose homeostasis are also crucial in muscle adaptation to exercise-induced ROS/RNS production. AMPK-dependent phosphorylation of Ser237 in TBC1 domain family member 1 (TBC1D1), a Rab-GTPase-activating protein, enhanced glucose uptake in skeletal muscle fibers during exercise by promoting the translocation of glucose transporter 4 (GLUT4) to the sarcolemma [90,91]. Cytosolic ROS production by NOX2 during moderate-intensity exercise is essential for this effect, as ROS production and GLUT4 translocation to the sarcolemma are impaired in mice with loss-of-function mutations of NOX2 regulatory subunits p47phox or Rac1/gp91phox [92]. Furthermore, improved insulin signaling and glucose tolerance in muscle-specific Keap1 knockout mice demonstrate NRF2 participation in glucose homeostasis of muscle [93]. H_2_O_2_ production by endothelial NOX4 is needed for metabolic adaptation to exercise. Endothelial-Nox4-deficient mice show impaired upregulation of hexokinase 2 and pyruvate dehydrogenase kinase 4 in response to acute exercise, as well as of citrate synthase and β-hydroxyacyl-CoA-dehydrogenase in response to chronic exercise [94].

Different types of muscle fibers vary in their speed of contraction and dependence upon oxidative and nonoxidative mechanisms of energy production: type I muscle fibers are slow-twitch oxidative, while type IIx muscle fibers are fast-twitch glycolytic, and type IIa muscle fibers are fast-twitch oxidative/glycolytic. Differences in oxidative capabilities depend on mitochondrial energetics. Glycolysis is necessary for fast contraction despite being relatively inefficient in terms of moles of ATP produced per mole of glucose. For its part, mitochondrial oxidative phosphorylation, which is slower but more efficient in terms of ATP production, is at the base of slow contraction [95]. The anaerobic sources of ATP provision operate rapidly and are crucial in high-intensity exercise situations such as sprinting, whereas the aerobic pathways of ATP production are dominant during endurance exercise but require time to fully activate [96]. Due to their characteristics, type II fast-twitch muscle fibers produce more ROS/RNS than type I slow-twitch muscle fibers [42]. In murine skeletal muscle, PGC-1α-mediated upregulation of HIF2α in response to exercise promotes the switch from type IIx fast-twitch glycolytic muscle fibers to type IIa fast-twitch oxidative/glycolytic and type I slow-twitch oxidative fibers by inducing the expression of myosin heavy chain type I (MyHC I), myoglobin, calmodulin 2, and troponin I, and repressing that of MyHC IIb, a marker of type IIb ultrafast-twitch glycolytic muscle fibers [97]. Muscle adaptation to contraction-induced ROS/RNS production during resistance training also involves increases in amino acid uptake and protein synthesis, both in positive correlation with MyHC I, as well as in the number of myonuclei per volume of cytoplasm and the percentage of myonuclei in skeletal muscle stem cells (SMSCs) [48]. These effects are dependent on the activation of AKT kinase (protein kinase B)/mechanistic target of rapamycin complex 1 (mTORC1)/p70S6K1 signaling, supporting muscle hypertrophy induced by mechanical load through the promotion of contractile protein synthesis. It is of note that mTORC1 is also a positive regulator of PGC-1α and, through it, mitochondrial biogenesis [98].

As mentioned, the PGC-1α/ERRα complex strongly induces the expression of VEGF and other angiogenic factors, such as platelet-derived growth factor subunit B (PDGFB) and angiopoietin 2 in cultured muscle cells and in skeletal muscle in vivo [80]. VEGF upregulation by PGC-1α/ERRα is both direct and mediated by the muscle-selective induction of hypoxia-induced factor 2α (HIF2α) [99]. Exercise-induced H_2_O_2_ production by endothelial NOX4 contributes to increasing VEGF expression, leading to capillary growth in mice [100].

The proliferation and myogenic differentiation of SMSCs are also redox-regulated by an intricate and not yet fully understood interplay between various ROS-producing enzymes, particularly NOX4, and antioxidant enzymes [59]. In particular, NRF2 promotes muscle regeneration by the proliferation of SMSCs [101]. In skeletal muscle cells, IL-15 controls intracellular ROS production and attenuates oxidative stress [102]. Furthermore, in contracting muscle cells, ROS-mediated signaling stimulates the secretion of myokines, such as IL-15, IL-6, IL-8, insulin-like growth factor-1 (IGF-1), irisin, leukemia inhibitory factor (LIF), monocyte chemotactic protein-1 (MCP-1), and oncostatin-M (OMS), that extend the beneficial effects of exercise to adipose tissue, the brain, blood vessels, the heart, and other organs [103]. Recently, exercise was shown to upregulate the expression of cardiotrophin-like cytokine factor 1 (CLCF1) in both humans and rodents, efficiently counteracting its age-associated decline, improving physical performance, glucose tolerance, and mitochondrial activity in muscle and protecting against age-associated bone decline [104].

Biological responses of muscle cells to contraction-induced increases in ROS/RNS can reflect post-translational oxidative modifications of muscle proteins. Ryanodine receptor 1 (RyR1) is a Ca^2+^ release channel in the sarcoplasmic reticulum, responsible for excitation–contraction coupling in skeletal muscle cells. It functions as a redox sensor thanks to nucleophilic cysteines that exist as thiolate anions at neutral pH. Their reversible S-glutathionylation can occur in the presence of H_2_O_2_ and GSH, in a process that involves their oxidation to sulfenic acid, followed by the formation of a mixed disulfide with GSH. In the presence of high levels of ROS/RNS, S-glutathionylation and mixed S-glutathionylation/S-nitrosylation of RyR1 are performed by oxidized glutathione (GSSG) and S-nitrosoglutathione (GSNO), respectively [30]. In muscle triads (structures consisting of a T tubule and two adjacent terminal cisternae of the sarcoplasmic reticulum) isolated from fast-twitch rabbit skeletal muscle, S-nitrosylation, but not S-glutathionylation, induced the dissociation of the FK506-binding regulatory protein (FKBP12) from RyR1, while both modifications induced the dissociation of Ca^2+^-calmodulin, which in turn stabilized the open channel conformation, thereby increasing Ca^2+^ release into the cytosol [105]. This was confirmed in triads isolated from murine muscle fibers, in which the generation of superoxide radical anion O_2_•^−^ and H_2_O_2_ was stimulated by NADPH [106]. It has been hypothesized that these effects may contribute to maintaining muscle responses to increased exercise [30]. Supporting this view are the increases in strength and resistance to fatigue of the tibialis anterior muscle in S-nitrosoglutathione reductase (GSNOR) knockout mice [107]. However, there is evidence that continued oxidation, S-nitrosylation, hyperphosphorylation, and depletion of FKBP12 during prolonged exercise and high-load work may ultimately reduce exercise tolerance (see also Section 3.3) [108].

Furthermore, in skinned type II fast-twitch skeletal muscle fibers from rat muscles and human muscle biopsies, the in vitro S-glutathionylation of Cys34 on the COOH-terminal domain of the fast troponin I isoform (TnIf) enhanced Ca^2+^ sensitivity by promoting the interaction of TnIf with troponin C at lower Ca^2+^ concentrations [109]. S-glutathionylation and S-nitrosylation had opposite and competitive effects, with the former increasing and the latter decreasing Ca^2+^ sensitivity, and one preventing the action of the other if performed first [110]. Furthermore, S-glutathionylation of TnIf helps maintain Ca^2+^ sensitivity in type II muscle fibers of healthy individuals after high-intensity exercise [111].

### 2.3. Oxidative Stress as a Pathophysiological Response of Muscle Cells in Ageing and Disease

In pathological conditions characterized by oxidative/nitrative stress, such as inflammation, the ROS/RNS produced include free radicals (e.g., superoxide radical anion O_2_•^−^ and hydroxyl radical •OH, nitric oxide •NO, and nitric dioxide •NO_2_), radical-derived species (e.g., peroxynitrite anion ONOO^−^ deriving from the reaction of •NO with O_2_•^−^) and non-radical species (e.g., H_2_O_2_ and hydroxyl anion OH^−^) [112], as discussed in detail in Section 3.2 below. Irreversible oxidative damage to DNA and proteins and deregulation of essential pathways promoting redox homeostasis may result from excessive ROS/RNS generation and/or defective oxidant clearance. Furthermore, as ageing progresses, the ability of cells to resist stress may be progressively compromised [113,114]. It is known that the extent of oxidative damage sharply increases in the last quarter of the life span [47]. The decreased capacity to detoxify ROS/RNS is one aspect of a more general decline in the reaction to damage that occurs in individuals experiencing unhealthy ageing. In this context, the decreased efficiency of innate and adaptive protective immune responses coexists with an altered regulation of tolerance to self-antigens and of the balance between pro- and anti-inflammatory cytokines and chemokines, leading to a chronic low-grade pro-inflammatory state marked by increased serum levels of IL-6, IL-6R, IL-8, TNF-α, TNF-R1, CXCL10, and TGF-α. Immunosenescence, characterized by phagocytic dysfunction, decreased responsiveness to antigenic stimuli, decreased number and efficiency of NK and T cells, and B-cell dysplasia, does not fully define this state, which is referred to as “inflamm-ageing” [115], and is interconnected with the derangements of all the hallmarks of ageing [51]. Nutritional imbalances and malnutrition play a key role in altering the immunological balance in a pro-inflammatory sense and in orienting the ageing trajectory toward frailty and are at the same time correlated with the development of sarcopenia [116]. Indeed, oxidative stress in human skeletal muscle cells is implicated in the pathogenesis of age-related muscle loss in sarcopenia, as progressive functional decline of mitochondria is associated with ROS/RNS accumulation [117,118]. In long-lived white-footed mice (*Peromyscus leucopus*), ROS/RNS production in mitochondria and the level of isoprostanes produced by lipid peroxidation (see Section 3.4, below) were lower and the activity of detoxifying enzymes was higher than in shorter-lived *Mus musculus* mice [119].

In sarcopenia, the decrease in lean muscle mass depends on the reduction in the total number of muscle fibers, particularly type II, resulting from both their numerical reduction and their trans-differentiation into type I fibers [120]. The latter has important metabolic implications and consequences on muscle mass, strength, and physical performance. Indeed, type I slow-twitch muscle fibers have smaller cross-sectional areas and motor unit sizes, and a correspondingly lower force-generating capacity compared to type II fast-twitch muscle fibers, despite their higher mitochondrial content and greater resistance to fatigue. Conversely, type IIa and IIx muscle fibers, which have an intermediate and low mitochondrial content, respectively, have larger cross-sectional areas and motor unit sizes and a greater force-generating capacity, along with greater fatigability, compared to type I fibers [121]. Thigh muscle volume and whole-body D^3^Cr muscle mass, physical performance as measured by peak oxygen consumption during exercise (VO_2_ peak), 400 m walking speed, leg strength, and maximal oxidative phosphorylation capacity were compared to the expression levels of 21 coding mRNAs associated with the oxidative stress response in skeletal muscle biopsies collected from 575 participants aged ≥70 years from the Study of Muscle, Mobility, and Aging (SOMMA). Increased VO_2_ peak and mitochondrial respiration were linked to increased expression of mRNAs encoding superoxide dismutase 2 (SOD2), thioredoxin 2 (TRX2), peroxiredoxin 3 and 5 (PRDX3, PRDX5), and glutaredoxin 2 (GRX2). Furthermore, increased expression of SOD2, PRDX3, and GRX2 was linked to improved muscle growth and athletic performance [122]. These data were consistent with the observation that *Sod1^−/−^* mice lacking Cu,Zn-SOD1 exhibited accelerated sarcopenia, with changes characteristic of aged muscle. The loss of muscle mass was accompanied by a progressive decline in mitochondrial function, increased mitochondrial ROS/RNS production, and more rapid induction of mitochondrial apoptosis and loss of myonuclei. *Sod1^−/−^* mice also displayed a strikingly increased number of dysfunctional mitochondria in the vicinity of neuromuscular junctions (NMJs). These appeared denervated, with dispersed, fragmented endplates and impaired assembly of acetylcholine receptor (AChR) clusters due to transcriptional downregulation. As a result, contractile force was significantly reduced [123]. Compared with 16-week-old young Wistar rats, 40-week-old aged rats showed reduced activities of mitochondrial respiratory complexes in soleus muscle fibers, increased ROS/RNS production, and reduced transcription of genes encoding SOD2, PPAR-γ coactivator-1β (PGC-1β), and sirtuin 1 (SIRT1) [124]. It was proposed that sarcopenia in *Sod1^−/−^* mice may result from a two-hit mechanism affecting both skeletal muscle and motor neurons. The first hit occurs when redox homeostasis is impaired in motor neurons, resulting in NMJ dysfunction, which causes the second hit, i.e., increased production of ROS/RNS in the mitochondria of skeletal muscle cells. Feedback from ROS/RNS onto NMJs causes their further disruption and further increases in ROS/RNS production in a vicious cycle that ultimately results in NMJ disintegration, denervation, and muscle fiber loss (Figure 2) [125].

Age-related deletion mutations and/or mitochondrial ETC abnormalities have been reported as possible primary causes of age-associated mitochondrial dysfunction in sarcopenia. A recent study in OKC-HET rats, which are a product of genetic crosses showing greater divergence in mitochondrial genomes compared to pure inbred strains, revealed increases in both mtDNA half-life and the frequency of mtDNA deletion mutations with age, indicating an age-related decrease in mtDNA turnover [126]. Using a high-throughput droplet digital PCR assay, an average 98-fold exponential increase in the frequency of mtDNA deletion mutations was observed in skeletal muscle cells from 14 healthy individuals aged 20 to 80 years [127]. This study followed a study in 30-month-old male Fischer 344 x Brown Norway F1 hybrid rats, in which induction of mtDNA deletion mutations by the addition of 1% β-guanidinopropionic acid to the diet for 4 months was associated with a 1200% increase in ETC-deficient muscle fibers harboring mtDNA deletion mutations, along with an 18% decrease in muscle fiber number and a 22% worsening of natural muscle mass loss starting at 30 months of age [128]. Attention was drawn to LONP1, a multifunctional protease that regulates mtDNA copy number maintenance, degradation of oxidized proteins, and folding of imported proteins into the mitochondria, and also mediates activation of the PINK1 (PTEN-induced kinase 1)/Parkin pathway required for myoblast differentiation. Specific ablation of LONP1 in murine muscle cells is associated with accumulation of mitochondrially retained proteins and a reduction in muscle fiber size and strength. Mutations in the *LONP1* gene cause a syndrome of cerebral, ocular, dental, auricular, and skeletal anomalies (CODAS), with hypotonia and ptosis as clinical signs of muscle impairment [129].

A genome-wide transcriptional analysis of muscle biopsies from 119 sarcopenic elderly men and age-matched controls showed that sarcopenia is characterized by transcriptional mitochondrial dysfunction in skeletal muscle cells, with downregulated expression of genes of oxidative phosphorylation, proteostasis, and PGC-1α/Estrogen related receptor α (ERRα)-dependent signaling, resulting in a reduced number of mitochondria, decreased activity of respiratory complexes, and low NAD^+^ levels [130]. Furthermore, the concentration of ETC complexes I, III, and IV, partially encoded by mtDNA, was significantly reduced in sarcopenic D257A mice, which express a proof-reading-deficient version of PolgA, the nuclear-encoded catalytic subunit of mitochondrial DNA polymerase gamma, resulting in the accumulation of mtDNA mutations. They exhibited impaired mitochondrial bioenergetics, with reduced ATP production, defective state 3 respiration, and a consequent decline in mitochondrial membrane potential. However, these malfunctions did not coincide with increased ROS/RNS production and oxidative damage. Thus, mtDNA mutations might impair the formation of functional ETC complexes, leading to skeletal muscle cell apoptosis and sarcopenia, independent of oxidative stress [131]. On the other hand, mtDNA alterations alone may not be sufficient to cause sarcopenia. In K320E mice expressing a dominant-negative variant of the mitochondrial helicase Twinkle, higher levels of mtDNA mutations in differentiated muscle cells, but not in quiescent SMSCs, compared to controls, were accompanied by a progressive increase in cytochrome c oxidase-deficient muscle cells in several skeletal muscles, although without an accelerated loss of muscle mass, strength, or physical performance [132].

It has been noted that the mitochondrial genome may be more susceptible to oxidative mutational damage than nuclear DNA due to its proximity to mitochondrial sources of ROS/RNS generation, absence of histones, and limited capacity for mtDNA repair [131]. Proximity may also increase the vulnerability of mitochondrial ETC complexes and matrix enzymes to oxidative damage. Hydroxyl radicals •OH produced in the Fenton reaction from the H_2_O_2_ resulting from the dismutation of the radical anion O_2_•^−^ generated by NADPH oxidases, xanthine oxidase or the reaction of electrons leaking from the ETC with O_2_ can damage both nucleic acids, including mtDNA, and proteins [133]. For example, it has long been known that ROS/RNS-mediated oxidation, S-glutathionylation, and/or S-nitrosylation of critical cysteine residues can negatively affect the catalytic activity of ATP synthase subunits in the inner mitochondrial membrane and promote the opening of the mitochondrial permeability transition pore (mPTP), which is actually a dimer of ATP synthase [134]. Prolonged opening of the mPTP, called mitochondrial permeability transition (MTP), causes mitochondrial osmotic swelling, which ends in cell apoptosis [135,136]. On the other hand, mtDNA fragmentation and accumulation of damaged mtDNA in the absence of DNA repair enzymes, such as DNA ligase III (LIG3) or exonuclease G (EXOG), can directly trigger autophagy and mitophagy, both of which are regulated by oxidative stress [137]. Oxidized mtDNA released from damaged cells can promote further inflammation through activation of inflammasomes [138].

Gene expression data from muscle biopsies from the aforementioned SOMMA study indicated that expression of autophagy-related genes was linked to measures of muscle strength, mobility, fitness, and mitochondrial energetics [139]. Mechanistic links between oxidative stress and cell survival and death have been examined [50,140]. In mammalian cells, autophagy is initiated by a molecular complex that includes the serine–threonine kinase Unc-51-like kinase-1 (ULK1). Increases in autophagy in the presence of high concentrations of ROS/RNS are regulated by the mechanistic target of rapamycin complex 1 (mTORC1) and the AMP-activated protein kinase (AMPK). ULK1 phosphorylation by mTOR (catalytic subunit of mTORC1) disrupts its interaction with AMPK and inhibits autophagy, whereas AMPK-mediated ULK1 phosphorylation and mTORC1 inactivation increase ULK1 activity and promote autophagy [141]. Ageing is known to alter the activity of AMPK, which in skeletal muscle cells controls several processes in response to the state of energy resources, namely the intracellular AMP/ATP ratio, via important modulators of protein turnover, such as sirtuins (SIRTs) and forkhead box O (FOXO) transcription factors. The importance of AMPK as a redox-sensitive kinase [142,143] has been clearly demonstrated in AMPK-deficient cells, which showed accelerated senescence concomitant with increased levels of ROS/RNS in mitochondria [144]. In sarcopenia, increased ROS/RNS can prevent the phosphorylation of AKT kinase, mTOR, and the downstream mTOR targets p70S6K and E4E-BP1, thus impairing the ability of muscle cells to adapt to exercise [137,145]. In senescent skeletal muscle cells, mitophagy is one aspect of a general deregulation of mitochondrial metabolism and dynamics, which also involves inefficient oxidative phosphorylation, oxidative stress, and regenerative failure with loss of SMSCs [146]. SMSCs preserve muscle capacity for adaptation and regeneration. Quiescent SMSCs maintain low levels of ROS/RNS by relying on energy generated by glycolysis. However, with ageing, SMSC proliferation and differentiation become less efficient [113,114]. This is partly due to the decline of autophagy, which is essential for maintaining stemness by preventing senescence [147], and partly to age-related changes that occur in several redox-sensitive signaling pathways [113,148]. In aged SMSCs, decreased mitochondrial bioenergetics and mtDNA damage may result from epigenetic silencing of the ageing suppressor α-Klotho [149]. A presenescent state characterized by reduced SMSC numbers, loss of proteostasis, mitochondrial dysfunction, and oxidative stress has been observed in aged mice in which overexpression of the cyclin-dependent kinase inhibitor p16^INK4A^ is driven by genetic loss of the BMI1 component of the polycomb repressive complex 1 (PRC1) [147]. Other conditions of defective myoblast production and depletion of the SMSC pool in aged muscle include nuclear localization of p27^KIP1^, which maintains autophagy when being retained in the cytoplasm following phosphorylation by AMPK [150], chronic activation of p38α/β MAPK [151], and JAK-STAT3 signaling [152].

Furthermore, sarcopenia is associated with alterations in the extracellular matrix (ECM) and a fatty infiltration into the skeletal muscle called myosteatosis [153]. Mesenchymal progenitor cells (MPCs), also called fibro-adipogenic progenitors (FAPs), are major stromal contributors to these ECM and fibro-adipose changes [120]. Ageing is associated with depletion of WNT1-inducible signaling pathway protein 1 (WISP1), which is normally deposited in muscle ECM by FAPs and is important for SMSC expansion and asymmetric commitment to myogenic differentiation via a cell division mode in which one daughter cell self-renews as a stem cell, while the other differentiates into a muscle cell [154]. Induction of senescence in 2G11 cells, clones of rat MPCs, by H_2_O_2_-induced oxidative stress led to the loss of fibro-adipogenic potential. Co-culture of myoblasts with senescent 2G11 cells prevented myotube formation, accelerating the onset of sarcopenia [155].

ROS/RNS promote muscle fatigue and contribute to muscle cell dysfunction, death, and inflammation in various chronic diseases [156,157]. Among 669 people in the Baltimore Longitudinal Study of Ageing, those with lower mitochondrial oxidative capacity showed more significant inflammation compared to those with better capacity [158]. As discussed in Section 2.2, NRF2, a master regulator of redox homeostasis, influences the development of sarcopenia by regulating inflammation and mitochondrial biogenesis and turnover. NFR2- and NF-κB-mediated signaling pathways functionally interact. While NF-κB controls NRF2 transcription and activity, NRF2 deficiency can enhance the pro-inflammatory effects of NF-κB activation [159,160]. Indeed, ROS/RNS stimulate the production of TNF-α and IL-1β through the activation of NF-κB-mediated pathways in skeletal muscle cells [161] and are involved in the loss of contractile strength caused by TNF-α [139,162]. IL-6 and TNF-α have been repeatedly shown to be associated with sarcopenia [163]. They can block protein synthesis in skeletal muscle cells, thus compromising muscle integrity and function [164]. Furthermore, the degradation of myofibrillar proteins by calpains and caspases can be activated by ROS/RNS [22,165]. A NOX-dependent overproduction of ROS/RNS has been observed in diabetic skeletal muscle cells, where ROS/RNS-stimulated glucose uptake may contribute to oxidative stress [22,166]. ROS/RNS have also been implicated in Duchenne’s muscular dystrophy [167].

In summary, oxidative stress can cause mitochondrial dysfunction by altering mitochondrial dynamics and quality control, which in turn leads to widespread damage to cellular lipids, proteins, and nucleic acids (see Section 3, below). Oxidative stress can also trigger abnormal accumulation of pro-inflammatory cytokines and activation of neutrophils and macrophages, which lead to the production and release of cytotoxic proteases and additional pro-inflammatory cytokines and ROS/RNS. At the same time, oxidative stress impairs cellular antioxidant capacity and prevents the proliferation and differentiation of SMSCs, causing losses of muscle strength, muscle quantity/quality, and physical performance. Oxidative stress acts by directly altering multiple molecular targets and affecting numerous signaling pathways involving muscle cell metabolism, protein turnover, and cell death and regeneration. Redox imbalance appears to be a major cause of the multiple tissue, cellular, and molecular disorders that characterize sarcopenia. Redox homeostasis relies on the dual role of ROS/RNS as signaling mediators involved in metabolic regulation and muscle cell adaptation and as toxic compounds [63]. However, providing a mechanistic description of what may make ROS/RNS homeostatic regulators or pathogenetic determinants of sarcopenia remains a challenging task [168]. While disruption of mitochondrial redox signaling contributes to ageing and a wide range of pathological conditions, including sarcopenia, emerging evidence highlights the beneficial effects of moderate increases in ROS/RNS levels induced by exercise (oxidative eustress) as a stimulus not only for adaptation to resistance training, but also for prolonging lifespan and muscle cell regeneration during ageing [45,63]. This gives rise to a new paradigm of oxidative stress, whose possible benefits in combating sarcopenia have yet to be fully exploited.

## 3. Molecular Mechanisms and Targets of Oxidative Damage in Sarcopenia

### 3.1. Molecular Pathogenesis of Sarcopenia: An Overview

Skeletal muscle is the most abundant tissue of the human body, contributing approximately 40% of body weight and approximately 30% of basal energy expenditure. The loss of muscle strength, muscle quantity/quality, and physical performance in sarcopenia is caused, at the molecular level, primarily by reduced production of myofibrillar and mitochondrial proteins and their increased proteolysis through the ubiquitin–proteasome system and the calcium-dependent activation of calpain and caspases. Furthermore, dysregulation of mitochondrial dynamics through mitochondrial fusion, fission, and autophagy is associated with a higher rate of mitochondrial degradation compared to biogenesis. Finally, sarcopenia is also characterized by an imbalance between muscle cell regeneration and apoptosis. These processes are promoted by upstream variables, such as insulin resistance, elevated production of pro-inflammatory cytokines, and reduced release of anabolic hormones. Overall, chronic oxidative stress appears to be critical [169]. ROS/RNS, advanced lipid peroxidation end products (ALEs) and advanced glycation/glycoxidation end products (AGEs, see below) contribute to sarcopenia by targeting essential constituents of skeletal muscle cells, such as proteins and DNA/RNA. Furthermore, they interact with several scavenger receptors (see Section 3.6 below).

### 3.2. ROS/RNS and Their Sources

There is more than one mechanism of cellular and tissue oxidative damage in sarcopenia. The direct action of ROS/RNS against essential constituents of muscle cells, such as lipids, carbohydrates, proteins, and DNA/RNA, represents the first step in the molecular mechanisms through which oxidative stress contributes to the development of the disease. The main responsible ROS/RNS are the following:

*Superoxide radical anion O*_2_•^−^: O_2_•^−^ is the product of the one-electron reduction of molecular oxygen O_2_ [24]:O_2_ + e^−^ → O_2_•^−^(1)

O_2_•^−^ is produced primarily in mitochondria as a byproduct of mitochondrial respiration due to the leakage of electrons along the ETC, particularly from respiratory complexes I and III. Its production is also catalyzed by NADPH oxidase in phagocytes, for O_2_-dependent killing of microbes, and by dual oxidases (DUOXs) in thyroid epithelial cells as an intermediate in H_2_O_2_ synthesis:2 O_2_ + NADPH → 2 O_2_•^−^ + NADP^+^(2)

O_2_•^−^ is also produced by tyrosinase in melanocytes, as a byproduct of the conversion of l-tyrosine to l-DOPA and l-DOPA-quinone, and by other enzymes, such as cyclooxygenase, lipoxygenase, and XO.

Furthermore, O_2_•^−^ can originate from the autoxidation of reduced transition metals:Fe^2+^ + O_2_ → Fe^3+^ + O_2_•^−^
(3)

In aqueous solution, O_2_•^−^ is in equilibrium with its protonated form, the hydroperoxyl radical HO_2_•, with the anion largely predominating at neutral pH. HO_2_• can initiate the radical chain reaction of lipid peroxidation (see Section 3.4, below). Moreover, O_2_•^−^ can catalyze the liberation of Fe^2+^ from proteins containing [Fe–S] clusters, thereby inactivating them and promoting the Fenton reaction. The superoxide radical anion O_2_•^−^ is converted to H_2_O_2_ and O_2_ by enzymatic dismutation catalyzed by superoxide dismutases (SODs):2 O_2_•^−^ + 2 H_2_O → H_2_O_2_ + 2 OH^−^ + O_2_(4)

O_2_•^−^ reacts rapidly with nitric oxide •NO to produce the peroxynitrite anion (ONOO^−^) (see below).

*Hydrogen peroxide (H*_2_*O*_2_*)*: H_2_O_2_ is the product of the spontaneous or enzymatic dismutation of O_2_•^−^ catalyzed by SODs (reaction 4). It can be harmful to cells even at concentrations as low as 10 nM in neurons and 100 μM in muscle cells [63] and readily diffuses across cell membranes. Fe^2+^- or Cu^2+^-catalyzed cleavage of H_2_O_2_ in the Fenton reaction is a major source of hydroxyl radicals •OH under physiological conditions [170].

*Hydroperoxyl radical HO*_2_•: HO_2_• (•OOH) can be formed by the interaction of radical species with cellular constituents, such as lipids and nucleobases. The hydroperoxyl radical HO_2_• is produced when the superoxide radical anion O_2_•^−^ is protonated, and it is the most basic of peroxyl radicals:O_2_•^−^ + H^+^ → HO_2_•(5)

About 0.3% of total O_2_•^−^ is in the protonated state in the cytoplasm of a typical cell. It can initiate the peroxidation of unsaturated fatty acid (see Section 3.4, below) and potentially promote tumor growth.

*Hydroxyl radical •OH*: •OH is produced by the Fenton reaction [170,171]:Fe^2+^ + H_2_O_2_ ⟶ Fe^3+^ + •OH + OH^−^(6)

Another source of •OH is the non-enzymatic reaction of O_2_•^−^ with H_2_O_2_ (Haber–Weiss reaction):O_2_•^−^ + H_2_O_2_ ⟶ O_2_ + •OH + OH^−^(7)

An additional amount of •OH is produced, along with nitrogen dioxide •NO_2_, by the decomposition of peroxynitrite ONOO^−^ in an acidic environment, via the peroxynitrous acid ONOOH intermediate (reaction 12, below).

•OH reacts strongly with biological macromolecules and can cause serious cellular damage.

*Hypochlorous acid (HClO)*: HClO is a weak acid that partially dissociates in water:HClO ⇆ ClO^−^ + H^+^(8)

HClO oxidizes sulfhydryl groups in proteins. A single sulfhydryl group can eliminate up to four HClO molecules, in a stepwise oxidation process that produces sulfenic acid (R-S-OH), sulfinic acid (R-SO_2_H), sulfonic acid (R-SO_3_H), and finally a disulfide with another protein -SH group. HClO can oxidize disulfides to sulfinic acid. HClO also reacts with secondary amino groups of amino acid side chains, as well as with heterocyclic -NH groups and secondary amino groups of nucleobases.

*Nitric oxide (nitrogen monoxide, •NO)*: •NO is a radical produced enzymatically by the three isoforms of nitric oxide synthase (NOS): neuronal (nNOS), endothelial (eNOS), and inducible (iNOS). •NO formed by NOS diffuses readily in cytoplasm and across cell membranes due to its amphiphilicity. Its multiple physiological roles include regulation of vasomotor tone through stimulation of protein kinases and guanylate cyclase in blood vessels, and participation in important metabolic reactions, such as the conversion of l-arginine to l-citrulline. Many of these effects are mediated by the reversible *S*-nitrosylation of the thiol groups of cysteine residues in proteins, primarily by *S*-nitrosylases, resulting in the formation of *S*-nitrosothiols (NSO), which is an important aspect of enzyme regulation and cell signaling.

•NO formed by NOS reacts with O_2_•^−^ to produce the peroxynitrite anion (ONOO^−^).•NO + O_2_•^−^ ⟶ ONOO^−^
(9)

ONOO^−^ is responsible for the nitration of tyrosine residues in proteins [172].

*Peroxynitrite anion (ONOO*^−^*)*: ONOO^−^ displays a reactivity similar to HClO and is lipid-soluble. As mentioned previously, it is formed by the reaction of O_2_•^−^ with •NO (reaction 9) [173].

In the presence of CO_2_, peroxynitrite generates the nitrosoperoxycarboxylate anion (ONOOCO_2_^−^):ONOO^−^ + CO_2_ ⟶ ONOOCO_2_^−^
(10)

ONOOCO_2_^−^ is a highly reactive toxic compound, the decomposition of which produces nitrogen dioxide (•NO_2_) and the carbonate radical anion CO_3_^−^• [174]:ONOOCO_2_^−^ ⟶ •NO_2_ + O=C(O•)O^−^(11)

In an acidic environment, ONOO^−^ decomposes into nitric dioxide (•NO_2_) and the hydroxyl radical •OH via the peroxynitrous acid (ONOOH) intermediate:ONOO^−^ + H^+^ ⟶ ONOOH ⟶ •NO_2_ + •OH (12)

Peroxynitrite and other nitrating agents, such as the myeloperoxidase (MPO)-H_2_O_2_-NO_2_^−^ system, are also oxidizing agents [174].

*Nitric dioxide (nitrogen dioxide, •NO*_2_*)*: •NO_2_ is a radical and a strong oxidant produced by the decomposition of peroxynitrite in an acidic environment (reaction 12, above). Its disproportionation in water generates nitric acid and nitric oxide:3 •NO_2_ + H_2_O → 2 HNO_3_ + •NO(13)

As seen, •NO_2_ is also generated by the decomposition of the nitrosoperoxycarboxylate anion (reaction 11), along with the carbonate radical anion CO_3_^−^•; reactions with •NO_2_ derived from peroxynitrite cause the nitration of proteins, lipids, and purine bases [175].

Caloric overload increases cellular and body fluid levels of ROS/RNS generated as byproducts of mitochondrial respiration through the leakage of electrons from the mitochondrial ETC, while inflammation increases ROS/RNS production by phagocytes and endothelial cells during the oxidative burst and the biosynthesis of arachidonic-acid-derived inflammatory mediators [24]. It is worth noting that tobacco smoke and air pollution, including exhaust fumes from internal combustion engines and refineries, and fumes generated by the combustion of biomass fuels, are rich sources of ROS/RNS such as H_2_O_2_, •OH, •NO, •NO_2_, and a variety of organic radicals that can be inhaled (see Section 3.4 below).

### 3.3. Direct Modifications of Biological Macromolecules by ROS/RNS

During oxidative stress, the metal-catalyzed oxidation of amino acid side chains can cause the conversion of arginine and proline to glutamic semialdehyde (GSA), lysine to aminoadipic semialdehyde (AASA), and threonine to amino-ketobutyrate. This results in the irreversible introduction of new carbonyl (C=O) groups into proteins, which are amenable to spectrometric or antibody-mediated detection once they have formed protein-bound 2,4-dinitrophenylhydrazones by reaction with 2,4-dinitrophenylhydrazine (DNPH). Neoformation of carbonyl groups in proteins can also result from the formation of stable adducts through the reaction of lysine residues with reducing sugars or reactive carbonyl species (RCS) resulting from free-radical attack on derivatives of lipid and carbohydrate metabolism (see Section 3.4 and Section 3.5, below) [170]. After undergoing conformational changes, oxidized proteins become more susceptible to the action of proteases [176]. Another reliable indicator of protein oxidation is the formation of 3-nytrotyrosine (3-NT). RNS are also oxidants, and protein nitration is a sign of oxidative damage by RNS. The formation of 3-NT occurs when the peroxynitrite anion ONOO^−^, produced by the reaction of •NO with O_2_•^−^ (reaction 9, above), nitrates tyrosine residues in proteins. Compared to younger controls, older animals had higher amounts of 3-nitrotyrosine in their total proteins from the external intercostal and quadriceps muscles [177].

Oxidative stress also promotes the oxidation of cysteine residues in key muscle proteins. A recent redox proteomic analysis by mass spectrometry on muscle biopsies from the aforementioned SOMMA study revealed that the oxidation of cysteines in proteins crucial for myocellular function was associated with worse performance in several mobility metrics [178]. Reversible *S*-nitrosylation of cysteine thiols in proteins is a prototypical redox-based homeostatic mechanism implicated in the regulation of numerous signaling pathways and enzymatic reactions. Although cysteine *S*-nitrosylation appears to be targeted and limited with the aid of specific *S*-nitrosylases, it is nevertheless a process conditioned by the abundance of •NO as a substrate and is therefore capable of mediating cellular responses to the increased production of this important effector and product of oxidative/nitrosative stress. We have already discussed in Section 2.2 the possible regulatory significance of reversible *S*-glutathionylation and *S*-nitrosylation of critical cysteine residues in the ryanodine receptor RyR1. However, in 24-month-old rats, intracellular calcium leakage from the sarcoplasmic reticulum, which occurs as a result of oxidation of the RyR1 Ca^2+^ channel (as inferred from new carbonyl formation), *S*-nitrosylation, and depletion of the Ca^2+^ channel-stabilizing component FKBP12 (calstabin1), in fast-twitch fibers were associated with reduced muscle strength and exercise capacity, apparently contributing to age-related loss of muscle function. *S*-nitrosylation of RyR1 in the presence of elevated ROS/RNS concentrations impaired the interaction between FKBP12 and RyR1, thereby increasing Ca^2+^ influx from the sarcoplasmic reticulum into the cytoplasm. Skeletal muscle contractility decreased as the Ca^2+^-depleted sarcoplasmic reticulum released less Ca^2+^, while elevated cytoplasmic Ca^2+^ exacerbated RyR1 *S*-nitrosylation, creating a vicious cycle [179]. Furthermore, upregulated iNOS-fueled *S*-nitrosylation of glyceraldehyde-3-phosphate dehydrogenase in C2C12 cells triggered skeletal muscle cell death [180]. Altered *S*-nitrosylation of p53 may also play a role in sarcopenia and other muscle disorders [181]. Conversely, muscle-specific overexpression of calpastatin, an endogenous inhibitor of Ca^2+^-dependent calpains, significantly reduced sarcopenia. *S*-nitrosylation of calpain in nonsarcopenic adult muscle prevents the breakdown of structural and regulatory proteins in myofibrils, but *S*-nitrosylation decreases and calpain-mediated proteolysis increases with age, suggesting that •NO regulation may offer an opportunity to prevent muscle loss in ageing [182]. Thus, ROS/RNS can be beneficial or detrimental to muscle, as discussed in Section 2.1 above. Distinguishing between oxidative eustress and distress may offer opportunities to counteract sarcopenia.

Oxidative stress also affects mitochondrial and nuclear DNA and RNA. Adenine, guanine, deoxyribose, and ribose are the most vulnerable to oxidative damage among DNA/RNA constituents, followed by thymine and cytosine [183]. Transcriptional and translational abnormalities and cellular malfunctions are caused by oxidative damage to nucleic acids, the main markers of which are 8-oxo-deoxyguanosine (8-OHdG) for DNA and 8-oxo-oxyguanosine (8-OHG) for RNA. Sarcopenia is associated with increased oxidative damage to DNA, particularly in mtDNA, in both rodents and humans [169].

### 3.4. Lipid Peroxidation and Advanced Lipid Peroxidation End Products (ALEs)

Lipid peroxidation (LPO) refers to reactions between free radicals and polyunsaturated fatty acids (PUFAs), particularly those present in the plasma membrane of cells. It consists of three phases: initiation, propagation, and termination (Figure 3). A chain reaction begins with the formation of a carbon-centered initiating lipid radical L•, following free-radical attack on the fatty acid (FA) by an oxygen radical, primarily •OH. Addition of O_2_ to L• during the propagation phase converts it to a lipid peroxyl (or lipoperoxyl) LOO• radical. A lipid hydroperoxide (LOOH) is formed when this LOO• radical is sufficiently reactive to abstract hydrogen from a second FA. Under normal atmospheric pO_2_ conditions, the rate of the latter reaction is much lower than that of LOO• formation, so the steady-state concentration of LOO• is much greater than that of L•. Consequently, termination occurs exclusively by reaction between two lipoperoxyl radicals LOO•, but in theory it can occur by reaction of a lipoperoxyl radical LOO• with a lipid radical L•, or between two lipid radicals L• [124,169]. Lipid hydroperoxides (LOOHs) are unstable: they generate new peroxyl and alkoxyl radicals and decompose to highly reactive degradation products, mainly reactive aldehydes. These include acrolein (ACR), malon(yl)dialdehyde (MDA), hexanal, 4-hydroxy-2,3-nonenal (4-HNE), and the small reactive aldehydes glyoxal (GO) and methylglyoxal (MGO). Free radicals produced during LPO have very local effects, due to their short half-lives, but secondary decomposition products of lipid hydroperoxides may serve as “second messengers”, or secondary intermediates of oxidative stress, due to their prolonged half-lives and their greater ability to diffuse from their sites of formation compared to free radicals [184,185]. Peroxidation and subsequent breakdown of membrane lipids with formation of reactive aldehydes, can dramatically alter the permeability and fluidity of cellular lipid bilayers, thus affecting cellular integrity. Gamma-hydroxyalkenals formed by the peroxydation of PUFAs in phosphatidylcholine and other phospholipids in cell membranes (as well as LDL) contribute to these effects [186].

Advanced lipid peroxidation end products (ALEs) are formed when LPO-derived reactive aldehydes form covalent adducts with proteins and DNA. HNE forms covalent hemiacetalic adducts via the Michael reaction with histidine, cysteine, and lysine residues in proteins, or pyrrole adducts via Schiff base formation with lysyl amino groups. HNE modification can also lead to cross-linking of two lysine residues via reversibly formed Schiff base Michael adducts, as well as irreversibly formed 2-hydroxy-2-pentyl-1,2-dihydropyrrol-3-one iminium groups (Figure 4). Covalent modification by acrolein affects the sulfhydryl group of cysteines. MDA interacts with primary amines to form *N^ε^*-(2-propenal)lysine, as well as lysine–lysine cross-links with 1-amino-3-iminopropene and pyridyledihydropyridine bridges. Addition of GO and MGO to the amino groups of lysine residues yields *N^ε^*-carboxymethyllysine (CML) and *N^ε^*-carboxyethyllysine (CEL), respectively. Thus, LPO’s products are RCS, the addition of which to nucleophilic groups introduces new carbonyl groups in proteins, which can be detected after conversion to 2,4-dinitrophenylhydrazone adducts by reaction with 2,4-dinitrophenylhydrazine (DNPH). The formation of such adducts can lead to the functional inactivation of proteins that are crucial for vital cellular processes, such as ion and nutrient transport across the plasma membrane, energy metabolism (glycolysis, mitochondrial electron transport, and oxidative phosphorylation), cell signaling, cytoskeletal organization, antioxidant defenses, stress responses, protein synthesis, signal transduction, and regulation of neurotransmission [186]. Reactive aldehydes also attack free −NH_2_ groups of DNA bases to form pro-mutagenic exocyclic DNA adducts, which likely contribute to the mutagenic and carcinogenic effects associated with oxidative stress-induced LPO [187].

LPO products can be introduced into the body from external sources. MDA and HNE are present in all foods that contain lipids. In high-fat foods, such as vegetal and animal oils, meat products, and fried foods, the content of MDA may vary between 0.1 and 10 mg/kg, and that of HNE between 0.01 and 1 mg/kg. Since PUFAs, and among them ω-6 fatty acids, are the precursors of MDA and HNE, respectively, the lower their proportion, the lower the amount of MDA and HNE in foods with the same degree of fat oxidation [188]. The contents of MDA and HNE increase with food processing, especially with heat processing. Frying and barbecuing produce the highest amounts of MDA and HNE, while boiling produces the least. The concentrations of MDA and HNE in foods fried at temperatures between 160 °C and 190 °C depend mainly on time and temperature, as well as on the culinary oil (soybean, olive, rapeseed, corn, sunflower, or linseed oil) used and the oil absorption capacity of the foods. MDA and HNE are formed faster and accumulate more at higher temperatures. Other aldehydes, such as acrolein, crotonaldehyde, 4-hydroxy-2-hexenal, 2,4-hexadienal, 2,4-heptadienal, 2-octenal, nonanal, and 2,4-decadienal can occur in mg/kg amounts in foods such as French fries and fried pork. Furthermore, repeated use of frying oil leads to further accumulation of aldehydes in the oil itself and in fried foods. To reduce their formation, it is necessary to increase the frequency of frying oil replacement and reduce the frying temperature [188]. Autoxidation and decomposition of unsaturated fatty acids are also significant during food drying, even at temperatures below 40 °C, and can be massive during hot air drying (50–90 °C). Aldehydes and ketones are also formed in abundance during the roasting of meat and vegetable oils, coffee beans, hazelnuts, nuts, and seeds at temperatures between 150 and 300 °C, but to a greater extent during roasting with direct exposure to an open flame and during the grilling of protein-rich sausages, steaks, and pork chops at 250 °C. Seafood is more vulnerable to oxidative deterioration due to its high content of PUFAs. The formation mechanism, detection methods, and toxicological effects of LPO products in foods have been reviewed [189]. Motor vehicle exhaust, particularly from diesel engines, in urban areas is an important source of aldehyde pollutants in the air, emitted directly or formed through the photochemical decomposition of emitted hydrocarbons, such as benzene. These include alkenals such as acrolein and crotonaldehyde, and α-oxoaldehydes such as GO and glycoladehyde. Other sources of aldehydes in the air include emissions from forest fires, agricultural waste burning, incinerators, and coal-fired power plants [190]. GO, MGO, acrolein, crotonaldehyde, and other saturated and unsaturated aldehydes can reach levels in residential and occupational environments an order of magnitude higher than in outdoor air, and their sources include the burning of biomass fuels (wood, crop residues such as straw) and coal for residential heating, domestic water heating, and cooking, as well as cigarette smoke [191,192]. The main components of cigarette smoke, in decreasing order (μg per cigarette), include acetaldehyde, acrolein, formaldehyde, crotonaldehyde, diacetyl, and MGO. The external sources, molecular toxicity mechanisms, and human health effects of reactive aldehydes have been reviewed [190]. It is difficult to distinguish human exposure to aldehydes from exogenous sources versus endogenous causes, as components in cigarette smoke can stimulate lipid peroxidation in vivo. For instance, exposure of human bronchial epithelial cells to increasing concentrations of cigarette smoke extract induced dose-dependent carbonyl neoformation in a large number of proteins as detected by mass spectrometry [193]. Analogously, exposure to diesel engine exhaust induced systemic lipid peroxidation, as judged from the levels of 8-isoprostanes, 12-hydroxyeicosatetraenoic acid (12-HETE), and 13-hydroxyoctadecadienoic acid (13-HODE) in plasma and MDA in the liver of apoE-deficient mice exposed to diesel exhaust [194] and the urinary levels of MDA and etheno-DNA adducts, such as 1,*N*6-etheno-2′-deoxyadenosine (εdA), among occupationally exposed workers [195].

As the association and possible pathogenetic relationship between the formation of ALEs and various neurodegenerative, inflammatory, and autoimmune diseases became better understood [196], the determination of GSA, AASA, CML, CEL, and *N^ε^*-MDA-lysine (MDAL) by chromatography coupled to mass spectrometry (GC/MS) and the spectrometric or antibody-mediated detection of protein-bound 2,4-dinitrophenylhydrazones, MDAL, and 4-HNE adducts with cysteine, histidine, and lysine residues in proteins have been widely exploited as measures of lipid peroxidation [197]. Since lipid hydroxyperoxyl radicals, in addition to the formation of aldehydes, undergo endocyclization to F2-isoprostanes (F2-IsoPs, including 8,12-iso-iPF2α-VI) and F4-neuroprostanes (F4-NPs), the levels of these compounds in cells and tissues have also been used as a measure of increased lipid peroxidation [198]. Detection technologies have evolved over time towards increasing sensitivity and accuracy through the combined use of capillary liquid chromatography (capLC) or nanoflow liquid chromatography (nanoLC) coupled with triple quadrupole tandem mass spectrometry (QQQ/MS/MS), or nanoelectrospray ionization tandem MS (nanoLC-NSI/MS/MS) with stable isotope dilution (SID) [199].

Several studies in experimental animals and humans highlight the pathogenetic role of apoptosis in the muscle cell loss observed in age-related sarcopenia and the role in this process of reactive LPO products, whose levels in cells and body fluids increase with age [200]. Quantitative redox proteomics was used to study the neoformation of carbonyl groups in mitochondrial proteins of Fischer 344 rat skeletal muscle, which depends on age and muscle fiber type (slow-twitch vs. fast-twitch). Compared to slow-twitch fibers, fast-twitch fibers had double the level of protein carbonyls. Age-dependent changes in carbonylation status were detected in 22 proteins, most of which had increased carbonylation levels and were related to cellular maintenance, fatty acid metabolism, and the citric acid cycle [201]. Ageing and inactivity have been associated with increased production of lipid hydroperoxides (LOOHs) in mice. Muscle weakness and atrophy were induced by LOOHs in a proteasomal-independent and lysosome-dependent manner; conversely, they were prevented in young and old mice by genetic and/or pharmacological neutralization of LOOHs or their aldehyde derivatives [202]. Levels of MDA, 4-HNE, and other 4-hydroxyalkenals (HAE) were higher in control muscles of old adult mice compared to young ones [32,203]. Administration of 5 mg/kg lipostatin-1, a LOOH scavenger, to adult male mice demonstrated the pathogenic role of LOOHs in denervation-induced muscle atrophy in vivo. Lipostatin-1 treatment preserved overall gastrocnemius muscle mass and muscle fiber cross-sectional area. It also reduced mitochondrial hydroperoxide formation in denervated and permeabilized muscle fibers by 80% in vitro and >65% in vivo and lowered the amount of 4-HNE formed after denervation by ~25% [204]. Furthermore, in various animal models, the level and activity of cytosolic phospholipase A2 (cPLA2) and downstream metabolites, including LOOHs, were increased by denervation, while atrophy was not prevented by increased mitochondrial H_2_O_2_ scavenging, indicating that ETC-generated H_2_O_2_ was not crucial. On the other hand, in vivo suppression of cPLA2 reduced oxidative damage, prevented LOOH formation, and preserved muscle fiber size [205].

In humans, 4-HNE content in sedentary older individuals was significantly higher than in young, active older individuals, reflecting an age-associated decline in antioxidant responses due in part to dysfunctions in redox signaling mediated by NRF2 and its inhibitor Keap1, which may be preserved by a recreationally active lifestyle in older adults [206]. Proteins involved in protein quality control and glycolytic enzymes were major targets of carbonylation via the formation of ALEs and advanced glycoxidation end products (AGEs, see Section 3.5 below) in human senescent muscle satellite cells [207]. Furthermore, target proteins of these modifications were identified in human rectus abdominis muscle obtained from healthy young and older human donors. In biopsies from older donors, 17 of the observed protein spots were shown to be more carbonylated than those in younger donors. Key cellular processes, such as muscle contraction, energy production and metabolism, cell shape, and ion transport, were mediated by these proteins [208]. Furthermore, more MDA–protein adducts were detected in the intercostal and quadriceps muscles of older individuals compared to younger individuals [177]. Reported evidence indicates that concomitant factors, such as an inefficient scavenger capacity of endogenous antioxidants, poor dietary habits, and a sedentary lifestyle, cooperate in triggering and maintaining mitochondrial oxidative stress and chronic inflammation that results in the development of muscle ageing and sarcopenia.

### 3.5. Advanced Glycation/Glycoxidation End Products (AGEs)

The formation of advanced glycation/glycoxidation end products (AGEs) occurs through a process that can be divided into three steps (Figure 5) [209]. 1. The first step is characterized by the Maillard reaction, which begins with the slow formation of Schiff bases, early unstable precursors of AGEs resulting from the condensation of the electrophilic carbonyl group of a reducing sugar with the free amino groups of lysine or arginine residues in proteins. With reducing aldoses, such as d-glucose, this is followed by the rearrangement of the unstable bases to stable ketoamines via the enol-to-keto group conversion (Amadori rearrangement) (Figure 5A). The analogous reaction that occurs with reducing ketoses, such as d-fructose, is the Heyns rearrangement. 2. In the second step of AGE formation, the still unstable products thus formed undergo non-enzymatic reactions that convert them to reactive dicarbonyl intermediates, i.e., RCS. For example, 1-deoxyglucosone (1-DG) and 3-deoxyglucosone (3-DG) can be formed by the enolization and dehydration of d-glucose and d-fructose; GO is produced by retroaldol condensation and the breakdown of d-glucose, either directly, or via autoxidation of a glycolaldehyde intermediate [210]; MGO is formed by the non-enzymatic phosphate elimination and spontaneous decomposition of glyceraldehyde 3-phosphate (G3P) and dihydroxyacetone phosphate (DHAP) deriving from glycolysis, or of G3P deriving from fructolysis. Importantly, reactive carbonyl intermediates can be produced by autoxidation of free or protein-bound sugars, often catalyzed by trace metals, such as Fe^3+^ or Cu^2+^, which is why AGEs formed in this way are referred to as advanced glycoxidation end products. The generation of AGEs by autoxidation of monosaccharides, Schiff bases (aldimins), and Amadori and Heyns products is called the Wolff pathway, the Namiki pathway, and the Hodges pathway, respectively (Figure 5B). Non-enzymatic oxidative reactions of early glycation products are favored over those involving free sugars. 3. In the third step, these reactive carbonyl intermediates react irreversibly with peptides/proteins to form stable AGEs and protein cross-links. Addition of GO to the amino groups of lysine residues and to the thiol groups of cysteine residues in proteins produces *N^ε^*-carboxymethyllysine (CML) and carboxymethylcysteine (CMC), respectively. CML may also derive from the oxidative cleavage of fructose–lysine, the Amadori adduct formed by isomerization of the protein glycation product with glucose. In contrast, addition of MGO to the same amino acid functional groups produces *N^ε^*-carboxyethyllysine (CEL) and carboxyethylcysteine (CEC), respectively (Figure 5C). Since GO and MGO could derive from free-radical attack on derivatives of carbohydrate and lipid metabolism, their adducts with lysine residues in proteins, such as CML and CEL, are referred to as mixed AGEs/advanced lipoxidation end products (ALEs) [211]. Furthermore, the addition of 3-DG to the amino group of lysine produces pyrraline. These reactions produce new 2,4-diphenylhydrazine (DNPH)-reactive carbonyl groups in proteins, amenable to detection by mass spectrometry or antibodies. Protein cross-linking products are also formed through the simultaneous addition of the aforementioned RCS to two amino acid residues. Bis(lysyl)imidazolium derivatives include the imidazolium cross-links between two lysines and GO (GOLD), two lysines and MGO (MOLD), and two lysines and 3-DG (DOLD). Conversely, the simultaneous addition of GO, MGO, and 3-DG to a lysine and an arginine residue produces the corresponding imidazolium cross-links, called GODIC, MODIC and DODIC, respectively. Note that arginine is converted to ornithine during cross-linking reactions. Pentosidine is a cross-link between Lys and Arg, formed with the Maillard reaction product of ribose. Pentosidine is fluorescent, as is MOLD (Figure 5C) [212].

There is a close interplay between AGE formation and oxidative stress. On the one hand, non-enzymatic oxidative reactions of free and protein-bound sugars and the formation of AGEs are accelerated by high glucose and ROS concentrations. On the other hand, binding of AGEs to their multiligand receptor, RAGE, induces ROS production, endothelial cell dysfunction, and activates inflammatory signaling cascades [213]. Furthermore, AGEs can combine with metal ions, such as Cu^2+^ and Fe^2+^, providing catalytic sites for the generation of ROS and RNS [214]. Importantly, lifestyle factors significantly increase human exposure to AGEs. Diets high in carbohydrates and calories and a sedentary lifestyle favor the endogenous formation of AGEs, while diets high in high-temperature processed foods result in the intake of AGEs formed through the Maillard reaction between sugars and proteins contained in the foods themselves, the formation of which is characterized by browning and the production of flavor and aroma. Dietary sources of AGEs, mechanisms of AGE formation during food processing and endogenous AGE production, and the health effects of AGEs have been reviewed alongside the methods for measuring AGEs in biological samples and in vivo [215].

Several studies have explored the effects of AGE formation on muscle ageing and in sarcopenia. Serum pentosidine level was one of the biomarkers of ageing associated with decreased appendicular lean mass in older women and has been suggested as a biomarker for sarcopenia [216]. Elevated serum pentosidine levels were independently associated with a higher prevalence of sarcopenia in middle-aged and older Chinese men with type-2 diabetes mellitus [217]. In middle-aged and older people from a Dutch population-based cohort, dietary consumption of AGEs, including CML, CEL, and MGH1 (MGO-derived hydroimidazolone 1, an adduct of MGO with arginine), was positively correlated with the prevalence, but not with the incidence, of sarcopenia, although no correlation was found between high dietary AGE intake and physical frailty [218]. Higher levels of skin AGEs measured by skin autofluorescence (SAF) were associated with a higher prevalence of sarcopenia in a cohort of 2744 elderly individuals of Northern European origin [219]. A similar association was reported in a cohort of 1991 elderly Chinese individuals, partially mediated by osteoporosis [220]. A systematic review of fourteen cross-sectional and one prospective observational study showed a negative correlation between muscle-related outcomes, such as muscle mass, strength, and physical function, and AGEs measured by SAF or circulating AGEs in adults aged ≥ 30 years [221]. At the cellular and molecular levels, there is evidence of an association between glycative stress and motor and muscle dysfunction. Furthermore, induction of AGE production in C2C12 skeletal muscle cells by 5-day treatment with glyoxylate, pyruvate, glycolaldehyde, and glucose caused N^ε^-CML accumulation and suppression of myotube formation. These effects were associated with disruption of signal transduction characterized by upregulation of STAT3 signaling via increased phosphorylation of Tyr705 and downregulation of both ERK signaling via reduced phosphorylation of Thr202/Tyr204, and insulin/insulin-like growth factor 1 (IGF-1) signaling. Similar effects on STAT3 and ERK signaling were also observed in mice fed a high-AGE diet for 16 weeks [222]. Marked cytotoxic effects also resulted from endogenous AGE overproduction induced in C2C12 myoblasts by 24 h treatment with glycolaldehyde-modified bovine serum albumin [223]. These effects were apparently mediated by suppression of MyoD, myogenin, and endogenous IGF-1 mRNA expression, and by inhibition of IGF-I-induced AKT activation [224]. AGEs have been reported to promote sarcopenia through changes in the basal lamina and vasculature, where abnormal cross-linking of AGE-modified collagen played a primary role, stiffening the muscle fiber microenvironment. AGE-dependent changes negatively impacted motor neurons, satellite cells, neuromuscular junctions (NMJs), and Schwann cells [225]. The reduction in muscle contractile activity appeared to be mediated by the modifications in myosin and actin, as well as collagen. Decreased myogenic capacity, suppression of protein synthesis, and increased protein degradation contributed to the loss of muscle mass, strength, and physical performance caused by AGEs, whose accumulation in the body has been proposed as a risk factor for decreased motor function [226]. Furthermore, sarcopenia has been frequently observed in patients with chronic kidney disease (CKD), where it was independently associated with morbidity and mortality. CKD is characterized by increased production and reduced renal excretion of AGEs, which are thought to be responsible for the progression and complications of CKD by promoting ROS overproduction, inflammation, and fibrosis, and potentially contributing to the loss of muscle mass, strength, and physical performance [227]. The role of AGEs in muscle ageing and sarcopenia has been reviewed [228].

### 3.6. Interactions of Oxidized Macromolecules with Scavenger Receptors

The reactive carbonyl species produced by the peroxidation of lipids in cell membranes and lipoproteins and by the glycoxidation of free and protein-bound sugars, as well as the AGEs/ALEs deriving from their addition to nucleophilic groups in proteins, bind to several scavenger receptors, with important consequences for the development of human diseases, a prominent example of which is atherosclerosis. Minimally oxidized LDLs (MM-LDL) contain oxidized lipids such as lysophosphatidylcholine; phosphatidylcholine breakdown products with truncated sn-2 acyl chains; their epoxy, hydroxy, and hydroperoxy derivatives; and cholesteryl esters of hydroxy and hydroperoxy derivatives of fatty acids. They undergo endocytic uptake and degradation by binding to class E scavenger receptor LOX-1 (lectin-like oxidized low-density lipoprotein receptor, SCARE1) expressed on endothelial cells and scavenger receptor B3 (SCARB3, CD36) on smooth muscle cells (SMCs), macrophages, and endothelial cells. They also bind to toll-like receptor 4 (TLR4) on macrophages. These interactions are potent triggers of endothelial dysfunction and inflammation [229]. Extensively oxidized LDLs (oxLDLs) contain abundant oxidized lipids and apolipoprotein B (ApoB) modified by the addition of ACR, MDA, and HNE, or other free or phospholipid-esterified alkanals and alkenals. They undergo uptake and degradation by binding to class A scavenger receptors SCARA1, SCARA2 (macrophage receptor with collagenous structure, MARCO), and CD36 [230]. In macrophages, the interaction between CD36 and oxidized LDLs induces the phosphorylation of the non-receptor tyrosine kinase Lyn and the activation of Jun-kinases (JNK) 1 and 2. The guanine nucleotide exchange factor Vav, the small GTPase immunity-related GTPase family M member 1 (IRGM1), and protein kinase ERK 1/2 also participate in the signaling cascades that mediate the internalization of oxLDL bound to CD36 [229]. MDA- and HNE-modified LDLs are the main cause of foam cell formation from macrophages [230]. These undergo cytoskeletal modifications, following the binding of oxLDL and oxidized phospholipids to CD36, that lead to their entrapment in atherosclerotic plaques [231]. RCS adducts in atherosclerotic lesions of humans and experimental animals have been revealed by immunohistochemistry with anti-MDA and anti-HNE antibodies [232].

The AGE receptor (RAGE, SCARJ1) is a 50–55 kDa integral membrane glycoprotein expressed in monocytes/macrophages, endothelial cells, fibroblasts, neurons, and smooth muscle cells. More than 20 isoforms with diverse biological functions resulting from alternative splicing have been described. In addition to AGEs, RAGE ligands include high-mobility group box 1 (HMGB1, a chromatin-binding protein), amyloid-β peptide, calcium-binding protein S100/calgranulin, and lysophosphatidic acid (LPA). Soluble RAGE (sRAGE) can competitively inhibit ligand-dependent RAGE signaling [233]. AGE-RAGE interactions trigger multiple signaling cascades, involving NADPH oxidase, PI3K/AKT/glycogen synthase kinase 3β (GSK-3β), Ras/MEK/ERK 1/2, Cdc42/MKK6/p38 MAPK, Cdc42/MKK4/7/SAPK/JNK, and JAK/STAT. These lead to the activation of various transcription factors, including NF-κB, early growth response-1 (Egr-1), AP-1, and STAT3 [234]. These interactions increase oxidative stress in muscle by activating NADPH oxidases [235] either directly or via protein kinase C-dependent pathways that stimulate NADPH oxidases and lipoxygenases [236]. RAGE normally supports adult myogenesis, by inducing the myoblast proliferation via the ERK1/2 pathway in response to S100B binding, and myogenic differentiation and myoblast fusion via the p38 MAPK pathway, in response to HMGB1 [237]. However, pathological upregulation of AGE production—as observed in unhealthy ageing, hyperglycemia, and long-term oxidative stress—and chronic inflammation, can promote abnormal muscle stem cell differentiation and muscle atrophy. The detrimental effects of AGE overproduction on muscle cells appear to reflect both the structural and functional damage directly inflicted by AGEs on muscle proteins [208], as well as the overstimulation of RAGE, which leads to NF-κB activation, the release of pro-inflammatory mediators such as IL-6, TNF-α, and C-reactive protein (CRP) [238], and increased protein degradation through activation of the ubiquitin–proteasome system (UPS) and autophagocytosis [239]. Transcription of UPS-related genes is stimulated by TNF-α-dependent release of myostatin, a member of the TGF-β family expressed primarily in muscle, which binds to the sarcolemmal activin receptor 2 (ActRIIB), thereby inducing activin receptor-like kinase (ALK)-4 and -5-mediated Smad2/3 phosphorylation [240]. Furthermore, pro-inflammatory cytokines inactivate the PI3K/AKT/mTOR pathway, inducing insulin resistance and reduced protein synthesis, while reduced AKT phosphorylation stimulates muscle proteolysis by caspase-3 and the UPS. Furthermore, NF-κB-dependent FOXO3 activation leads to the proteolysis of myosin, actin, and other filament proteins, inducing the expression of muscle-specific E3 ubiquitin ligases, such as atrogin-1/muscle atrophy F-box (MAFbx) and muscle RING finger protein 1 (MuRF1), and autophagy-related (Atg) proteins. RAGE signaling is also involved in skeletal muscle regeneration after acute injury [237]. AGEs exerted a negative regulation of myogenesis in myotubes differentiated from mouse myoblasts and primary human skeletal muscle progenitor cells, inducing myotube atrophy and atrogin-1 expression via RAGE-dependent AMPK phosphorylation and AKT dephosphorylation, which could be counteracted by sRAGE [241]. Other receptors that bind AGEs include AGE receptor 1 (AGE-R1/OST-48), AGE-R2, and AGE-R3 (galectin-3), and the scavenger receptors SCARA1, SCARA2 (MARCO), SCARB1, SCARB3 (CD36), FEEL-1 (SCARH2) and FEEL-2 (SCARH1). With the exception of AGE-R2 and AGE-R3, all of them are able to mediate the endocytic uptake of AGEs and AGE-modified proteins [242]. AGE-R1 exerts a protective role by competing with RAGE for AGE binding, suppressing RAGE expression and counteracting cellular oxidative stress via dephosphorylation and activation of FOXO3 [243]. In contrast, the role of CD36 in skeletal muscle is still controversial and needs to be clarified. Furthermore, in a cohort of 189 older adults (mean age 77.19 ± 6.12 ys), CD36 mRNA levels in peripheral blood mononuclear cells were independently and positively correlated with pre-frailty and frailty status, although not with appendicular skeletal muscle mass (ASM) [244].

## 4. Biochemical Mechanisms of Containment and Disposal of Oxidants in Muscle Cells

The regulation of redox homeostasis is entrusted to various effectors of both endogenous and exogenous origin. Cells resist oxidative stress through a variety of complementary mechanisms, including the following:(1)Detoxification by enzymatic antioxidants.(2)Detoxification by non-enzymatic antioxidants.(3)Expression of protein chaperones, including heat-shock proteins (HSP) and glucose-regulated proteins (GRP).(4)Removal of damaged molecules by lysosomal or proteasomal digestion and autophagy.(5)Expression of DNA repair enzymes.(6)Expression of anti-apoptotic factors and mitochondrial membrane stabilizing factors.(7)Expression of growth factors, such as brain-derived neurotrophic factor (BDNF), glial-derived neurotrophic factor (GDNF), IGFs, soluble amyloid β precursor protein (AβPP) in brain, and cytokines, such as TNF-α. These activate the expression of genes encoding antioxidant enzymes, anti-apoptotic proteins, and regulators of ion transport (e.g., calbindin and glutamate receptors).

In this section we will focus on endogenous enzymatic and non-enzymatic antioxidants.

### 4.1. Endogenous Enzymatic Antioxidants

The most important endogenous enzymatic antioxidants in the context of sarcopenia are the following:

*Superoxide dismutases (SODs)*: Three isoforms of superoxide dismutase are known in humans: SOD1, SOD2 and SOD3. SOD1 is a dimeric enzyme expressed in the cytosol and mitochondrial intermembrane space, inhibited by cyanide (CN^−^) and H_2_O_2_; SOD2 is a tetramer expressed in the mitochondrial matrix, inhibited by H_2_O_2_; SOD3 is a tetramer expressed in the extracellular space, inhibited by CN^−^ and H_2_O_2_ [245]. These enzymes catalyze the dismutation of the superoxide radical anion O_2_•^−^ [246]:2 O_2_•^−^ + 2H^+^ → H_2_O_2_ + O_2_(14)

SOD1 and 3 use Cu^1+^ and Zn^2+^ as cofactors, while SOD2 uses Mn^2+^. In catalysis, O_2_•^−^ is both oxidized to O_2_ and reduced to H_2_O_2_. From 15% to 35% of the total SOD activity in skeletal muscle cells is found in the mitochondria, with the remaining 65% to 85% in the cytosol [245]. Small glycolytic type IIx fibers have the lowest levels of SOD activity compared to oxidative type I or type IIa fibers, and muscles with a higher proportion of oxidative fibers have increased SOD activity [28,247]. SOD activity in skeletal muscle can be influenced by the history of muscle activity [248].

*Catalase (CAT)*: CAT is an antioxidant homotetrameric and ubiquitous enzyme expressed in peroxisomes, the cytosol, and mitochondria of some tissues, which catalyzes the breakdown of H_2_O_2_ into H_2_O and O_2_:2 H_2_O_2_ → 2 H_2_O + O_2_(15)

CAT has high catalytic activity but a low affinity for H_2_O_2_, making it important when H_2_O_2_ levels are very high [249]. Muscle fibers with a high oxidative capacity have the highest CAT activity, whiles those with a low oxidative capacity have the lowest [247]. Whether endurance exercise influences CAT activity in skeletal muscle is controversial.

*Glutathione peroxidases (GPXs)*: There are five isoforms of glutathione peroxidase (GPX15) in humans. GPX1 is a tetrameric enzyme ubiquitously expressed in the cytosol and mitochondria; GPX2 is a cytosolic tetramer expressed primarily in the stomach and intestine; GPX3 is a tetramer ubiquitously expressed in the extracellular space and cytosol; GPX4 is a monomer expressed in the nuclear membrane and mitochondria of testes, spermatozoa, heart, and brain; GPX5 is a dimer expressed in the plasma membrane of epididymis, spermatozoa, liver, and kidney. The reaction catalyzed by these enzymes is the reduction of H_2_O_2_ or organic hydroperoxide (ROOH) to H_2_O or alcohol (ROH), respectively, using GSH as the electron source. In the reaction, two GSH molecules are oxidized to glutathione disulfide (GSSG):2 GSH + H_2_O_2_ → GSSG + 2 H_2_O(16)
or2 GSH + ROOH → SSH + ROH + H_2_O(17)

GPX expression varies in different types of skeletal muscle fibers. In particular, mouse type IIx muscle fibers with limited oxidative capacity exhibit the lowest GPX levels, while type I oxidative fibers exhibit the highest GPX activity [250]. Furthermore, skeletal muscles recruited during exercise training exhibit increased GPX activity.

*Glutathione reductase (GR)*: GR, also known as glutathione disulfide reductase (GSR), is a homodimeric enzyme containing a FAD prosthetic group that regenerates GSH in cells by reducing one molar equivalent of GSSG to two molar equivalents of GSH, using NADPH as an electron donor [251]:GSSG + NADPH + H^+^ → 2 GSH + NADP^+^(18)

*Glutathione S-transferases (GSTs)*: GSTs are a superfamily of ubiquitously expressed detoxifying enzymes that act both as catalysts for the conjugation of GSH to toxic reactive compounds, promoting their inactivation and elimination, and as ligandins for toxic non-reactive compounds.

*Peroxiredoxins (PRDXs)*: PRDXs have been implicated in muscle ageing and sarcopenia in several ways. In cells, peroxides are largely scavenged by PRDXs [252]. These enzymes utilize cysteine residues (called peroxidatic Cys) for catalysis and do not require a cofactor like CAT and GPXs. Furthermore, PRDXs are expressed in human cells at higher levels than other enzymatic antioxidants and scavenge over 90% of peroxides in the mitochondria and over 99% of peroxides in the cytosol [250,253]. The number and location of conserved, redox-sensitive Cys residues within PRDXs provide the basis for their categorization into three families. In both typical and atypical two-cysteine (2-Cys) PRDXs, a peroxidatic cysteine directly reduces a variety of peroxide substrates, while a resolving cysteine restores the peroxide-reducing activity of the peroxidatic cysteine. In typical PRDXs, the peroxidatic and resolving cysteines are found in two distinct molecules, thus functioning as homodimers whose subunits are linked by a disulfide bond; in atypical PRDXs, which are monomeric, the same molecule contains both the resolving and peroxidatic cysteines. The third class of PRDXs includes one-cysteine (1-Cys) PRDXs, which contain only the peroxidatic cysteine and lack the resolving cysteine [254]. There are six isoforms of PRDX: PRDX1, PRDX2, PRDX3, and PRDX4 are typical 2-Cys peroxiredoxins—PRDX1/2 are usually found in the cytosol, PRDX3 in the mitochondria, and PRDX4 in the extracellular space and endoplasmic reticulum; PRDX5 is an atypical 2-Cys peroxiredoxin, mainly present in the cytosol, mitochondria and peroxisomes; and PRDX6 is a 1-Cys peroxiredoxin present in the cytosol and lysosomes [253]. The catalytic activity of PRDXs involves five chemical events and two distinct conformational states. The chemical steps are: (1) peroxidation, (2) resolution, (3) recycling, (4) hyperoxidation, (5) resurrection; the two conformational states are the fully folded (FF) state and the locally unfolded (LU) state [250,252,253,255] (Figure 6).

*Thioredoxins (TRXs)*: TRXs are highly conserved inducible enzymatic antioxidants. Two TRX isoforms exist in human cells: cytosolic TRX1 and mitochondrial TRX2. Both enzymes control oxidative stress by reducing disulfide bonds in proteins and enzymes, in a process that involves the formation of a reduced enzymatic state (TRX1r–TRX2r) as a reaction intermediate, thanks to electrons supplied by NADPH. TRXs provide peroxyredoxins and methionine sulfoxide reductases (MSRs), among others, with the electrons they need to perform their antioxidant activities [256,257]. Oxidized TRXs utilize electrons from NADPH to recover from their oxidized forms, either directly or mainly through the action of thioredoxin reductases (TRXRs), which are antioxidant enzymes with an FAD prosthetic group and a selenylsulfide bond in their active site [258] (Figure 7).

Numerous physiological roles of TRX have been identified in the context of skeletal muscle ageing and sarcopenia, including preventing protein oxidation, reducing transcription factors, mitigating oxidative stress, and controlling apoptosis [259,260,261,262].

*Glutaredoxins (GRXs)*: GRXs are thiol–disulfide oxidoreductases that also participate in maintaining cellular redox balance. They are glutathione (GSH)-dependent enzymes with overlapping activity and a structure similar to thioredoxins. GRXs reduce various protein disulfides (PS-SP) and mixed GSH-protein disulfides (GS-SP) through two different processes, using GSH as an electron donor. Class I GRXs utilize the dithiol-type process based on the Cys-Pro-Tyr-Cys motif contained in their active site, while class II GRXs utilize the monothiol-type process exploiting the unique N-terminal cysteine of the Cys-Gly-Phe-Ser motif in their active site. Furthermore, GRXs are the catalysts of glutathionylation, a post-transcriptional modification involving the disulfide exchange between GSSG and protein thiols (P-SH). GRX-catalyzed glutathionylation is the major defense mechanism against the irreversible oxidation of cysteine residues and represents a significant event in signal transduction. High levels of oxidative stress increase protein glutathionylation, which effectively counteracts the adverse effects of ROS/RNS by reducing the rate of irreversible and damaging cysteine modification [258,263].

Human cells contain five different GRXs. The GRXs most implicated in the development of sarcopenia are GRX1, which is mainly found in the cytosol, and GRX2 and GRX5, both localized in the mitochondria [260]. Knockout of *Grx1* in mice causes skeletal muscle atrophy through impaired control of intramuscular lipid deposition and glucose consumption. Therefore, modulation of GRX1 expression or activity could represent a therapeutic target in disorders, such as sarcopenia, characterized by reductions in muscle strength, muscle quantity/quality, and physical performance [264]. GRX2 exists in two isoforms: GRX2a targets the mitochondria, while GRX2b operates in the nucleus. GRX2 is a highly efficient catalyst of monothiol-type processes that has a strong affinity for mixed protein–GSH disulfides and is not hindered by oxidation of structural cysteine residues. Furthermore, thioredoxin reductase (TRXR), which supports both monothiol- and dithiol-type processes, can supply electrons to GRX2 in addition to GSH and TRXs [262]. Several studies have revealed that GRX2 mediates reversible modification of mitochondrial proteins in response to changes in the redox state of the GSH pool in the mitochondrial matrix [265,266,267]. Furthermore, mice with *Grx2* knockdown are protected from diet-induced weight gain, an effect associated with increased stage 3 (phosphorylating) respiration, increased proton leak, and improved protection from oxidative stress in muscle mitochondria [268]. The absence of *Grx5* in yeast is associated with oxidative damage mediated by iron accumulation in cells and inactivation of enzymes that depend on [Fe–S] clusters for their function. Indeed, GRX5 is a component of the mitochondrial machinery required to synthesize and assemble [Fe–S] centers [269].

*Heme oxygenase 1 (HO-1)*: HO-1, the inducible form of heme oxygenase, also known as heat-shock protein 32 (HSP32), is the rate-limiting enzyme in bilirubin production that catalyzes degradation of heme, producing CO, ferrous iron (Fe^2+^), and biliverdin-IXa, which is further reduced by biliverdin reductase-A (BVR-A) to bilirubin-IXa, a potent antioxidant [270]. Expression of the gene encoding HO-1 is redox-regulated and is induced by ROS/RNS, ischemia, heat shock, and lipopolysaccharide (LPS). Apparently, adaptive increases in HO-1 expression and its activation via serine phosphorylation induced by oxidative stress can be counteracted by structural alteration and functional impairment of HO-1, via tyrosine nitration and the formation of HNE adducts with HO-1. HO-1 activation has been shown to protect against or reduce sarcopenia by promoting anti-inflammatory, antiapoptotic, and antioxidant responses that stimulate angiogenesis, prevent cellular differentiation, and trigger mitochondrial autophagy [271].

*Aldehyde dehydrogenases (ALDHs)*: ALDHs are key components of important biosynthetic pathways and are essential for cellular homeostasis. They are detoxifying enzymes that protect against reactive aldehydes resulting from LPO, blocking their accumulation and damaging effects on cells and tissues. Nineteen ALDH isoforms have been described, of which ALDH2 is the most expressed in the mitochondria of liver, cardiac, and nerve cells [272]. A conserved and highly reactive cysteine residue in the active site of ALDH (Cys302) functions as a catalytic thiol that undergoes water-mediated deprotonation by Glu268; subsequent nucleophilic attack on the aldehyde carbonyl by the thiolate group of Cys302 leads to the formation of a thiohemiacetal intermediate, coupled to hydride transfer to the cofactor NAD(P). Hydrolysis of the resulting thioester intermediate by nucleophilic attack on the carbonyl by a water molecule bound to Glu268 releases the previous aldehyde substrate, now a carboxylic acid, and the thiol-containing enzyme, which, in turn, releases NAD(P)H and is regenerated by NAD(P) [272]. ALDH1A1, ALDH2, and ALDH3A have been shown to oxidize the carbonyl group of HNE [273]. Another important function of ALDHs is their participation in the synthesis of NAD(P)H, a critical electron donor that acts both as a direct radical scavenger and as an electron supplier for biological reactions, such as the production of O_2_•^−^ or H_2_O_2_ by NADPH oxidases, and the detoxification of metabolites and xenobiotics by P450 enzymes. A protective role for ALDH2 in post-cardiac arrest myocardial dysfunction and ischemia/reperfusion injury has been suggested, involving suppression of mitochondrial ROS/RNS production, through oxidative inactivation of HNE [274].

### 4.2. Endogenous Non-Enzymatic Antioxidants

The most important non-enzymatic antioxidants in the context of sarcopenia are as follows:

*Reduced glutathione (GSH)*: The water-soluble tripeptide γ-glutamic acid–cysteine–glycine (glutathione, GSH) is the most biologically important example of a thiol that functions as a preventive antioxidant. Its reduced form eliminates hydroperoxides formed under aerobic conditions, with the catalysis by glutathione peroxidases (GPXs). In this process, GSH serves as a stoichiometric reducing agent and is oxidized to GSSG. GSH reacts with oxygen radicals, as well as with non-radical singlet oxygen (^1^O_2_), hypochlorous acid, and the peroxynitrite anion. Furthermore, GSH plays a crucial role in iron metabolism/homeostasis, DNA synthesis, gene expression, redox homeostasis, antioxidant protection, detoxification, cell signaling, protein cysteine metabolism, and cell proliferation and differentiation or death, including apoptosis and ferroptosis. Ageing is generally associated with decreased GSH levels in cells and tissues, and a positive correlation has been observed between GSH levels and physical and mental health [251].

*Coenzyme Q_10_ (CoQ_10_, ubiquinone-10)*: Coenzyme Q_10_ is a lipophilic biochemical cofactor and a free-radical-scavenging antioxidant, that is both produced by the human body and obtained from dietary sources. It consists of a 1,4-benzoquinone moiety and an isoprenoid side chain with 10 isoprenyl units. CoQ_10_ is a component of the ETC in the inner mitochondrial membrane, where it acts as a mobile electron carrier, transferring electrons from complex I (NADH-ubiquinone reductase) and complex II (succinate-ubiquinone reductase) to complex III (ubiquinol-cytochrome-c reductase). Stepwise reduction converts CoQ_10_ (ubiquinone-10) to the partially reduced intermediate semiquinone radical CoQ_10_H• (ubisemiquinone-10) and then to the fully reduced CoQ_10_H_2_ (ubiquinol-10) (Figure 8). The ability of CoQ_10_ to act as both a two-electron carrier (moving between the quinone and quinol forms) and a one-electron carrier (moving between the semiquinone and each of the other two forms) is critical to its role in both electron transfer between complexes I, II, and III, whose [Fe-S] clusters accept only one electron at a time, and as a free-radical-scavenging antioxidant. Peroxyl ROO• (lipoperoxyl LOO•) radicals in cell membrane lipids can be scavenged by CoQ_10_H_2_, which is critical for the prevention of ferroptosis [275]. Preclinical data suggest a potential role of CoQ_10_ in several conditions that predominantly affect older adults, including sarcopenia, but well-planned and -conducted randomized case–control studies are needed to evaluate the impact of CoQ_10_ supplementation on clinically meaningful outcomes in sarcopenia [276].

*Uric acid*: Uric acid exhibits both antioxidant and pro-oxidant properties in vitro. The urate monoanion, which is the predominant form at physiological pH, contributes 30–50% of the total serum antioxidant capacity. It reacts with peroxyl radicals ROO• but, unlike l-ascorbate, is unable to reduce the α-tocopheroxyl radical α-TO• back to α-T. A cross-sectional study determined that uric acid could contribute to increased oxidative stress independently of xanthine oxidoreductase activity by increasing ROS/RNS production [277].

### 4.3. NRF2, the Master Regulator of Antioxidant Responses

Specific treatment is required for NRF2, the master inducible regulator of oxidative stress responses. Regulation of NRF2 expression by ROS/RNS is one of the most prominent examples of a redox-sensitive gene regulatory event, a definition that encompasses several ways in which ROS/RNS mediate the induction and promoter-specific recruitment of transcriptional activators and chromatin-modifying enzymes, along with increased ubiquitinylation and reduced levels and promoter occupancy by counteracting transcriptional repressors and modifying enzymes. Some of these have been discussed in Section 2.2 above. NRF2 is constitutively targeted for proteasomal degradation through binding to Keap1, which maintains it at low levels under typical eustress conditions. Oxidants, electrophiles, and, in general, disturbances of redox homeostasis inhibit NRF2 proteasomal degradation and permit the translocation of NRF2 to the nucleus, where it catalyzes the transcription of important antioxidant genes by binding to AREs in their promoters. Keap1 loses its function when exposed to compounds like tert-butyl hydroquinone (tBHQ) or sulforaphane (SFN), which alter the reduction state of some highly reactive Cys residues (Cys151, Cys226/Cys613, Cys-273/Cys-288, Cys-434). Keap1 can therefore be considered a sensor of oxidative stress [85,278]. •NO, H_2_O_2_, acrolein, 4-hydroxynonenal, prostaglandins, fumarate, cyclopentanone, nitro-fatty acids, and 8-nitroguanosine are endogenous thiol-reactive electrophiles and oxidants that can also deactivate Keap1 [279,280,281]. NRF2 regulation is mediated by several mechanisms besides Keap1, such as interactions with other transcription factor or proteins, polymorphisms in its gene promoter, and post-translational modifications, such as phosphorylation and acetylation.

NRF2 plays a critical role in redox homeostasis, iron/heme metabolism, metabolic regulation, mitochondrial energy production, drug/xenobiotic metabolism, protein quality control, DNA damage repair, prevention of apoptosis, and inflammation. NRF2-mediated redox homeostasis involves (a) the reductive regeneration of cysteine thiol groups in proteins and enzymes that have been oxidized to sulfenic acid (CysSOH), sulfinic acid (CysSO2H), or disulfides (S-S) during oxidative stress; (b) the induction of genes encoding essential antioxidant effectors, such as glutathione reductase 1 (GSR-1), PRDXs, TRXs, GRXs, and NADPH-regenerating agents, all of which facilitate the catalytic and non-catalytic scavenging of ROS/RNS [86,87,282]. Downregulation of NRF2 transcriptional activity is a driving force of sarcopenia, while its upregulation helps maintain mitochondrial homeostasis and skeletal muscle strength and quantity/quality. NRF2 deficiency exacerbates mitochondrial and muscle dysfunction in aged mice by dysregulating cellular autophagy, senescence, and apoptosis [283,284]. *Nrf2*-knockout mice suffer from increased fatigue and decreased muscle strength compared to wild-type animals [285,286]. NRF2 absence in the context of ROS/RNS overproduction impairs muscle contraction [287]. Conversely, upregulation of NRF2/HO-1 expression inhibits skeletal muscle cell death, counteracting muscle atrophy and fibrosis [288]. As seen in Section 2.2, NRF2 opposes the development of sarcopenia by regulating mitochondrial biogenesis and turnover [84] and by promoting SMSC proliferation [101].

A novel emerging role of NRF2 in sarcopenia is linked to sestrins [289]. Sestrins are a family of evolutionarily conserved, stress-inducible proteins encoded by the *SESN1*, *SESN2*, and *SESN3* genes. Their products reduce oxidative stress and control metabolic homeostasis by regulating the AMPK and mTOR pathways [290], being distinctly downregulated with ageing [291]. The NRF2/ARE pathway regulates the induction of sestrin-2 expression [292] and, in turn, sestrin-2 activates NRF2 by promoting the p62-dependent autophagic degradation of Keap1 [293]. Sestrin-1 overexpression is sufficient to prevent muscle atrophy, but its downregulation induced by inactivity or its genetic absence exacerbates muscle wasting. Protection occurs through mTORC1 inhibition, which upregulates autophagy, and through the activation of AKT, which inhibits FOXO-regulated ubiquitin–proteasome-mediated proteolysis [294]. Furthermore, exercise reduces the chronic inflammatory response induced by a high-fat diet in an NRF2- and sestrin-2-dependent manner [295].

**Figure 8 ijms-26-07787-f008:**
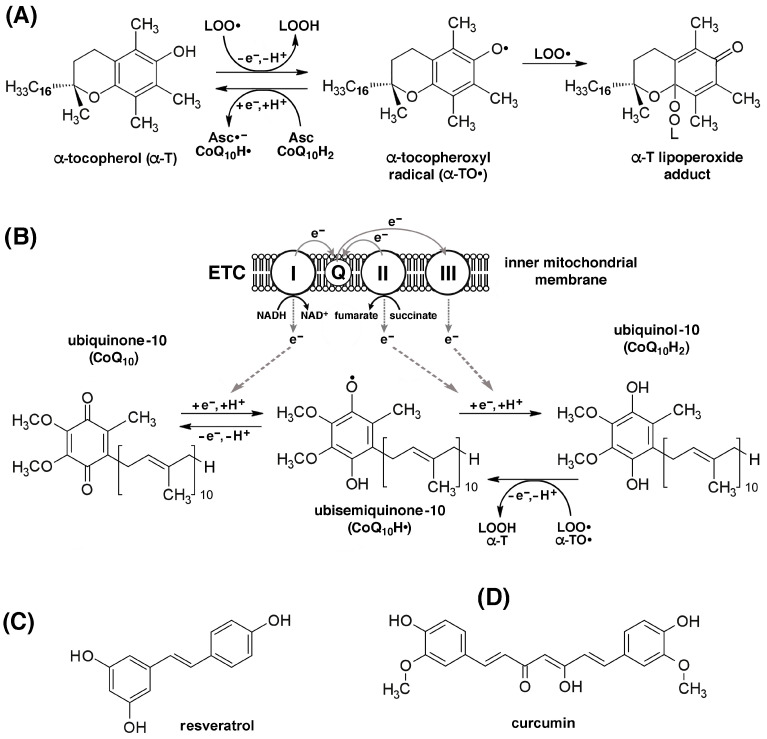
Structures and mechanisms of action of the main phenolic radical-trapping antioxidants: (**A**) α-tocopherol (vitamin E): the scavenging of a lipoperoxyl radical LOO• by α-tocopherol (α-T), with formation of an α-tocopheroxyl radical α-TO• is illustrated; the scavenging of a second radical LOO•, with formation of a lipoperoxide adduct of α-T, and the reductive regeneration of α-T by l-ascorbate (Asc) or coenzyme Q_10_H_2_ (ubiquinol-10) are also illustrated. (**B**) The stepwise reduction of coenzyme Q_10_ (CoQ_10_, ubiquinone-10) to the ubisemiquinone radical CoQ_10_H• and subsequently to the fully reduced CoQ_10_H_2_ (ubiquinol-10) is depicted, via electrons “leaking” from respiratory complexes I, II, and III of the mitochondrial electron transport chain (ETC). The scavenging of a lipoperoxyl radical LOO• and the regeneration of α-T by CoQ_10_H_2_ are also shown. (**C**) The structure of resveratrol; (**D**) the structure of curcumin.

## 5. A Critical Reassessment of Possibly Useful Exogenous Antioxidants in Sarcopenia

### 5.1. A Chemical Mechanistic and Physiological Classification of Antioxidants

Determining cost-effective strategies to preserve muscle strength, muscle quantity/quality, and physical performance in the elderly is one of the most significant public health concerns. Improving the quality of life of patients affected by sarcopenia by preventive strategies and therapeutic measures may help lower the financial burden associated with the disease. Physical exercise is the most successful preventive measure [296,297,298,299]. However, many elderly people are unable to engage in training activities either because of comorbidities or due to the lack of motivation, limiting this intervention on a wider scale. In this context, the use of natural antioxidants, either alone or in combination, has been repeatedly recommended. Some, like l-ascorbate (vitamin C), coenzyme Q_10_ (CoQ_10_H_2_, ubiquinol-10), retinol, and carotenoids (vitamin E), act as ROS/RNS scavengers, while others, like resveratrol, lipoic acid, and ergothioneine, act as both scavengers and indirect antioxidants by affecting enzymatic activities and/or signal transduction pathways. Further strategies include the administration of biosynthetic precursors of main endogenous antioxidants, such as GSH. As soon as radical-mediated oxidation of biological macromolecules became implicated in all kinds of degenerative, metabolic, inflammatory, and neoplastic diseases as well as ageing, nutritional biochemists, pharmacologists, and physicians undertook a massive quest for natural or synthetic antioxidant molecules for preventive or therapeutical purposes. However, simple replenishment of pools of electron donor molecules in cells and body fluids may not necessarily succeed, as the biochemistry of antioxidants is more complex than generally understood [248,300], as it needs to be placed in the context of physiological chemistry and within an adequate pharmacokinetic and pharmacodynamic framework.

Antioxidants can be divided into two main classes:

*(A) Preventive antioxidants*: These reduce the rate at which new radical chains are initiated; this category includes (a) many of the endogenous enzymatic antioxidants, such as CAT or SOD, that directly target ROS/RNS, (b) the compounds able to sequestrate metal ions that mainly work as Fenton reaction inhibitors, (c) sulfur compounds such as GSH that are able to eliminate hydroperoxides.

*(B) Radical-Trapping Antioxidants (RTAs)*: RTAs, also known as *chain-breaking antioxidants*, are able to interrupt the chain of radical-mediated autoxidation of biomolecules, by capturing peroxyl radicals ROO•, thereby preventing them from abstracting H-atoms from RH or ROOH donors and terminating the chain. Several plant-derived vitamins and other phytochemicals have been tested as therapeutic agents in oxidation-related diseases. Phenolic RTAs have been studied the most [301], their model being 2*R*, 4′*R*, 8′*R*-α-tocopherol (α-T), the most biologically active of the four tocopherols that constitute vitamin E, also the lipid-soluble compound with the highest RTA activity in vitro [302]. It is worth noting that many plant-derived polyphenols are good one-electron reducing agents. However, they are usually tested by the spectrophotometric titration of alcoholic extracts of fruits, roots, leaves, or other portions of the source plants, based on the decolorization of one-electron oxidizing agents, such as DPPH (2,2-diphenyl-1-picryl-hydrazyl) and 2,2′-azino-bis(3-ethylbenzothiazoline-6-sulfonic acid) (ABTS). In this regard, Ingold and Pratt have warned that these titrations concern only the total reducing capacity of the extracts, and not the RTA activities, nor the H-atom-donating abilities of the extracted compounds. Indeed, in alcohol solvents and in water, phenols are partially ionized. The phenoxide anions ArO^−^ are oxidized much more rapidly than the corresponding phenols ArOH by the one-electron-deficient oxidizing compounds used in these titrations, in a reaction called *Sequential Proton-Loss Electron Transfer* (SPLET), which is not the direct transfer of an H-atom to a peroxyl radical ROO•. The latter is usually much slower and can only be measured under conditions that completely suppress SPLET, such as the addition of acetic acid to the alcohol solvent [302]. Non-phenolic RTAs also compose a large and important class of radical scavengers in living organisms. They react with peroxyl radicals by various mechanisms, such as (a) H-atom donation from weak O-H, N-H, or S-H bonds, as in the case of l-ascorbate, uric acid, and thiols (the main one being reduced glutathione, GSH), respectively; (b) addition to polyunsaturated compounds, as in the case of β-carotene and carotenoids in general, with formation of carbon free radicals poorly reactive towards O_2_; (c) co-oxidation processes with fast cross-termination reactions, as in the case of γ-terpinene; (d) catalytic quenching of the superoxide radical anion O_2_•^−^, with a SOD-type mechanism, as in the case of dialkyl nitroxides and FeCl_3_ [303].

Considering the exact mechanism of action of exogenous antioxidants is crucial to validate their usefulness in sarcopenia or any other disease. There is an ample range of mechanistic issues that need to be addressed by a variety of tests that can be used for this purpose. These include

The reaction mechanisms at the base of the electron/proton- or H-atom-donating abilities of antioxidants;Their dependence upon solvent characteristics (pH, polarity, etc.):Their water and/or lipid solubility;Their abilities to quench different kinds of radicals (O_2_•^−^, HO_2_•, •OH, •NO, •NO_2_);Their abilities to participate in the enzymatic decomposition of ROS/RNS and hydroperoxides;Their abilities to restore major antioxidants of biological significance in human metabolism (α-tocopherol, glutathion) to their reduced forms;Their prevalent intracytoplasmic or intranuclear activity.

Several bioactive compounds are continuously proposed as useful measures against oxidative stress in various pathologies. All the aspects mentioned above need to be considered to validate their preventive and/or therapeutic use in specific conditions, such as sarcopenia [299].

### 5.2. A Discussion of the Possible Usefulness of Major Antioxidants in Sarcopenia

The current management of sarcopenia consists of a combination of pharmacological and non-pharmacological approaches. Despite the great number of clinical trials conducted in recent years, truly effective pharmacological options for the treatment of this disease are lacking. Examining sarcopenia from a redox perspective as a basis for preventive and therapeutic interventions involves the evaluation of how antioxidants operate at the molecular level and which are their main characteristics in terms of pharmacokinetics (absorption, bioavailability, distribution, metabolism, and excretion) and pharmacodynamics (receptor binding, post-receptor effects, and chemical interactions). The numerous variables involved explain why many compounds in dietary supplements, despite their high antioxidant activity in vitro, may prove ineffective for the treatment of a specific disease or even harmful [249]. Oxidative stress is a fundamental component in the etiology and pathophysiology of sarcopenia; therefore, the use of antioxidant substances is desirable, at least to delay its long-term harmful effects. The use of natural antioxidants in sarcopenia needs to be critically evaluated, taking into account what we know of the redox alterations typically found in sarcopenia. A review of the main antioxidants could help identify those that may merit further evaluation in double-blind, randomized, placebo-controlled clinical trials in humans.

#### 5.2.1. Phenolic Radical-Trapping Antioxidants

*(A) Vitamin E*: Vitamin E is an essential nutrient comprising eight water-insoluble and fat-soluble tocopherols and tocotrienols, classified as α, β, γ, and δ based on the number and position of methyl groups, of which α-tocopherol (α-T) is the most biologically active. The basic structure of vitamin E, called tocol, is composed by a chromane ring with a hydroxyl group in position 6 (6-hydroxychromane), and a methyl group and a phytyl side chain of 16 carbon atoms, formed by the union of three isoprene units, in position 2. The isoprenoid chains of tocopherols are saturated, while those of tocotrienols have double bonds between carbons 3′-4′, 7′-8′, and 11′-12′ [304]. Wheat, wheat germ, nuts, cottonseed, and sunflower oils are important dietary sources of α-T. After entering the enterocytes by passive diffusion, vitamin E is incorporated into chylomicrons and is secreted at the basolateral pole of enterocytes into the lymph collected by the lymphatic capillaries of the intestinal villi, from where it reaches the bloodstream via the thoracic duct. Vitamin E in chylomicron remnants is taken up by the liver, incorporated into VLDL, and released into the circulation, where after conversion of VLDL to LDL, it enters cells through LDL receptors. Vitamin E is also transferred from HDL to VLDL and LDL in the bloodstream. Tocopherols and tocotrienols contained in vitamin E act as antioxidants due to the ease with which the -OH group at position 6 donates a hydrogen atom, protecting cells from lipid peroxidation. However, they also have shown to have pro-oxidant activity. α-T and virtually all phenols that are effective as RTAs act by capturing two (lipo)peroxyl radicals ROO• (LOO•) each. The first radical ROO• (LOO•) abstracts an H-atom from α-T, which becomes an α-tocopheroxyl radical α-TO• (Figure 8). The reaction between α-TOH and ROO• occurs either through concerted hydrogen transfer or via sequential electron transfer followed by proton transfer to form LOOH and α-TO•. The latter is not sufficiently reactive to abstract an H-atom, in turn, from a second RH (LH) or ROOH (LOOH), in which case the phenolic compound would be ineffective as RTA and would behave as a chain transfer reagent, continuing the radical autoxidation chain, although at a reduced rate. As with other phenols that are effective as RTAs, the radical α-TO• couples with a second radical ROO• (LOO•), thereby forming a peroxide adduct with the phenoxyl group. After reacting with free radicals, vitamin E is regenerated to its initial state by reduced coenzyme Q_10_ (CoQ_10_H_2_, ubiquinol-10), l-ascorbate, or dihydrolipoic acid (DHLA) (Figure 8). Due to continuous regeneration of α-TOH by l-ascorbate, ROO• (LOO•) scavenging by α-TOH allows suppression of lipid peroxidation at concentrations typically as low as one molecule of α-TOH per thousand phospholipids. The radical-trapping activity of α-T is most effectively coupled with the enzymatic decomposition of H_2_O_2_ and organic hydroperoxides to water and alcohol, respectively, by GPXs [302]. Despite its antioxidant potential, the use of vitamin E as a nutraceutical in sarcopenia is still limited, due to its poor bioavailability to skeletal muscle, where LDL receptor expression is low (Figure 9). However, dietary supplementation with vitamin E has been associated with an improvement in the oxidative stress index and a reduction in the risk of sarcopenia [305,306].

*(B) Coenzyme Q_10_ (CoQ_10_, ubiquinone-10)*: Coenzyme Q_10_ is a fat-soluble cofactor poorly absorbed from the small intestine. It has already been discussed in Section 4.2. After being incorporated into chylomicrons in enterocytes, it reaches the lymphatic system and then the bloodstream, as already explained for vitamin E. Circulating chylomicron remnants are rapidly taken up by the liver, where CoQ_10_ is repackaged into VLDL/LDL particles and secreted back into the bloodstream, where 95% is found in a fully reduced form (CoQ_10_H_2_, ubiquinol-10). Endogenous CoQ_10_ is produced primarily in tissues with high energy requirements or high metabolic activity, such as the heart, kidneys, liver, and muscles, and it appears that under normal circumstances the CoQ_10_ pool does not depend on exogenous CoQ_10_ intake. Furthermore, other tissues only absorb a small fraction of the dietary CoQ_10_ secreted in association with lipoproteins from the liver into the bloodstream. However, chronic intake of relatively high doses of CoQ_10_ has been shown to increase CoQ_10_ concentrations, especially in the mitochondria of the heart and brain in rodent models [307]. CoQ_10_ may serve as a fat-soluble counterpart to the water-soluble l-ascorbate. As mentioned in Section 4.2, CoQ_10_ plays a central role in the transfer of electrons between complexes I, II, and III of the ETC, and in the scavenging of lipoperoxyl radicals LOO• formed from lipids in cell membranes, as well as in the regeneration of α-tocopherol (α-T) (Figure 8). Despite substantial evidence of protection afforded by coenzyme Q_10_ to muscle cells in vitro and in rodents, animal studies with CoQ_10_ specifically regarding sarcopenia are scarce and controlled clinical trials in sarcopenic humans are lacking.

*(C) Resveratrol*: Resveratrol (3,5,4′-trihydroxy-trans-stilbene) (Figure 8) is a polyphenolic stilbenol extracted from grape skin. It is a compound with low bioavailability, due to its poor solubility in water, limited intestinal absorption, and intense hepatic metabolism. Resveratrol exerts both direct and indirect antioxidant actions. Its chemical structure allows it to function as a radical scavenger and chain-breaking antioxidant by donating electrons to free radicals, a role it best plays in synergy with other antioxidants such as l-ascorbate or GSH [308]. However, some of resveratrol’s indirect antioxidant actions are better characterized. Resveratrol and its catabolites inhibit the generation of O_2_•^−^ by XO, by binding specifically to the FAD site in XO [309]. Encouraging evidence suggests a possible protective role in sarcopenia in rodents. The development of tubular aggregates in type IIb muscle fibers was inversely correlated with capillarization in the tibialis anterior muscle of aged C57BL/6J mice. Long-term treatment with resveratrol induced improved capillarization in aged mice and a reduction of tubular aggregates, highlighting its ability to protect skeletal muscle tissue from age-related alterations [310]. Short-term (5-week) treatment of middle-aged male mice with 20 mg/kg body weight of resveratrol daily was associated with decreased Cox-2 activity in treated muscles, suggesting that the anti-inflammatory action of resveratrol may prevent skeletal muscle fiber changes [311]. Resveratrol prevented sarcopenic obesity by reversing mitochondrial dysfunction and oxidative stress through the PKA/LKB1/AMPK pathway in rats fed a high-fat diet [312]. The sirtuin system appears to be critical for the beneficial effects of resveratrol [313]. Significant dose- and time-dependent effects of dietary resveratrol on weight loss, similar to those of exercise training, were observed in aged rats, including inhibition of protein degradation and protection against oxidative stress. Resveratrol activates the signaling pathways related to IGF-1, AKT, AMPK, and SIRT1 in myoblasts and induces the expression of genes involved in neural and neuromuscular synapse function [314]. Sirtuin 1 plays an essential role in preventing muscle atrophy after cerebral ischemic stroke. Loss of sirtuin 1 function in post-stroke muscle was responsible for the activation of the ubiquitin–proteasomal system through increased expression of atrogin-1, muscle RING finger protein 1 (MuRF1), and zinc finger protein 216 (ZNF216) [315]. Treatment of rodents with resveratrol inhibited the ageing-related decline in rotarod riding time, preserved muscle mass and attenuated cardiomyocyte hypertrophy, concomitant with the restoration of autophagy in skeletal muscle and myocardium. Conversely, SIRT1 deletion in skeletal muscle impaired autophagic flux and caused muscle fiber atrophy [316]. Limited controlled trials have been conducted with resveratrol in sarcopenic patients: supplementation with resveratrol combined with exercise in older adults was associated with improved mitochondrial function, hypertrophy of type I and IIA muscle fibers and increased numbers of myonuclei [317].

*(D) Curcumin*: Curcumin is a phenolic diarylheptanoid derived from turmeric (*Curcuma longa*) (Figure 8). Numerous studies have reported pleiotropic antioxidant effects in multiple pathologies but are still controversial. The ability to scavenge free radicals is the most important chemical property of curcumin and the basis of its medicinal functionality [318]. Encouraging results have recently emerged with curcumin in relation to muscle damage and sarcopenia in rodents. In a murine model of acute skeletal muscle injury, curcumin demonstrated protective effects, increasing the number of SMSCs and in the diameter of myotubes in injured muscles, and reducing muscle inflammation and fibrosis, through the expression of Wnt5a [319]. Furthermore, curcumin administration significantly reduced mortality, prevented muscle mass loss, and attenuated soleus muscle strength loss in aged animals. It also prevented the development of soleus sarcopenia in aged 10ScSn mice, by preserving type I myofiber size and inducing of type IIa fiber hypertrophy [320]. Furthermore, curcumin showed anti-fatigue, anti-apoptosis, anti-inflammatory, and antioxidant effects on skeletal muscle during chronic forced exercise in aged mice, through the expression of genes related to muscle protein synthesis [321]. It should be noted that the potential utility of curcumin as a therapeutic agent is limited by its chemical instability, water-insolubility, pleiotropic effects and poor bioavailability after oral administration, with negligible drug distribution to the liver and other tissues beyond the gastrointestinal tract, and rapid and extensive metabolism [322]. Physical adsorption of curcumin onto a hydroxyapatite surface (Cur-SHAP) enabled sustained release after intramuscular administration to rats. The particle size of Cur-SHAP (500–1500 nm), was suitable for endocytic cellular uptake. After prolonged treatment, rats receiving Cur-SHAP effectively recovered from LPS-induced sarcopenia [323]. In humans, the role of curcumin in musculoskeletal health has been the subject of limited controlled investigation. A randomized, placebo-controlled, double-blind clinical trial was conducted in 30 subjects using an oral formulation with improved bioavailability, based on a complete natural turmeric matrix (CNTM) that included turmeric essential oil and other sugar and protein components. Supplementation produced marginal increases in grip strength, weight-lifting capacity, and distance walked before fatigue compared to controls [324].

#### 5.2.2. Non-Phenolic Radical-Trapping Antioxidants

*(A) L-Ascorbate (vitamin C)*: Vitamin C is the L-ascorbate stereoisomer of ascorbic acid and is an essential nutrient and the most effective water-soluble antioxidant in plasma. It has the structure of a six-carbon α-ketolactone. Vitamin C exists in vivo as both the fully reduced L-ascorbate anion (Asc), which is in equilibrium with ascorbic acid, and oxidized dehydroascorbic acid (DHA), with the ascorbyl radical anion Asc•^−^ as an intermediate between the two. Fully reduced Asc is the predominant form at physiological pH. Sources of vitamin C include citrus and other fruits, berries, and vegetables. Vitamin C absorption occurs primarily in the distal ileum. L-ascorbate is efficiently transported across the apical membrane of enterocytes via the solute carrier family 23 member 1 (SLC23A1, or SCTV1) transporter. Furthermore, several glucose transporters facilitate the absorption of DHA, while the low pH of the distal ileum facilitates its passive diffusion. The release of vitamin C into the bloodstream is vital for the absorption process, and the rapid appearance of vitamin C in the blood strongly implicates the existence of as-yet-undiscovered channels and/or transporters. Biodistribution to cells from the bloodstream is mediated by both facilitated diffusion of DHA and the low-capacity/high-affinity solute carrier family 23 member 2 (SLC23A2, or SVCT2) transporter, which is widely expressed in all organs, including muscle [325] (Figure 9). Ascorbate can regulate oxidative homeostasis as an antioxidant or as a pro-oxidant. The L-ascorbate anion (Asc) is an excellent reducing agent. When it participates in enzymatic reactions, it usually loses two electrons, resulting in the formation of DHA. At physiological pH, Asc reacts also with free radicals. The reaction of Asc with a radical species by one-electron oxidation produces the ascorbyl radical anion Asc•^−^. Because the unpaired electron of Asc•^−^ is in a highly delocalized π-system, it is a relatively unreactive free radical, making it a superior biological antioxidant. It is thermodynamically at the bottom of the hierarchy of oxidizing free radicals, so all radicals with higher reduction potentials will accept electrons from it [326]. This is why L-ascorbate reacts with (lipo)peroxyl radicals ROO• (LOO•) at lipid/water interfaces, with water-soluble radicals, such as O_2_•^−^, HO_2_•, •OH, •NO, GS•, and the urate radical •UH^−^, and with non-radical species, such as the peroxynitrite anion HOOH^−^ and HClO. More importantly, L-ascorbate helps extend the utility of α-tocopherol (α-T), by functioning as a sacrificial reducing agent, whereby α-tocopheroxyl radicals α-TO• are reduced back to α-T at the lipid/water interface of LDL [327] (Figure 10). Dimerization between two radical anions Asc•^−^, followed by reversible disproportionation, can lead to the formation of stable Asc and DHA. Interaction of Asc•^−^ with another radical species can also lead to the formation of DHA. Both Asc•^−^ and DHA can be reduced back to Asc either by ascorbic acid reductase (AAR), using NAD(P)H as an electron donor, or via non-enzymatic reaction with GSH [328]. Extensive experimentation in vitro and in rodents has convincingly demonstrated the protective effects of ascorbate on muscle cells. However, specific studies on sarcopenia are scarce. As with vitamin E, vitamin C supplementation has been associated with an improvement in oxidative stress index and a reduction in the risk of sarcopenia [306,329,330].

*(B) Lipoic Acid (LA)*: Lipoic acid (LA, (*R*)-5 (1,2-dithiolan-3-yl)pentanoic acid) is an amphiphilic organosulfur compound that exists in nature exclusively as a biologically active (*R*)-enantiomer, present in both oxidized (LA) and reduced forms (dihydrolipoic acid, DHLA) (Figure 10). LA is a cofactor in various enzyme systems involved in energy and amino acid metabolism, such as the pyruvate dehydrogenase complex (PDC) and the branched-chain α-ketoacid dehydrogenase complex (BCKDC), where it is covalently linked via an amide bond to a terminal lysine residue of the lipoyl domain of the enzyme. It is produced endogenously from a precursor, octanoic (caprylic) acid, which binds to the enzyme before the enzymatic insertion of the two sulfur atoms in its 1,2-thiolane ring, resulting in the production of only non-free LA. LA is present in very low doses in many foods, where it is bound to lysine in proteins, such as liver, spinach, broccoli, and yeast extract, and is absorbed in the small intestine in a proton-dependent manner via the monocarboxylic acid transporter (MCT, or SLC16A). Certain compounds, such as medium- and short-chain fatty acids, and food intake, in general, negatively influence absorption, while a low pH on an empty stomach promotes absorption by the gastric mucosa. After absorption, LA increases in the bloodstream in complex with albumin or as free LA and enters primarily into hepatic, cardiac, and skeletal muscle cells via MCT. In cells, the reduction of LA to DHLA is carried out by NAD(P)H enzymes, such as cytosolic GSR and TRX1, and mitochondrial TRX2 and lipoamide dehydrogenase. In erythrocytes, DHLA is released into the extracellular space. The short half-life and low biodistribution, due to reduced solubility and instability in the stomach, hepatic degradation and/or conjugation, and biliary and urinary excretion, reduce the usefulness of LA as a natural therapeutic compound. However, this has been partly improved by the use of amphiphilic matrices and liquid formulations that increase its intestinal absorption and bioavailability [331]. Lipoic acid has a high antioxidant capacity, both direct and indirect. In its reduced form (DHLA), it is able to provide hydrogen atoms that regenerate the antioxidant activity of the α-tocopheroxyl radical α-TO•, dehydroascorbate and oxidized CoQ10, and recover GSH from GSSG, having a standard reduction potential even lower of that of GSH (−0.29 V vs. −0.24V). In its oxidized form, which has a tense conformation, it is able to remove HClO. Its indirect antioxidant actions include the ability to chelate iron and copper ions and to promote the activation of NRF2 and the expression of HO-1 [332]. However, it has been suggested that the modulation of the expression of antioxidant enzymes may reflect the promotion by LA of an oxidative stress response [333]. Furthermore, LA has been shown to inhibit NF-κB activation and promote glucose uptake in insulin-resistant skeletal muscle cells through tyrosine phosphorylation of insulin receptor substrate-1 (IRS-1) and redistribution of glucose transporters to the plasma membrane [334]. Studies regarding lipoic acid administration to sarcopenic animals or humans are very scarce. Recently, lipoic-acid-modified nanoparticles in mice have been shown to promote M2 polarization of macrophages and their interaction with SMSCs, through which macrophages inhibit apoptosis and enhance SMSC differentiation [335].

*(C) Vitamin A and retinoids*: Vitamin A is the collective name for several substances whose structure is related to all-*trans*-retinol, a water-insoluble and lipid-soluble vitamin. They form a family of compounds that includes (a) animal-derived retinoids, such as retinol and retinyl esters; (b) various plant-derived carotenoids, such as α-carotene, β-carotene, γ-carotene, and the xantophyll β-cryptoxanthin, which serve as precursors to vitamin A in herbivorous and omnivorous animals that possess the enzymes necessary to cleave and convert provitamin A carotenoids to retinol in the liver. The structure of carotenoids consists of a six-carbon β-iononic ring with a side chain composed of two isoprene units and a primary alcohol. The trans isomers in all double bonds (all-*trans*) of the chain have the highest activity. β-Carotene is a polyunsaturated hydrocarbon present in the hydrophobic domains of LDL, cells, and tissues. It is part of the large family of carotenoids, which includes over 1100 carotenes and xanthophylls. These are pigmented 40-carbon tetraterpenes produced in plants, algae, fungi, and bacteria. β-carotene is synthesized from geranylgeranyl pyrophosphate and has eight isoprene units and a cyclohexene ring at both ends. β-carotene quenches the nitrogen dioxide radical •NO_2_, the peroxyl radical ROO•, but not the α-tocopheroxyl radical α-TO•. At low O_2_ pressure, the radical β-carotene-ROO• can quench a second radical ROO•. Indeed, the antioxidant capacity of β-carotene is strongly dependent on the partial pressure of O_2_, possibly being overcome by the pro-oxidant toxic effects at high partial pressure of O_2_ in the lungs. Vitamin A is obtained mainly orally through the diet. Natural sources of vitamin A are milk, cheese, butter, eggs, fish, and kidney meat [304]. Vitamin A is rapidly absorbed by passive diffusion from the gastrointestinal tract where, after incorporation into chylomicrons, retinyl esters (REs) and β-carotene are secreted into the lymphatic system, reach the blood, and are subsequently transported to the liver, which serves as the main retinoid target and storage organ. Hepatocytes release vitamin A into the bloodstream in association with retinol-binding protein (RBP), less so than with albumin and lipoproteins. The RBP receptor and stimulated by retinoic acid receptor 6 (STRA6) allow its transport to target tissues. In cells, retinol (ROH) and retinyl esters are further oxidized to all-*trans*-retinoic acid (ATRA) (Figure 10), which is responsible for vitamin A functions in the body. Vitamin A appears to reach skeletal muscle with poor efficiency, but further studies are needed to clarify its pharmacokinetics [336]. According to studies in vitro and in rodents, retinoids and carotenoids confer protection to muscle cells [337]. However, specific animal studies for sarcopenia are lacking. Controlled clinical trials in human sarcopenia are also limited. A positive correlation exists between vitamin A intake and muscle mass among American adult males [338]. Furthermore, eating more foods rich in antioxidants, such as vitamins A, E, and selenium, at lunch has been associated with a reduced risk of sarcopenia and improved handgrip strength [305,306].

*(D) Ergothioneine (ET)*: Ergothioneine (2-mercaptohistidine, trimethyl-betaine, ET) is a naturally occurring, water-soluble sulfur amino acid derived from l-histidine found in ergot (*Claviceps purpurea*), with a trimethylated amino group and a mercapto group at position 2 of the imidazole ring (Figure 10). ET exists tautomerically in a thiol and a thione form, with the latter, more stable, being favored at physiological pH. ET is efficiently absorbed in the gut thanks to a specific transporter, the organic cation transporter novel 1, or solute carrier family 22, member 4 (OCTN1/SLC22A4). [339,340]. After intestinal absorption, ET reaches the liver through the portal circulation and, beyond it, other organs and tissues, where it is absorbed through the same transporter. The preferential sites of accumulation are erythrocytes, liver, kidneys, spleen, and brain, the latter poorly reached due to its limited ability to cross the blood–brain barrier. ET possesses anti-inflammatory properties and significant antioxidant activity, which it exerts in various ways, both through the scavenging of ROS/RNS (•OH, HClO, HNOO^−^, and singlet oxygen ^1^O_2_) and through the chelation of divalent metal ions (Fe^2+^, Cu^2+^), but also through the induction of NRF2 [340]. Several studies indicate ET as one of the most promising natural antioxidants in a wide range of pathological conditions. It also exhibits an anti-ageing capacity that promotes longevity. Indeed, low plasma levels of ET are associated with increased risk of frailty, cardiovascular disease, cognitive impairment, dementia, and Parkinson’s disease. However, no correlation with sarcopenia has been reported [341,342]. To date, there are no published data demonstrating any impact of ET in animal models of sarcopenia or in sarcopenic patients, although its use in these settings may be conceivable, given the measurable, albeit low, expression of the OCTN1 transporter in muscle cells (Figure 9).

#### 5.2.3. Antioxidants That Act Through Different Mechanisms

*Vitamin D*: Vitamin D is a conditionally essential nutrient, fat-soluble and resistant to heat and oxidation. There are two forms of vitamin D: vitamin D2, or ergocalciferol, of plant origin, and vitamin D3, or cholecalciferol. The best dietary source of vitamin D is fatty fish, with smaller amounts provided by beef liver, eggs, and cheese. With adequate exposure of the skin to the ultraviolet B (UV-B) radiation component of sunlight, cholecalciferol is synthesized from 7-dehydrocholesterol in the lower layer of the epidermis. Vitamin D is absorbed from enterocytes in the small intestine by simple diffusion and, after incorporation into chylomicrons, is secreted at the basolateral pole of enterocytes, from where it reaches the lymphatic system and ultimately the bloodstream. Human in vitro models of the intestinal epithelial barrier suggest active bidirectional transport of vitamin D by cholesterol transporters in enterocytes. Maturation of vitamin D to its active form involves two hydroxylation steps. The first hydroxylation is catalyzed by cholecalciferol-25-hydroxylase in the liver and other tissues to generate 25-hydroxycholecalciferol (25-OH-D3, calcidiol), which is coupled to vitamin-D-binding protein (DBP) (85–88%) and albumin (12–15%). The second hydroxylation occurs in the kidney, where the DBP complex is filtered through the glomeruli and 25-OH-D3 is transported into the mitochondria of tubular cells, where 25-OH-D3-1α-hydroxylase produces dihydroxycholecalciferol [1,25-(OH)_2_-D3, calcitriol]. Vitamin D exerts its actions by binding to the vitamin D receptor (VDR) in the cytoplasm. After binding, VDR heterodimers with the transcription factor RXR induce the expression of vitamin-D-regulated genes by binding to vitamin D response elements (VDRE) in DNA. Vitamin D’s actions on skeletal muscle include its ability to influence muscle contraction by regulating calcium and phosphate transport [343] and glucose uptake [344]. Furthermore, calcitriol also displays the ability to participate in the regulation of myoblast proliferation and differentiation, as well as protein synthesis and mitochondrial metabolism of muscle cells [345]. Several clinical studies on the use of vitamin D have shown little or no effect in improving muscle function in patients with sarcopenia. Furthermore, the observed poor effects could not be attributed with certainty to vitamin D supplementation, since it was administered together with other substances, such as hydroxybutyrate and essential amino acids, or the supplementation was associated with physical exercise [345,346,347,348].

At the end of this review on the main antioxidants, we could summarize their properties as follows: (1) l-ascorbate exhibits excellent (lipo)peroxyl and α-tocopheroxyl radical-trapping activities and a hydrogen atom donation capacity second only to GSH, along with good intestinal absorption and high bioavailability to muscle; (2) coenzyme Q_10_ exhibits almost equally excellent scavenging and hydrogen atom donation activities as l-ascorbate, but poor intestinal absorption; (3) dihydrolipoic acid is endowed with even greater radical-scavenging and hydrogen atom donation capacity towards DHA, oxidized CoQ_10_, α-TO•, and GSSG, having a lower standard reduction potential than all of these, but suffers from poor solubility and biodistribution and a short half-life in the blood; (4) α-tocopherol (vitamin E) exhibits very good scavenging ability towards lipoperoxyl radicals, but no ability to regenerate the aforementioned antioxidants, along with good intestinal absorption but poor bioavailability; (5) ergothioneine and retinol have more limited radical-scavenging activities, good absorption, but moderate and low bioavailability to muscle, respectively. Vitamin D may be useful as an indirect antioxidant, while the rational use of resveratrol and curcumin as antioxidants still requires careful evaluation.

## 6. Concluding Remarks

From the present, extensive review of the literature concerning the contribution of oxidative stress to the loss of muscle strength, muscle quantity/quality, and physical performance that characterizes sarcopenia, the ambivalent role of redox imbalance clearly emerges. On the one hand, the activation of moderate oxidative stress responses appears to play a beneficial role in muscle adaptation to increased workload and in muscle regeneration. On the other hand, it is equally clear that greater intensity of oxidative stress, such as occurs with excessively intense and prolonged exercise, caloric overload, inflammation, and the harmful combination of the two that occurs in overweight individuals, trigger and promote the progression of sarcopenia, especially when supported by the decline in antioxidant defenses that accompanies ageing in individuals who lack a healthy lifestyle based on nutrition and physical activity. A key causal role in the cellular and organ damage associated with chronic oxidative stress is played not only by primary determinants, such as ROS and RNS, but also by their “second messengers”, i.e., the advanced products of lipid peroxidation and glycoxidation that are generated endogenously but can also be introduced from outside, as indicated above.

Among the main antioxidants examined here as potential resources against sarcopenia, a careful comparison of their biochemical, pharmacokinetic, and pharmacodynamic characteristics reveals that some stand out as having superior qualities over others, particularly l-ascorbate, coenzyme Q_10_, dihydrolipoic acid (with some limitations), and α-tocopherol. Despite substantial evidence regarding the protective effects of some of these antioxidants on muscle cells in vitro and in rodents, animal studies specific to sarcopenia are scarce. Furthermore, there are a limited number of controlled clinical trials demonstrating the ability of these antioxidants to exert beneficial effects on sarcopenia in humans. With these exceptions, the few available trials should be the subject of a systematic meta-analysis; however, this is beyond the scope of this review. Many of the available studies have been conducted with combinations of bioactive compounds, sometimes in association with physical exercise. This limits the ability to attribute the observed effects with certainty to one compound or another, but it also reflects a reality well-known in medicine and clinical nutrition: that there is likely no single “ideal antioxidant” that allows for muscle strength, muscle quantity/quality, and physical performance, outside of an integrated approach based on the administration of assorted antioxidant compounds with different activities (for example, one capable of effectively scavenging free radicals, such as α-tocopherol, and another capable of regenerating the former, enhancing and prolonging its effect, such as l-ascorbate), and on the mutually beneficial combination of antioxidants and exercise. Furthermore, antioxidant supplementation cannot replace a healthy lifestyle, of which healthy ageing is only the latest noteworthy aspect. The above allows us to better understand how sarcopenia, like other diseases that arise with ageing but are not necessarily related to it, can be prevented or contained by a lifestyle that includes moderate and regular physical exercise; prevention and treatment of inflammatory diseases of any kind; a diet rich in vitamins and natural antioxidants and with a total caloric intake appropriate to the individual’s body mass and activity; avoidance of sources of lipid peroxidation and glycoxidation products, such as cigarette smoke and the consumption of foods subjected to manufacturing, storage, and cooking processes that maximize the generation of ALEs and AGEs; and the possible use of antioxidant supplements appropriately selected based on their biochemical properties and bioavailability. Finally, we believe it is appropriate to encourage more controlled clinical studies on the topic, especially with emerging compounds and with new formulations that maximize the bioavailability of already known antioxidants.

## Figures and Tables

**Figure 2 ijms-26-07787-f002:**
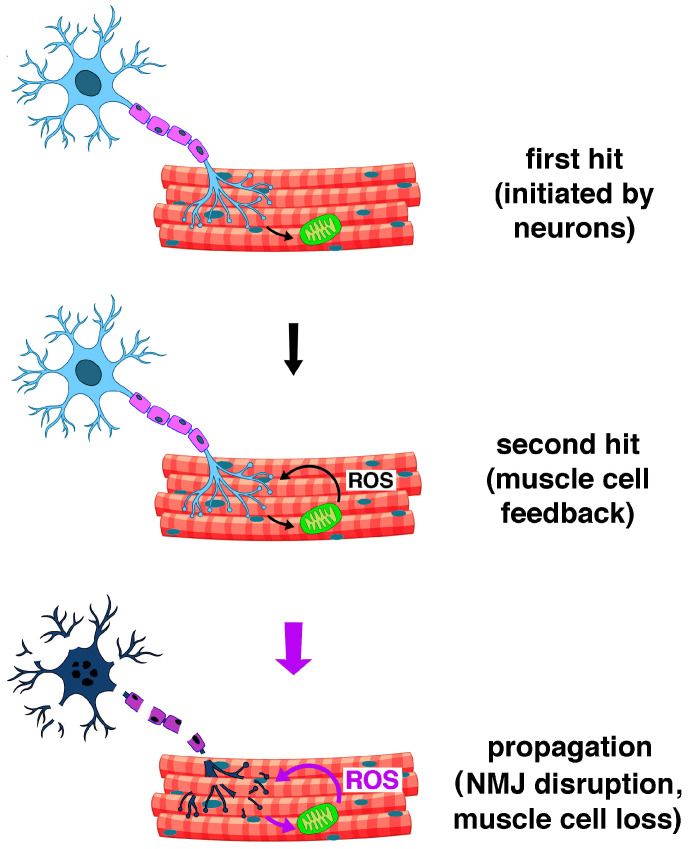
The proposed mechanism of loss of muscle mass and function in *Sod1* knockout mice, according to the two-hit model (see text). Oxidative stress in neurons initiates NMJ disruption by promoting mitochondrial dysfunction and increased ROS production in muscle cells. Retrograde feedback from muscle cells augments NMJ dysfunction by inciting further ROS production in neurons, which leads to a vicious cycle that ultimately results in NMJ disaggregation, denervation, and loss of muscle fibers. Increased ROS production is symbolized by thicker purple arrows and larger purple font in the text label (redrawn from Deepa et al., 2019 [125] using graphic elements drawn by brgfx/Freepik (http://www.freepik.com)).

**Figure 3 ijms-26-07787-f003:**
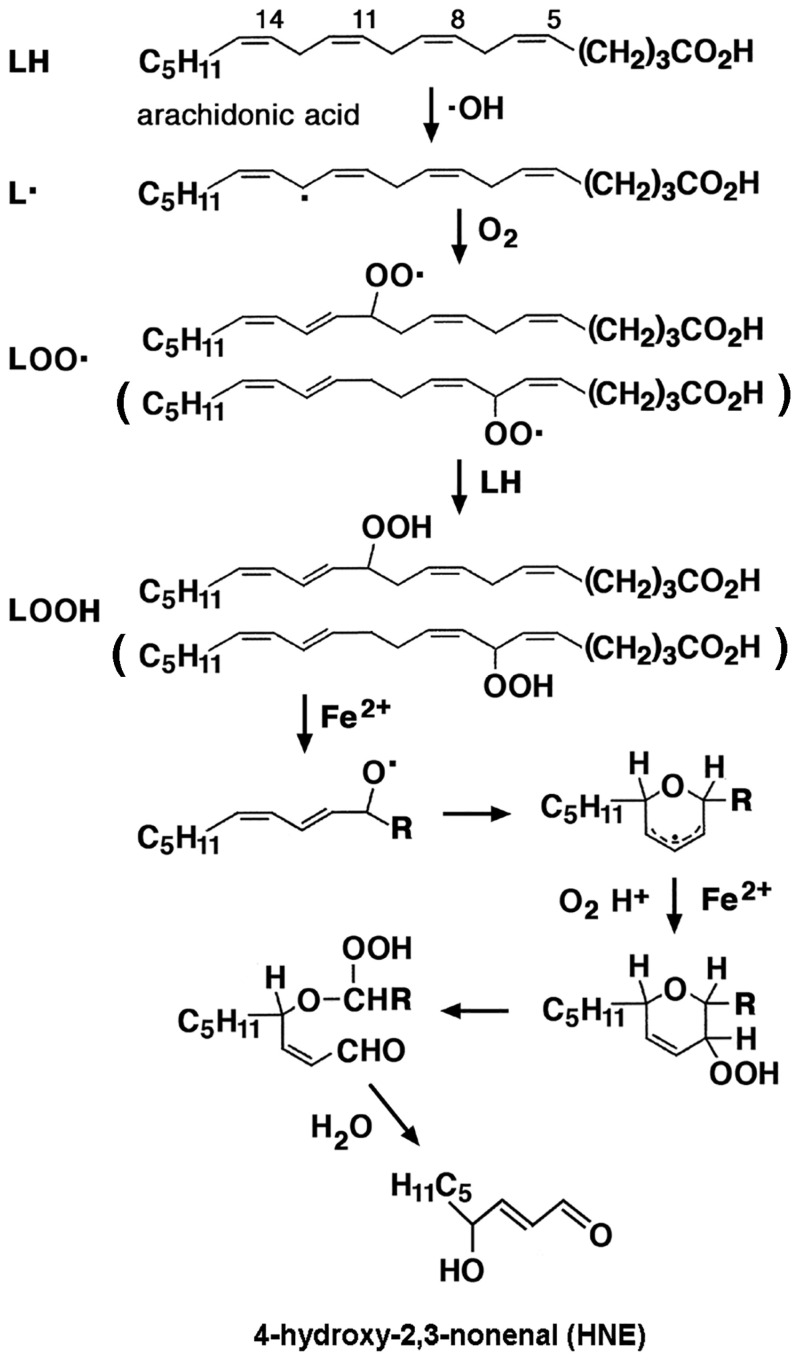
Mechanism of formation of the reactive aldehyde 4-hydroxy-2,3-nonenal (HNE) by non-enzymatic lipid peroxidation (LPO) of 5,8,11,14-eicosatetraenoic acid (20:4[n-6], arachidonic acid, AA). In the presence of the hydroxyl radical •OH, AA is oxidized to its lipid radical L•, centered on carbon atom 13. The lipid radical L• reacts with O_2_, producing an 11-lipoperoxyl radical LOO• (and its positional isomer, the 15-lipoperoxyl radical, in brackets), from which 11-hydroperoxy-arachidonic acid (and 15-hydroperoxy-arachidonic acid, in brackets) are obtained by reaction with other LH lipids. The decomposition of 11-hydroperoxy-AA produces HNE, along with other compounds. HNE can be produced similarly from esterified AA contained in phospholipids, such as 1-palmitoyl-2-arachidonoyl-sn-glycero-3-phosphocholine, or from free or esterified 9,12-octadecadienoic acid (18:2[n-6], linoleic acid, LA). The simplified figure does not show the formation of alkoxyl radicals LO• from 11-hydroperoxy-AA, which contribute to the propagation of LPO.

**Figure 4 ijms-26-07787-f004:**
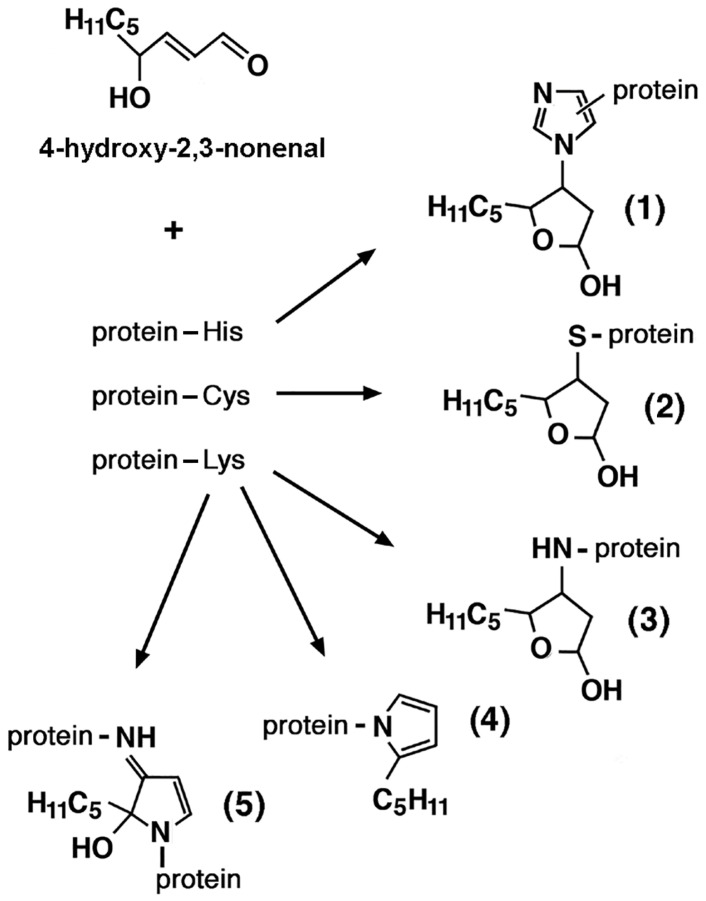
The mechanism of production of advanced lipid peroxidation end products (ALEs) by the formation of covalent adducts of HNE with proteins. Hemiacetalic adducts formed by the Michael reaction of HNE with (1) histidine, (2) cysteine, (3) lysine residues; (4) protein modified with 2-pentyl-pyrrole by the addition of HNE to the lysyl amino group, forming a Schiff base; (5) the fluorescent adduct of HNE with two lysine residues, resulting in cross-linking.

**Figure 5 ijms-26-07787-f005:**
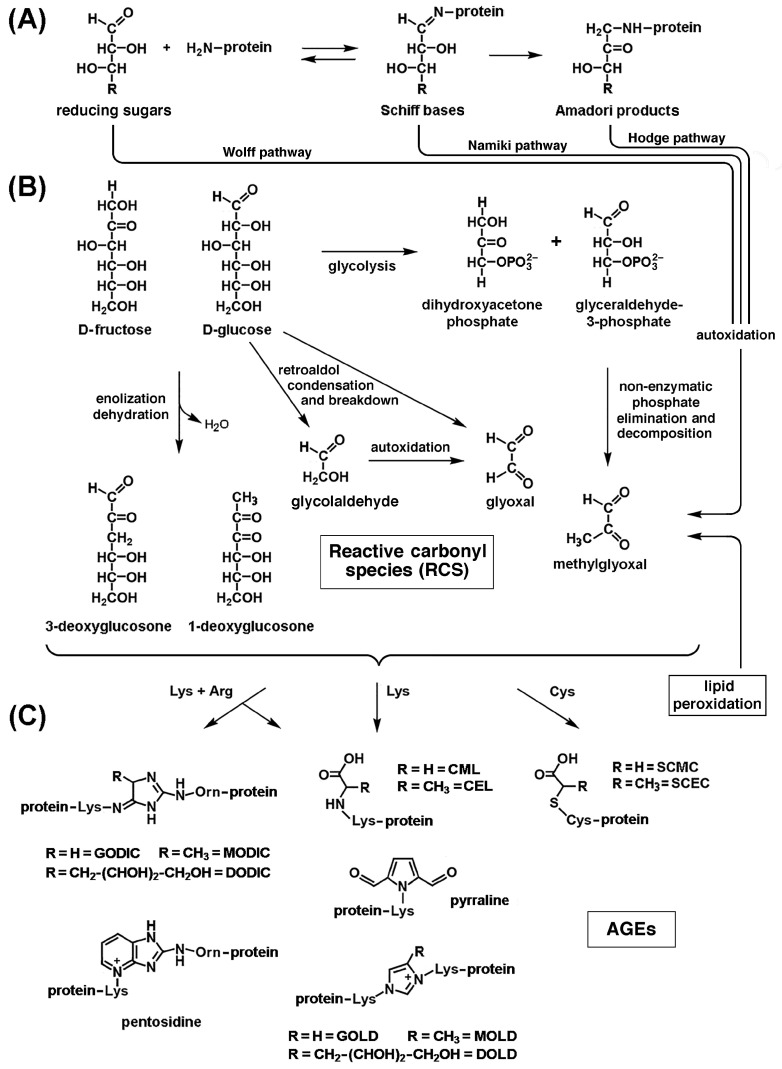
Mechanisms of formation of advanced glycation/glycoxidation end products (AGEs): (**A**) A Maillard reaction between a reducing sugar and the lysyl amino group in a protein, proceeding through the formation of a Schiff base, followed by the Amadori rearrangement, i.e., the isomerization of the aldose to the corresponding 1-amino-1-deoxy-ketose. (**B**) The most common reducing sugars are illustrated, along with some of their degradation and oxidation products that serve as intermediates in the formation of advanced glycation/glycoxidation end products (AGEs). (**C**) Typical AGEs formed by the addition of reactive intermediates to lysine, arginine, and cysteine residues in peptide/proteins. Legend: CMC: *N^ε^*-carboxymethylcysteine; CEC: *N*^ε^-carboxyethylcysteine; CML: *N*^ε^-carboxymethyllysine; CEL: *N^ε^*-carboxyethyllysine; GOLD, MOLD, DOLD: imidazolium cross-links between two lysines and GO, MGO, and 3-deoxyglucosone (3-DG), respectively. GODIC, MODIC, and DODIC: imidazolium cross-links between lysine, arginine, and GO, MGO, and 3-DG, respectively. Note the conversion of arginine to ornithine.

**Figure 6 ijms-26-07787-f006:**
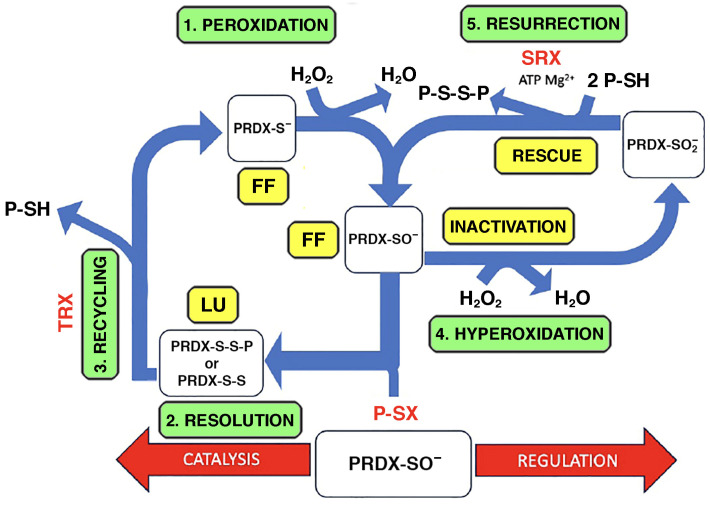
The peroxiredoxin (PRDX) cycle: (1) H_2_O_2_ first peroxidizes the thiolate group (PRDX-S^−^) in the fully folded (FF) active site of PRDX to yield a sulfenic acid (PRDX-SOH)/sulfenate (PRDX-SO^−^); (2) following a conformational shift by which the PRDX active site becomes locally unfolded (LU), the -SOH/-SO^−^ group forms an intramolecular disulfide bond with a second “resolving” Cys residue or a mixed disulfide with the thiol group of a protein-bound Cys residue (P-SH); (3) the thiol functional group in the FF active site is subsequently recycled by catalytic reduction by thioredoxin (TRX) or a TRX-like protein; (4) inactivation of PRDX by hyperoxidation occurs when the PRDX-sulfenate (R-SO^−^) undergoes further oxidation to the inactive sulfinate (R-SO_2_^−^) at high peroxide concentrations; (5) the PRDX-SO^−^ sulfenate group is “resurrected” by ATP-dependent reduction of the sulfinate by sulfiredoxin (SRX), using two thiol-reducing equivalents.

**Figure 7 ijms-26-07787-f007:**
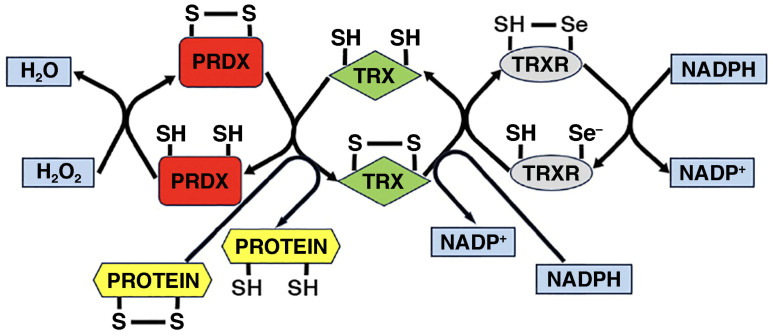
The thioredoxin (TRX) cycle: TRXs restore the antioxidant capacity of oxidized enzymatic and non-enzymatic antioxidant proteins in the cytosol and mitochondrial matrix by regenerating their reduced forms. For example, TRXs catalyze the reductive recycling of oxidized peroxiredoxins (PRDXs). In turn, the reductive regeneration of TRXs is catalyzed by thioredoxin reductases (TRXRs), using NADPH as a two-electron donor.

**Figure 9 ijms-26-07787-f009:**
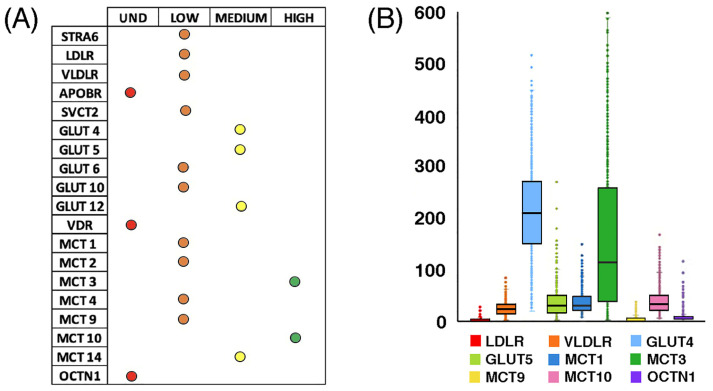
Expression of antioxidant receptors in human skeletal muscle: (**A**) Protein expression levels measured by immunostaining of tissue microarrays using CODEX (CO-Detection by indEXing) technology. Knowledge-based annotation data from the Human Protein Atlas (HPA) (https://www.proteinatlas.org/). Legend: UND, undetectable; STRA6, stimulated by retinoic acid 6; LDLR, LDL receptor; VLDLR, VLDL receptor; APOBR, apo B receptor; SVCT2, solute carrier family 23 member 2; GLUTn, glucose transporter 4, 5, 6, 10, 12; VDR, vitamin D receptor; MCTn, mono-carboxylic acid transporter 1, 2, 3, 4, 9, 10, 14; OCTN1, organic cation transporter novel type 1. (**B**) mRNA expression levels measured by RNA sequencing. Data from the Genotype-Tissue-Expression (GTEx) project for human tissues of the Human Protein Atlas reported as means of normalized transcripts per million (nTPM). Legend as for panel (**A**). Values for APOBR, SVCT2, GLUT6, GLUT10, GLUT12, VDR, MTC2, MTC4, MTC14, and STRA6 were very low or undetectable.

**Figure 10 ijms-26-07787-f010:**
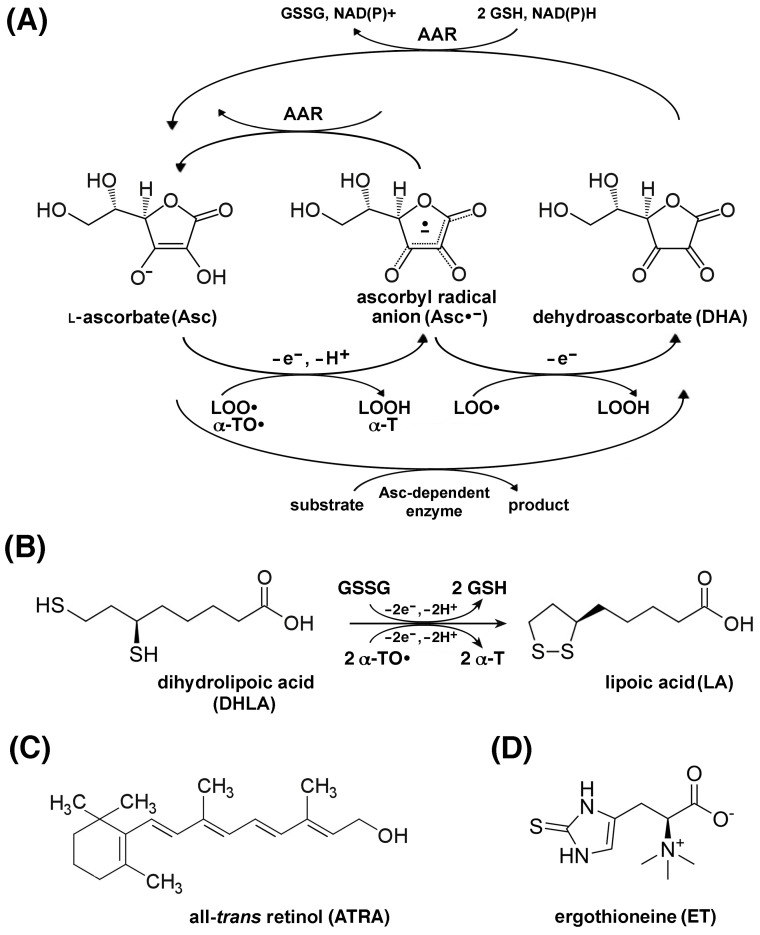
Structure and mechanisms of action of the main non-phenolic radical-trapping antioxidants: (**A**) l-ascorbate (Asc, vitamin C), ascorbyl radical anion Asc•^−^, and dehydroascorbate (DHA); (**B**) dihydrolipoic acid (DHLA) and lipoic acid (LA). The scavenging of lipoperoxyl radicals LOO• by l-ascorbate and by the ascorbyl radical anion Asc•^−^ is depicted. The reductive regeneration of α-tocopherol by l-ascorbate and DHLA, and of reduced glutathione (GSH) by DHLA, is also illustrated. (**C**) Vitamin A (all-*trans* retinol); (**D**) ergothioneine. Legend: AAR, ascorbic acid reductase; α-T, α-tocopherol (vitamin E); α-TO•, α-tocopheroxyl radical; GSSG, oxidized glutathione; LOOH, lipid hydroperoxide.

**Table 1 ijms-26-07787-t001:** EWGSOP2 tests and cut-off points for sarcopenia.

Test	Cut-Off Points (Men)	Cut-Off Points (Women)
EWGSOP2 sarcopenia cut-off points for low strength by chair and grip strength		
Grip strength	<27 kg	<16 kg
Chair strength	>15 s for 5 rises	
EWGSOP2 sarcopenia cut-off points for low muscle quantity		
ASM	<20 kg	<15 kg
ASM/height^2^	<7.0 kg/m^2^	<5.5 kg/m^2^
EWGSOP2 sarcopenia cut-off points for low performance		
Gait speed	≤0.8 m/s	
SPPB	≤0.8 point score	
TUG	≥20 s	
400 m walk test	Non-completion or ≥6 min for completion	

ASM, Appendicular Skeletal Muscle Mass; EWGSOP2, European Working Group on Sarcopenia in Older People 2; SPPB, Short Physical Performance Battery; TUG, Timed Up and Go test.

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
