# Peer review of "A Reassessment of Sarcopenia from a Redox Perspective as a Basis for Preventive and Therapeutic Interventions"

_ijms, 2025, doi:10.3390/ijms26167787_

Round 1
Reviewer 1 Report
Comments and Suggestions for Authors
Dear Authors,
This review is well-written, with a wealth of information and a clear structure that aligns with the summary presented in the Abstract.
I have just a few minor comments.
Table 1 Legend (line 93)
- The abbreviations should be corrected: AMS should be changed to ASM, and SBBP to SPPB.
- I recommend adding the full name for the abbreviation EWGSOP2 in the Table 1 legend.
I also suggest consistently using the abbreviation EWGSOP2 (from 2018) in the Introduction (section 1.1, lines 66, 72, and 77).
Lines 132-134
“These include the monitoring of oxygen partial pressure in blood for the control of breathing, erythropoietin synthesis, and the regulation of vascular tone. (xi).
The sentence appears disconnected from the preceding text. Please consider clarifying its relevance or removing it. Specifically, how is monitoring partial pressure of oxygen or EPO synthesis related to the harmful or beneficial effects of ROS/RNS, i.e. their high or moderate concentration?
Line 637
The full name of the abbreviation RCS was already provided earlier (line 542), so it may not be necessary to repeat it.
The reference list is not fully in line with the journal’s formatting guidelines. Please revise all references carefully to ensure consistency with the required style.
Author Response
Dear Reviewer,
We gratefully acknowledge your review of our manuscript and useful suggestions to improve it. We have responded to your comments as detailed below:
Table 1 Legend (line 93):
- The abbreviations should be corrected: AMS should be changed to ASM, and SBBP to SPPB.
- I recommend adding the full name for the abbreviation EWGSOP2 in the Table 1 legend.
I also suggest consistently using the abbreviation EWGSOP2 (from 2018) in the Introduction (section 1.1, lines 66, 72, and 77).
The requested changes have been made (lines 97-98 in Table 1 legend, and lines 71 and 81 in the Introduction).
Lines 132-134:
“These include the monitoring of oxygen partial pressure in blood for the control of breathing, erythropoietin synthesis, and the regulation of vascular tone. (xi)”:
The sentence appears disconnected from the preceding text. Please consider clarifying its relevance or removing it. Specifically, how is monitoring partial pressure of oxygen or EPO synthesis related to the harmful or beneficial effects of ROS/RNS, i.e. their high or moderate concentration?
This sentence has been removed. EPO synthesis and the regulation of vascular tone were examples of regulatory actions unrelated to muscle cells enabled in living organisms by oxidative modifications of proteins. In rewriting section 2.2 (formerly section 2.1), to address comments from other reviewers, we have introduced a detailed discussion of specific examples of biological responses of muscle cells to increased ROS/RNS levels associated with muscle contraction, which reflect oxidative modifications of muscle proteins. These include the opening of the Ca2+ release channel in the sarcoplasmic reticulum (lines 492-514) and the enhancement of the interaction between the fast isoform of troponin I (TnIf) and troponin C, both mediated by S-glutathionylation (lines 515-522).
Line 637:
The full name of the abbreviation RCS was already provided earlier (line 542), so it may not be necessary to repeat it.
The requested change has been made.
The reference list is not fully in line with the journal’s formatting guidelines. Please revise all references carefully to ensure consistency with the required style.
Consistency in reference formatting has been ensured, in accordance with the Journal’s formatting criteria.

Reviewer 2 Report
Comments and Suggestions for Authors
The research paper titled " A reassessment of sarcopenia from a redox perspective as a basis for preventive and therapeutic interventions" discusses the perturbations of redox balance as a contributing factor to the pathogenesis of sarcopenia. The review distinguishes between oxidative eustress as a physiological response of muscle cells to mild stimulation. The role of oxidative damage to biological macromolecules, both direct and mediated by advanced lipid peroxidation endproducts and advanced glycation/glycoxidation endproducts, is examined in detail. Next, the review discusses antioxidant defense mechanisms. The review then discusses criteria for a rational classification of non-enzymatic antioxidants, according to their biochemical properties and mechanisms of action, considering also their pharmacokinetics and pharmacodynamics. This is followed by a review of some of the main radical-trapping antioxidants, both phenolic and non-phenolic, whose characteristics are compared. This study provides a basis for the selection of appropriate and rational use of antioxidants to prevent, delay and counteract the adverse consequences of sarcopenia.
However, I therefore have to point out some comments:
Abstract: It is recommended to articulate the study's significance, necessity, and innovative aspects at the outset of the abstract.
Line 98-100: Explan “the clinical and the pathogenetic standpoint” by providing information.
Line106: The term " frailty spectrum" should be defined for clarity.
Line 111-112: Please support the statement that 'PF has been shown to be predictive of adverse outcomes' with specific evidence or references to relevant studies
Line 125-130: This section makes several claims regarding ROS/RNS without citing appropriate references. Please include supporting literature.
Line 130: Please provide specific examples of the types of cellular or tissue toxicity caused by ROS/RNS at high concentrations.
Line 151–154: "In sarcopenia, the decrease in lean muscle mass is a reflection of the decreased number of total muscle fibers, particularly of type II..." This important point about fiber-type switching deserves a citation to recent literature and would benefit from further elaboration on the metabolic implications of type I fiber predominance.
Line 196–197: "Increased ROS/RNS production is necessary for the adaptation of skeletal muscle to physical exercise of sufficient intensity..." This is a central idea (oxidative eustress), but the statement remains too general. Please specify what level/intensity or duration of exercise is associated with this beneficial response.
Line 237–239: "Furthermore, as ageing progresses, the ability of cells to resist stress may be progressively compromised..." Consider linking this concept with immunosenescence or inflammaging, both of which are redox-relevant and commonly discussed in aging muscle physiology.
Line 267–271: "Sarcopenia in Sod1-/- mice may result from a two-hit mechanism..." This mechanism is compelling. Please consider adding a simple schematic figure to illustrate this two-hit pathway involving motor neurons and NMJ feedback.
Line 284-293: Clarify the term " nor any significant increase" and “significantly reduced” by providing specific data.
Line 330: Clarify the term “the state of energy resources”.
Line 384–387: "It appears that redox imbalance is a main cause of multiple tissue, cell and molecular disorders..." This sentence is too broad and should be supported by specific references to major findings from Sections 2 and 3. Otherwise, it reads like an unsupported claim.
Line414: "The effects of ROS/RNS, ALEs, and AGEs on scavenger receptor pathways should be elaborated upon. Consider discussing specific receptor types (e.g., RAGE, CD36) and downstream consequences."
Line 578–580: "These data show that not all ROS/RNS are harmful for muscle..." Important conceptual statement, but it would benefit from inclusion of a specific example of protective ROS (e.g., H₂O₂-induced PGC-1α signaling) to reinforce the redox balance concept.
Line 608-610, 647-652, 653-658, 695-696, 775-778, 1029-1033, 1033-1037,1564-1567, 1319-1320: add reference.
Line 696–697: "The age-associated decline in antioxidant responses was due in part to dysfunctions in redox signaling mediated by NRF2..." NRF2 is mentioned throughout the paper but inconsistently contextualized. Consider adding a dedicated paragraph summarizing its central role in redox adaptation and therapeutic targeting.
Line 713–716 (start of Section 3.5): "Glycation endproducts may be formed by the non-enzymatic, non-oxidative addition of glucose..." Please clarify the link between AGEs and redox status, as glycation is typically considered a parallel but not directly ROS-mediated process. How does glycoxidation differ in aging muscle?
Line 763, 945, 968, 1300: The font should be consistent
References
Line1651-1654: Several references cited are over 10 years old. Please update key citations with more recent literature (within the last 5 years) to reflect the current state of knowledge.
The DOI is correctly formatted but check if all digital object identifiers (DOIs) are included for other references where available.
The order of references should follow “1,2,3…..”
Authors should be separated by “;” and font size should be standardized.
The use of journal name abbreviations should be consistent. If " J Gerontol A Biol Sci Med Sci" is abbreviated here, ensure all other journal names are treated similarly.
Author Response
Reply to Reviewer no. 2
Dear Reviewer,
We gratefully acknowledge your careful and detailed review of our manuscript. Your expert suggestions prompted us to subject it to a very thorough and comprehensive revision, which included numerous additions and the rewriting of large portions of the body of the text. We believe that this effort, while demanding, has led to a significant improvement in our manuscript, and we express our appreciation for your thoughtful review.
We have responded to all of your comments as detailed below:
Abstract: It is recommended to articulate the study's significance, necessity, and innovative aspects at the outset of the abstract.
The initial part of the abstract has been reworded (please see lines 43-49)
Line 98-100: Explain “the clinical and the pathogenetic standpoint” by providing information.
We have better elaborated and reworded this paragraph (please see lines 103-115):
Line106: The term " frailty spectrum" should be defined for clarity.
A paragraph concerning the spectrum of functional impairments and vulnerabilities included in the definition of frailty, their contributing and risk factors has been added (please see lines 121-130).
Line 111-112: Please support the statement that 'PF has been shown to be predictive of adverse outcomes' with specific evidence or references to relevant studies
A paragraph in regard has been added (please see lines 137-148)
Line 125-130: This section makes several claims regarding ROS/RNS without citing appropriate references. Please include supporting literature.
We have moved this paragraph to the beginning of renumbered section 2.3 (lines 524-529) and we have added at this point a novel reference to an authoritative review article (Moldogazieva et al., 2020) that covers extensively the topic. We have also added a reference to section 3.2, titled ”ROS/RNS and their sources”, where the topic is discussed in more detail, with supporting bibliographical citations: “as discussed in further detail in Section 3.2 below”.
Line 130: Please provide specific examples of the types of cellular or tissue toxicity caused by ROS/RNS at high concentrations.
Line 196–197: "Increased ROS/RNS production is necessary for the adaptation of skeletal muscle to physical exercise of sufficient intensity..." This is a central idea (oxidative eustress), but the statement remains too general. Please specify what level/intensity or duration of exercise is associated with this beneficial response.
These two points are tightly linked and therefore have been addressed jointly.
We have addressed these points robustly by introducing a new section 2.1, titled “The continuum of reactive oxygen/nitrogen species (ROS/RNS)-incited responses in skeletal muscle” (lines 160-307). We describe therein: 1) the sources of ROS/RNS production in muscle cells during muscular contraction; 2) the concept of mitohormesis and the distinctive criteria between oxidative eustress and oxidative distress, with special regard to: a) the H2O2 concentrations that were shown to be necessary for physiological quiescent metabolism; b) those that can sustain muscle cell adaptation to exercise; and c) those that can lead to muscle and neuronal cell damage, muscle injury, and inflammation; 3) the limitations of such estimates, the technical difficulties of real-time assays of intracellular H2O2 levels in contracting muscles, and an analytical methodology for the H2O2 measurement in muscle fibers; 4) the different regimens of physical exercise, and those among them that have been reported to be associated with beneficial effects in sarcopenic older adults. These points are discussed extensively with many updated bibliographic references.
In view of the addition of this new section 2.1, we have been prompted to rewrite and rename large parts of section 2.2 (formerly 2.1), which is now titled “ROS/RNS-mediated adaptation of skeletal muscle to exercise” (lines 309-319, 377-409, 415-440, 456-522). The improved section 2.2 now contains a more comprehensive and detailed description of the redox-regulated responses that mediate muscle cells adaptation to exercise. A new Figure #1 has been added to accompany the new version of this section.
Furthermore, following the changes made to sections 2.1 and 2.2, we have also rewritten parts of section 2.3 (formerly 2.2), titled “Oxidative stress as a pathophysiological response of muscle cells in ageing and disease” (lines 524-529, 605-630, 639-645, 653-663, 667-668, 684-701, 705-709). Major changes and additions to this section pertain to oxidative mtDNA damage, the effects of S-glutathionylation and S-nitrosylation on ATP synthase, and a final paragraph that summarizes the entire section 2 (lines 742-752).
Line 151–154: "In sarcopenia, the decrease in lean muscle mass is a reflection of the decreased number of total muscle fibers, particularly of type II..." This important point about fiber-type switching deserves a citation to recent literature and would benefit from further elaboration on the metabolic implications of type I fiber predominance.
We have moved this point from section 2.2 (formerly 2.1) to section 2.3 (formerly 2.2) and further elaborated it, adding an additional reference (please see lines 553-563):
Line 237–239: "Furthermore, as ageing progresses, the ability of cells to resist stress may be progressively compromised..." Consider linking this concept with immunosenescence or inflammaging, both of which are redox-relevant and commonly discussed in aging muscle physiology.
We have added a new paragraph at this point (please see lines 532-546)
Line 267–271: "Sarcopenia in Sod1-/- mice may result from a two-hit mechanism..." This mechanism is compelling. Please consider adding a simple schematic figure to illustrate this two-hit pathway involving motor neurons and NMJ feedback.
A new schematic figure (Fig. 2) has been added, as requested.
Line 284-293: Clarify the term " nor any significant increase" and “significantly reduced” by providing specific data.
These terms refer to comparative measurements performed and reported in the article cited, whose significance resides not in their absolute values, but in the absence/presence of significant variation between test and control conditions.
Line 330: Clarify the term .“the state of energy resources”.
The sentence has been integrated as follows: “… the state of energy, resources, namely the intracellular AMP/ATP ratio“ (line 677)
Line 384–387: "It appears that redox imbalance is a main cause of multiple tissue, cell and molecular disorders..." This sentence is too broad and should be supported by specific references to major findings from Sections 2 and 3. Otherwise, it reads like an unsupported claim.
This statement has been deleted from revised section 2.2 (formerly 2.1).
Line 414: "The effects of ROS/RNS, ALEs, and AGEs on scavenger receptor pathways should be elaborated upon. Consider discussing specific receptor types (e.g., RAGE, CD36) and downstream consequences."
We have added at this point a reference to section 3.6 “Interactions of oxidized macromolecules with scavenger receptors” lines 770-771). Section 3.6 has been largely rewritten and expanded with new data and several novel references (please see main text, lines 1268-1345).
Line 578–580: "These data show that not all ROS/RNS are harmful for muscle..." Important conceptual statement, but it would benefit from inclusion of a specific example of protective ROS (e.g., H₂O₂-induced PGC-1α signaling) to reinforce the redox balance concept.
The dual nature of ROS/RNS-mediated effects in muscle across a spectrum of concentrations is now discussed in the dedicated section 2.1. Oxidation-induced PGC-1α signaling is also discussed extensively in the revised section 2.2 (formerly 2.1), along with other examples of protective actions of ROS. The sentence in line 578 has been reworded as follows:
“Thus, ROS/RNS may be beneficial or detrimental to muscle, as it has been discussed in section 2.1 above. Distinguishing between oxidative eustress and distress may provide opportunities to counteract sarcopenia.” (lines 938-940).
Line 608-610, 647-652, 653-658, 695-696, 775-778, 1029-1033, 1033-1037,1564-1567, 1319-1320: add reference.
Lines 608-610 (now lines 965-969):
The following references have been added:
Jaganjac, M.; Tirosh, O.; Cohen, G.; Sasson, S.; Zarkovic, N. Reactive aldehydes – second messengers of free radicals in diabetes mellitus. Free Radic. Res. 2013, 47(suppl 1), 39-48. doi: 10.3109/10715762.2013.789136. PMID: 23521622.
Schaur, R.J.; Siems, W.; Bresgen, N.; Eckl, P.M. 4-Hydroxy-nonenal - A bioactive lipid peroxidation product. Biomolecules 2015, 5, 2247-337. doi: 10.3390/biom5042247. PMID: 26437435; PMCID: PMC4693237.
Lines 647-652:
This paragraph has been rewritten and enlarged with new bibliographic references (please see main text, lines 1036-1085):
Lines 653-658:
This paragraph has been rewritten, with four novel references added (please see main text, lines 1086-1100):
Lines 695-696 (now lines 1127-1130):
This sentence is one with the following sentence, with which it shares the reference. The two propositions have been brought together in the same period, to make this more explicit.
Lines 775-778 (now lines 1227-1231):
The same consideration made in the previous point also applies to this sentence and the following one, which have been joined too.
Lines1029-1033 and 1033-1037 (now lines 1528-1536):
There were already relevant references in this section, but they needed to be better positioned, which we have done.
Lines 1319-1320 (now line 1811):
The following reference has been added:
Meng, X.; Zhou, J.; Zhao, C.N.; Gan, R.Y.; Li, H.B. Health benefits and molecular mechanisms of resveratrol: A narrative review. Foods 2020, 9, 340. doi: 10.3390/foods9030340. PMID: 32183376; PMCID: PMC7143620.
Lines 1564-1567:
This sentence has been reworded, with two novel references added (please see lines 2013-2017):
Lines 696–697: "The age-associated decline in antioxidant responses was due in part to dysfunctions in redox signaling mediated by NRF2..." NRF2 is mentioned throughout the paper but inconsistently contextualized. Consider adding a dedicated paragraph summarizing its central role in redox adaptation and therapeutic targeting.
The role of NRF-2 in muscle adaptation to oxidative stress is discussed in the revised section 2.2 (formerly 2.1), consistent with the purpose of that section. A dedicated section on NRF2 follows (section 4.3 “NRF2, the master regulator of antioxidant responses”), which was already present in the first version of the manuscript.
Lines 713–716 (start of Section 3.5): "Glycation endproducts may be formed by the non-enzymatic, non-oxidative addition of glucose..." Please clarify the link between AGEs and redox status, as glycation is typically considered a parallel but not directly ROS-mediated process. How does glycoxidation differ in aging muscle?
We have integrated and rewritten a large part of section 3.5 to address this point (please see the revised maintext, lines 1146-1171, 1181-1182), highlighting the close interplay between the formation of AGEs and oxidative stress. We have also further elaborated on the effects of AGEs on ageing muscle and the peculiarities of glycoxidation in sarcopenia (lines 1222-1266). Figure 4 (formerly Fig. 3) has been redrawn and seven updated references have been added to reflect these improvements.
Lines 763, 945, 968, 1300: The font should be consistent
Font consistency has been ensured.
Line 1651-1654: Several references cited are over 10 years old. Please update key citations with more recent literature (within the last 5 years) to reflect the current state of knowledge.
Sixty-four (64) obsolete references have been replaced with more recent ones, to take into account developments in the field. In several cases, this has necessitated rewording the text containing the citation. Additionally, due to the reorganization of large parts of the body of the text, some references introduced to replace old ones may have been relocated. Numerous additional updated references have also been added, reflecting the additions made to the text in response to other comments.
Obsolete references eliminated (64):
[7] Campbell, A.J.; Buchner D.M. Unstable disability and the fluctuations of frailty. Age Ageing 1997, 26, 315-318. doi: 10.1093/ageing/26.4.315. PMID: 9271296.
[11] Dröge, W. Free radicals in the physiological control of cell function. Physiol. Rev. 2002, 82, 47-95. doi: 10.1152/physrev.00018.2001. PMID: 11773609.
[17] Jackson, M.J. Free radicals generated by contracting muscle: by-products of metabolism or key regulators of muscle function? Free Radic. Biol. Med. 2008, 44, 132-141. doi: 10.1016/j.freeradbiomed.2007.06.003. PMID: 18191749.
replaced with
[21] Jackson, M.J. Exercise-induced adaptations to homeostasis of reactive oxygen species in skeletal muscle. Free Radic. Biol. Med. 2024, 225, 494-500. doi: 10.1016/j.freeradbiomed.2024.10.270. PMID: 39427746.
[20] Turrens, J.F. Mitochondrial formation of reactive oxygen species. J. Physiol. 2003, 552(Pt 2), 335-344. doi: 10.1113/jphysiol.2003.049478. PMID: 14561818; PMCID: PMC2343396.
replaced with
[27] Molnar, A.M.; Servais, S.; Guichardant, M.; Lagarde, M.; Macedo, D.V.; Pereira-Da-Silva, L.; Sibille, B.; Favier, R. Mitochondrial H2O2 production is reduced with acute and chronic eccentric exercise in rat skeletal muscle. Antioxid. Redox Signal. 2006, 8, 548-558. doi: 10.1089/ars.2006.8.548. PMID: 16677099.
[22] Le Moal E, Pialoux V, Juban G, Groussard C, Zouhal H, Chazaud B, Mounier R. Redox Control of Skeletal Muscle Regeneration. Antioxid Redox Signal. 2017 Aug 10;27(5):276-310. doi: 10.1089/ars.2016.6782. Epub 2017 Feb 6. PMID: 28027662; PMCID: PMC5685069
replaced with
[24] Andrés, C.M.C.; Pérez de la Lastra, J.M.; Andrés Juan, C.; Plou, F.J.; Pérez-Lebeña, E. Superoxide Anion Chemistry - Its Role at the Core of the Innate Immunity. Int. J. Mol. Sci. 2023, 24, 1841. doi: 10.3390/ijms24031841. PMID: 36768162; PMCID: PMC9916283.
[23] Sies H. Role of metabolic H2O2 generation: redox signaling and oxidative stress. J Biol Chem. 2014 Mar 28;289(13):8735-41. doi: 10.1074/jbc.R113.544635. Epub 2014 Feb 10. PMID: 24515117; PMCID: PMC3979367
[25] Murphy MP. How mitochondria produce reactive oxygen species. Biochem J. 2009 Jan 1;417(1):1-13. doi: 10.1042/BJ20081386. PMID: 19061483; PMCID: PMC2605959.
[28] Loureiro AC, do Rêgo-Monteiro IC, Louzada RA, Ortenzi VH, de Aguiar AP, de Abreu ES, Cavalcanti-de-Albuquerque JP, Hecht F, de Oliveira AC, Ceccatto VM, Fortunato RS, Carvalho DP. Differential Expression of NADPH Oxidases Depends on Skeletal Muscle Fiber Type in Rats. Oxid Med Cell Longev. 2016;2016:6738701. doi: 10.1155/2016/6738701. Epub 2016 Oct 26. PMID: 27847553; PMCID: PMC5101397
[29] Sakellariou GK, Davis CS, Shi Y, Ivannikov MV, Zhang Y, Vasilaki A, Macleod GT, Richardson A, Van Remmen H, Jackson MJ, McArdle A, Brooks SV. Neuron-specific expression of CuZnSOD prevents the loss of muscle mass and function that occurs in homozygous CuZnSOD-knockout mice. FASEB J. 2014 Apr;28(4):1666-81. doi: 10.1096/fj.13-240390. Epub 2013 Dec 30. PMID: 24378874; PMCID: PMC3963022
[36] Stamler, J.S.; Meissner, G. Physiology of nitric oxide in skeletal muscle. Physiol. Rev. 2001, 81, 209-237. doi: 10.1152/physrev.2001.81.1.209. PMID: 11152758.
replaced with
[30] Witherspoon, J.W.; Meilleur, K.G. Review of RyR1 pathway and associated pathomechanisms. Acta Neuropathol. Commun. 2016, 4, 121. doi: 10.1186/s40478-016-0392-6. PMID: 27855725; PMCID: PMC5114830.
[37] Ji, L.L. Modulation of skeletal muscle antioxidant defense by exercise: Role of redox signaling. Free Radic. Biol. Med. 2008, 44, 142-152. doi: 10.1016/j.freeradbiomed.2007.02.031. PMID: 18191750.
replaced with
[54] Ji, L.L.; Kang, C.; Zhang, Y. Exercise-induced hormesis and skeletal muscle health. Free Radic. Biol. Med. 2016, 98, 113-122. doi: 10.1016/j.freeradbiomed.2016.02.025. PMID: 26916558.
[38] Morris, G.; Anderson, G.; Berk, M.; Maes, M. Coenzyme Q10 depletion in medical and neuropsychiatric disorders: potential repercussions and therapeutic implications. Mol. Neurobiol. 2013, 48, 883-903. doi: 10.1007/s12035-013-8477-8. PMID: 23+]761046.
replaced with
[28] Powers, S.K.; Goldstein, E.; Schrager, M.; Ji, L.L. Exercise training and skeletal muscle antioxidant enzymes: An update. Antioxidants (Basel) 2022, 12, 39. doi: 10.3390/antiox12010039. PMID: 36670901; PMCID: PMC9854578.
[43] Powers, S.K.; Radak, Z.; Ji, L.L.; Jackson, M. Reactive oxygen species promote endurance exercise-induced adaptations in skeletal muscles. J. Sport Health Sci. 2024, 13, 780-792. doi: 10.1016/j.jshs.2024.05.001. PMID: 38719184; PMCID: PMC11336304.
[39] Vollaard, N.B.; Shearman, J.P.; Cooper, C.E. Exercise-induced oxidative stress: myths, realities and physiological relevance. Sports Med. 2005, 35, 1045-1062. doi: 10.2165/00007256-200535120-00004. PMID: 16336008.
replaced with
[76] Dent, J.R.; Stocks, B.; Campelj, D.G.; Philp, A. Transient changes to metabolic homeostasis initiate mitochondrial adaptation to endurance exercise. Semin. Cell Dev. Biol. 2023, 143, 3-16. doi: 10.1016/j.semcdb.2022.03.022. PMID: 35351374.
[40] Merry, T.L.; Steinberg, G.R.; Lynch, G.S.; McConell, G.K. Skeletal muscle glucose uptake during contraction is regulated by nitric oxide and ROS independently of AMPK. Am. J. Physiol. Endocrinol. Metab. 2010, 298, E577-585. doi: 10.1152/ajpendo.00239.2009. PMID: 20009026.
replaced with
[81] Kong, S.; Cai, B.; Nie, Q. PGC-1a affects skeletal muscle and adipose tissue development by regulating mitochondrial biogenesis. Mol. Genet. Genomics 2022, 297, 621–633. https://doi.org/10.1007/s00438-022-01878-2
[41] Powers SK, Talbert EE, Adhihetty PJ. Reactive oxygen and nitrogen species as intracellular signals in skeletal muscle. J Physiol. 2011 May 1;589(Pt 9):2129-38. doi: 10.1113/jphysiol.2010.201327. Epub 2011 Jan 4. PMID: 21224240; PMCID: PMC3098692.
[42] Xie Z, Dong Y, Zhang M, Cui MZ, Cohen RA, Riek U, Neumann D, Schlattner U, Zou MH. Activation of protein kinase C zeta by peroxynitrite regulates LKB1-dependent AMP-activated protein kinase in cultured endothelial cells. J Biol Chem. 2006 Mar 10;281(10):6366-75. doi: 10.1074/jbc.M511178200. Epub 2006 Jan 9. PMID: 16407220
replaced with
[78] Kjøbsted, R.; Hingst, J.R.; Fentz, J.; Foretz, M.; Sanz, M.N.; Pehmøller, C.; Shum, M.; Marette, A.; Mounier, R.; Treebak, J.T.; Wojtaszewski, J.F.P.; Viollet, B.; Lantier, L. AMPK in skeletal muscle function and metabolism. FASEB J. 2018, 32, 1741-1777. doi: 10.1096/fj.201700442R. PMID: 29242278; PMCID: PMC5945561.
[43] Schroeder MA, Atherton HJ, Ball DR, Cole MA, Heather LC, Griffin JL, Clarke K, Radda GK, Tyler DJ. Real-time assessment of Krebs cycle metabolism using hyperpolarized 13C magnetic resonance spectroscopy. FASEB J. 2009 Aug;23(8):2529-38. doi: 10.1096/fj.09-129171. Epub 2009 Mar 27. PMID: 19329759; PMCID: PMC2717776.
[44] Wakil SJ, Abu-Elheiga LA. Fatty acid metabolism: target for metabolic syndrome. J Lipid Res. 2009 Apr;50 Suppl(Suppl):S138-43. doi: 10.1194/jlr.R800079-JLR200. Epub 2008 Dec 1. PMID: 19047759; PMCID: PMC2674721
[45] Erickson JR, Joiner ML, Guan X, Kutschke W, Yang J, Oddis CV, Bartlett RK, Lowe JS, O'Donnell SE, Aykin-Burns N, Zimmerman MC, Zimmerman K, Ham AJ, Weiss RM, Spitz DR, Shea MA, Colbran RJ, Mohler PJ, Anderson ME. A dynamic pathway for calcium-independent activation of CaMKII by methionine oxidation. Cell. 2008 May 2;133(3):462-74. doi: 10.1016/j.cell.2008.02.048. PMID: 18455987; PMCID: PMC2435269
[46] Erickson JR, He BJ, Grumbach IM, Anderson ME. CaMKII in the cardiovascular system: sensing redox states. Physiol Rev. 2011 Jul;91(3):889-915. doi: 10.1152/physrev.00018.2010. PMID: 21742790; PMCID: PMC3732780.
[47] Hawley, J.A.; Hargreaves, M.; Zierath, J.R. Signalling mechanisms in skeletal muscle: role in substrate selection and muscle adaptation. Essays Biochem. 2006, 42, 1-12. doi: 10.1042/bse0420001. PMID: 17144876.ß
replaced with
[90] Vichaiwong, K.; Purohit, S.; An, D.; Toyoda, T.; Jessen, N.; Hirshman, M.F.; Goodyear, L.J. Contraction regulates site-specific phosphorylation of TBC1D1 in skeletal muscle. Biochem. J. 2010, 431, 311-320. doi: 10.1042/BJ20101100. PMID: 20701589; PMCID: PMC2947193.
[48] Egan B, Zierath JR. Exercise metabolism and the molecular regulation of skeletal muscle adaptation. Cell Metab. 2013 Feb 5;17(2):162-84. doi: 10.1016/j.cmet.2012.12.012. PMID: 23395166.
replaced with
[91] Frøsig, C.; Pehmøller, C.; Birk, J.B.; Richter, E.A.; Wojtaszewski, J.F. Exercise-induced TBC1D1 Ser237 phosphorylation and 14-3-3 protein binding capacity in human skeletal muscle. J. Physiol. 2010, 588, 4539-4548. doi: 10.1113/jphysiol.2010.194811. PMID: 20837646; PMCID: PMC3008856.
[49] Cuschieri J, Maier RV. Mitogen-activated protein kinase (MAPK). Crit Care Med. 2005 Dec;33(12 Suppl):S417-9. doi: 10.1097/01.ccm.0000191714.39495.a6. PMID: 16340409.
[50] Powers SK, Duarte J, Kavazis AN, Talbert EE. Reactive oxygen species are signalling molecules for skeletal muscle adaptation. Exp Physiol. 2010 Jan;95(1):1-9. doi: 10.1113/expphysiol.2009.050526. Epub 2009 Oct 30. PMID: 19880534; PMCID: PMC2906150.
replaced with
[28] Powers, S.K.; Goldstein, E.; Schrager, M.; Ji, L.L. Exercise training and skeletal muscle antioxidant enzymes: An update. Antioxidants (Basel) 2022, 12, 39. doi: 10.3390/antiox12010039. PMID: 36670901; PMCID: PMC9854578.
[51] Bassel-Duby R, Olson EN. Signaling pathways in skeletal muscle remodeling. Annu Rev Biochem. 2006;75:19-37. doi: 10.1146/annurev.biochem.75.103004.142622. PMID: 16756483.
replaced with
[59] Zhou, Y.; Zhang, X.; Baker, J.S.; Davison, G.W.; Yan, X. Redox signaling and skeletal muscle adaptation during aerobic exercise. iScience 2024, 27, 109643. doi: 10.1016/j.isci.2024.109643. PMID: 38650987; PMCID: PMC11033207.
[65] Jackson, M.J.; Stretton, C.; McArdle, A. Hydrogen peroxide as a signal for skeletal muscle adaptations to exercise: What do concentrations tell us about potential mechanisms? Redox Biol. 2020, 35, 101484. doi: 10.1016/j.redox.2020.101484. PMID: 32184060; PMCID: PMC7284923.
[54] Harman D. The free radical theory of ageing. Antioxid Redox Signal. 2003 Oct;5(5):557-61. doi: 10.1089/152308603770310202. PMID: 14580310.
[55] Le Bourg, E. Delaying ageing: could the study of hormesis be more helpful than that of the genetic pathway used to survive starvation? Biogerontology 2003, 4, 319-324. doi: 10.1023/a:1026255519223. PMID: 14618029.
replaced with
[115] Santoro, A.; Bientinesi, E.; Monti, D. Immunosenescence and inflammaging in the aging process: age-related diseases or longevity? Ageing Res. Rev. 2021, 71, 101422. doi: 10.1016/j.arr.2021.101422. PMID: 34391943.
[56] Arumugam, T.V.; Gleichmann, M.; Tang, S.C.; Mattson, M.P. Hormesis/preconditioning mechanisms, the nervous system and ageing. Ageing Res. Rev. 2006, 5, 165-178. doi: 10.1016/j.arr.2006.03.003. PMID: 16682262.
replaced with
[51] López-Otín, C.; Blasco, M.A.; Partridge, L.; Serrano, M.; Kroemer, G. Hallmarks of aging: An expanding universe. Cell 2023, 186, 243-278. doi: 10.1016/j.cell.2022.11.001. PMID: 36599349.
[65] Aiken, J.; Bua, E.; Cao, Z.; Lopez, M.; Wanagat, J.; McKenzie, D.; McKiernan, S. Mitochondrial DNA deletion mutations and sarcopenia. Ann. N. Y. Acad. Sci. 2002, 959, 412-423. doi: 10.1111/j.1749-6632.2002.tb02111.x. PMID: 11976214.
[66] Bua E, Johnson J, Herbst A, Delong B, McKenzie D, Salamat S, Aiken JM. Mitochondrial DNA-deletion mutations accumulate intracellularly to detrimental levels in aged human skeletal muscle fibers. Am J Hum Genet. 2006 Sep;79(3):469-80. doi: 10.1086/507132. Epub 2006 Jul 7. PMID: 16909385; PMCID: PMC1559550.
replaced with
[137] Van Houten, B.; Hunter, S.E.; Meyer, J.N. Mitochondrial DNA damage induced autophagy, cell death, and disease. Front. Biosci. (Landmark Ed.) 2016, 21, 42-54. doi: 10.2741/4375. PMID: 26709760; PMCID: PMC4750375.
[67] Hiona, A.; Leeuwenburgh, C. The role of mitochondrial DNA mutations in ageing and sarcopenia: implications for the mitochondrial vicious cycle theory of ageing. Exp. Gerontol. 2008, 43, 24-33. doi: 10.1016/j.exger.2007.10.001. PMID: 17997255; PMCID: PMC2225597.
replaced with
[131] Hiona, A.; Sanz, A.; Kujoth, G.C.; Pamplona, R.; Seo, A.Y.; Hofer, T.; Someya, S.; Miyakawa, T.; Nakayama, C.; Samhan-Arias, A.K.; Servais, S.; Barger, J.L.; Portero-Otín, M.; Tanokura, M.; Prolla, T.A.; Leeuwenburgh, C. Mitochondrial DNA mutations induce mitochondrial dysfunction, apoptosis and sarcopenia in skeletal muscle of mitochondrial DNA mutator mice. PLoS ONE 2010, 5, e11468. doi: 10.1371/journal.pone.0011468. PMID: 20628647; PMCID: PMC2898813.
[68] Shah, V.O.; Scariano, J.; Waters, D.; Qualls, C.; Morgan, M.; Pickett, G.; Gasparovic, C.; Dokladny, K.; Moseley, P.; Raj, D.S. Mitochondrial DNA deletion and sarcopenia. Genet. Med. 2009, 11, 147-152. doi: 10.1097/GIM.0b013e31819307a2. PMID: 19367187; PMCID: PMC3737247.
[71] Alexeyev, M.F.; Ledoux, S.P.; Wilson, G.L. Mitochondrial DNA and ageing. Clin. Sci. (Lond.) 2004, 107, 355-364. doi: 10.1042/CS20040148. PMID: 15279618.
[73] Wanagat, J.; Ahmadieh, N.; Bielas, J.H.; Ericson, N.G.; Van Remmen, H. Skeletal muscle mitochondrial DNA deletions are not increased in CuZn-superoxide dismutase deficient mice. Exp. Gerontol. 2015, 61, 15-19. doi: 10.1016/j.exger.2014.11.012. PMID: 25449857; PMCID: PMC4289650.
[74] Cosso, R.G.; Turim, J.; Nantes, I.L.; Almeida, A.M.; Di Mascio, P.; Verces, A.E. Mitochondrial permeability transition induced by chemically generated singlet oxygen. J. Bioenerg. Biomembr. 2002, 34, 157-163. doi: 10.1023/a:1016075218162. PMID: 12171065.
replaced with
[134] Kaludercic, N.; Giorgio, V. The dual function of reactive oxygen/nitrogen species in bioenergetics and cell death: The role of ATP synthase. Oxid. Med. Cell Longev. 2016, 2016, 3869610. doi: 10.1155/2016/3869610. PMID: 27034734; PMCID: PMC4806282.
[75] Lemasters, J.J.; Nieminen, A.L.; Qian, T.; Trost, L.C.; Elmore, S.P.; Nishimura, Y.; Crowe, R.A.; Cascio, W.E.; Bradham, C.A.; Brenner, D.A.; Herman, B. The mitochondrial permeability transition in cell death: a common mechanism in necrosis, apoptosis and autophagy. Biochim. Biophys. Acta 1998, 1366, 177-196. doi: 10.1016/s0005-2728(98)00112-1. PMID: 9714796.
replaced with
[135] Bonora, M.; Giorgi, C.; Pinton, P. Molecular mechanisms and consequences of mitochondrial permeability transition. Nat. Rev. Mol. Cell Biol. 2022, 23, 266-285. doi: 10.1038/s41580-021-00433-y. PMID: 34880425.
[76] Lemasters, J.J.; Theruvath, T.P.; Zhong, Z.; Nieminen, A.L. Mitochondrial calcium and the permeability transition in cell death. Biochim. Biophys. Acta 2009, 1787, 1395-1401. doi: 10.1016/j.bbabio.2009.06.009. PMID: 19576166; PMCID: PMC2730424.
replaced with
[136] Endlicher, R.; Drahota, Z.; Štefková, K.; Červinková, Z.; Kučera, O. The mitochondrial permeability transition pore - Current knowledge of its structure, function, and regulation, and optimized methods for evaluating its functional state. Cells 2023, 12, 1273. doi: 10.3390/cells12091273. PMID: 37174672; PMCID: PMC10177258.
[77] Kirkinezos, I.G.; Moraes, C.T. Reactive oxygen species and mitochondrial diseases. Semin. Cell Dev. Biol. 2001, 12, 449-457. doi: 10.1006/scdb.2001.0282. PMID: 11735379.
replaced with
[133] Juan, C.A.; Pérez de la Lastra, J.M.; Plou, F.J.; Pérez-Lebeña, E. The chemistry of reactive oxygen species (ROS) revisited: Outlining their role in biological macromolecules (DNA, Lipids and Proteins) and induced pathologies. Int. J. Mol. Sci. 2021, 22, 4642. doi: 10.3390/ijms22094642. PMID: 33924958; PMCID: PMC8125527.
[78] Cadenas, E.; Davies, K.J. Mitochondrial free radical generation, oxidative stress, and ageing. Free Radic. Biol. Med. 2000, 29, 222-230. doi: 10.1016/s0891-5849(00)00317-8. PMID: 11035250.
replaced with
[137] Van Houten, B.; Hunter, S.E.; Meyer, J.N. Mitochondrial DNA damage induced autophagy, cell death, and disease. Front. Biosci. (Landmark Ed.) 2016, 21, 42-54. doi: 10.2741/4375. PMID: 26709760; PMCID: PMC4750375.
[81] Akhtari M, Jalalvand M, Sadr M, Sharifi H. Autophagy in the Cellular Consequences of Tobacco Smoking: Insights into Senescence. J Biochem Mol Toxicol. 2024 Dec;38(12):e70065. doi: 10.1002/jbt.70065. PMID: 395
[82] Li, L.; Li, W.; Liu, Y.; Han, B.; Yu, Y.; Lin, H. MEHP induced mitochondrial damage by promoting ROS production in CIK cells, leading to apoptosis, autophagy, cell cycle arrest. Comp. Biochem. Physiol. C Toxicol. Pharmacol. 2025, 288, 110064. doi: 10.1016/j.cbpc.2024.110064. PMID: 39586385.
[84] Kaushik, S.; Cuervo, AM. Autophagy as a cell-repair mechanism: activation of chaperone-mediated autophagy during oxidative stress. Mol. Aspects Med. 2006, 27, 444-454. doi: 10.1016/j.mam.2006.08.007. PMID: 16978688; PMCID: PMC1855281.
[85] Terman, A.; Brunk, U.T. Myocyte ageing and mitochondrial turnover. Exp. Gerontol. 2004, 39, 701-705. doi: 10.1016/j.exger.2004.01.005. PMID: 15130664.
[91] Scherz-Shouval, R.; Elazar, Z. ROS, mitochondria and the regulation of autophagy. Trends Cell Biol. 2007, 17, 422-427. doi: 10.1016/j.tcb.2007.07.009. PMID: 17804237.
replaced with
[50] Voronina, M.V.; Frolova, A.S.; Kolesova, E.P.; Kuldyushev, N.A.; Parodi, A.; Zamyatnin, A.A. Jr. The intricate balance between life and death: ROS, cathepsins, and their interplay in cell death and autophagy. Int. J. Mol. Sci. 2024, 25, 4087. doi: 10.3390/ijms25074087. PMID: 38612897; PMCID: PMC11012956.
[146] Hong, X.; Isern, J.; Campanario, S.; Perdiguero, E.; Ramírez-Pardo, I.; Segalés, J.; Hernansanz-Agustín, P.; Curtabbi, A.; Deryagin, O.; Pollán, A.; González-Reyes, J.A.; Villalba, J.M.; Sandri, M.; Serrano, A.L.; Enríquez, J.A.; Muñoz-Cánoves, P. Mitochondrial dynamics maintain muscle stem cell regenerative competence throughout adult life by regulating metabolism and mitophagy. Cell Stem Cell 2022, 29, 1298-1314.e10. doi: 10.1016/j.stem.2022.07.009. PMID: 35998641. Erratum in: Cell Stem Cell 2022, 29, 1506-1508. doi: 10.1016/j.stem.2022.09.002. PMID: 36206734.
[147] García-Prat, L.; Martínez-Vicente, M.; Perdiguero, E.; Ortet, L.; Rodríguez-Ubreva, J.; Rebollo, E.; Ruiz-Bonilla, V.; Gutarra, S.; Ballestar, E.; Serrano, A.L.; Sandri, M.; Muñoz-Cánoves, P. Autophagy maintains stemness by preventing senescence. Nature 2016, 529, 37-42. doi: 10.1038/nature16187. PMID: 26738589.
[149] Sahu, A.; Mamiya, H.; Shinde, S.N.; Cheikhi, A.; Winter, L.L.; Vo, N.V.; Stolz, D.; Roginskaya, V.; Tang, W.Y.; St Croix,C.; Sanders, L.H.; Franti, M.; Van Houten, B.; Rando, T.A.; Barchowsky, A.; Ambrosio, F. Age-related declines in α-Klotho drive progenitor cell mitochondrial dysfunction and impaired muscle regeneration. Nat. Commun. 2018, 9, 4859. doi: 10.1038/s41467-018-07253-3. PMID: 30451844; PMCID: PMC6242898.
[150] White, J.P.; Billin, A.N.; Campbell, M.E.; Russell, A.J.; Huffman, K.M.; Kraus, W.E. The AMPK/p27Kip1 Axis regulates autophagy/apoptosis decisions in aged skeletal muscle stem cells. Stem Cell Reports 2018, 11, 425-439. doi: 10.1016/j.stemcr.2018.06.014. PMID: 30033086; PMCID: PMC6093087.
[151] Bernet, J.D.; Doles, J.D.; Hall, J.K.; Kelly Tanaka, K.; Carter, T.A.; Olwin, B.B. p38 MAPK signaling underlies a cell-autonomous loss of stem cell self-renewal in skeletal muscle of aged mice. Nat. Med. 2014, 20, 265-271. doi: 10.1038/nm.3465. PMID: 24531379; PMCID: PMC4070883.
[152] Tierney, M.T.; Aydogdu, T.; Sala, D.; Malecova, B.; Gatto, S.; Puri, P.L.; Latella, L.; Sacco, A. STAT3 signaling controls satellite cell expansion and skeletal muscle repair. Nat. Med. 2014, 20, 1182-1186. doi: 10.1038/nm.3656. PMID: 25194572; PMCID: PMC4332844.
[92] Radak, Z.; Chung, H.Y.; Goto, S. Systemic adaptation to oxidative challenge induced by regular exercise. Free Radic. Biol. Med. 2008, 44, 153-159. doi: 10.1016/j.freeradbiomed.2007.01.029. PMID: 18191751.
[93] Radak, Z.; Chung, H.Y.; Koltai, E.; Taylor, A.W.; Goto, S. Exercise, oxidative stress and hormesis. Ageing Res. Rev. 2008, 7, 34-42. doi: 10.1016/j.arr.2007.04.004. PMID: 17869589.
replaced with
[53] Radak, Z.; Ishihara, K.; Tekus, E.; Varga, C.; Posa, A; Balogh, L.; Boldogh, I.; Koltai, E. Exercise, oxidants, and antioxidants change the shape of the bell-shaped hormesis curve. Redox Biol. 2017, 12, 285-290. doi: 10.1016/j.redox.2017.02.015. PMID: 28285189; PMCID: PMC5345970.
[110] Ebrahimian, T.G.; Heymes, C.; You, D.; Blanc-Brude, O.; Mees, B.; Waeckel, L.; Duriez, M.; Vilar, J.; Brandes, R.P.; Levy, B.I.; Shah, A.M.; Silvestre, J.S. NADPH oxidase-derived overproduction of reactive oxygen species impairs postischemic neovascularization in mice with type 1 diabetes. Am. J. Pathol. 2006, 169, 719-728. doi: 10.2353/ajpath.2006.060042. PMID: 16877369; PMCID: PMC1698801.
[112] Tidball, J.G.; Wehling-Henricks, M. The role of free radicals in the pathophysiology of muscular dystrophy. J. Appl. Physiol. (1985) 2007, 102, 1677-1686. doi: 10.1152/japplphysiol.01145.2006. PMID: 17095633.
replaced with
[167] Casati, S.R.; Cervia, D.; Roux-Biejat, P.; Moscheni, C.; Perrotta, C.; De Palma, C. Mitochondria and reactive oxygen species: The therapeutic balance of powers for Duchenne muscular dystrophy. Cells 2024, 13, 574. doi: 10.3390/cells13070574. PMID: 38607013; PMCID: PMC11011272.
[117] Thomas Brioche. Sarcopenie : mécanismes et prévention : rôle de l'exercice et de l'hormone de croissance : implication du stress oxydant et de la glucose-6-phosphate déshydrogénase. Education. Université Rennes 2; Universitat de Valencia (Espagne), 2014. Français. ⟨NNT : 2014REN20044⟩.
[125] Grune, T.; Merker, K.; Sandig, G.; Davies, K.J. Selective degradation of oxidatively modified protein substrates by the proteasome. Biochem. Biophys. Res. Commun. 2003, 305, 709-718. doi: 10.1016/s0006-291x(03)00809-x. PMID: 12763051.
replaced with
Jung, T.; Höhn, A.; Grune, T. The proteasome and the degradation of oxidized proteins: Part II - protein oxidation and proteasomal degradation. Redox Biol. 2014, 2, 99-104. doi: 10.1016/j.redox.2013.12.008. PMID: 25460724; PMCID: PMC4297946.
[146] Davies, K.J.; Delsignore, M.E. Protein damage and degradation by oxygen radicals. III. Modification of secondary and tertiary structure. J. Biol. Chem. 1987, 262, 9908-9913. PMID: 3036877.
replaced with
Pamplona, R.; Dalfó E.; Ayala V.; Bellmunt M.J.; Prat J.; Ferrer I.; Portero-Otín M. Proteins in human brain cortex are modified by oxidation, glycoxidation, and lipoxidation. Effects of Alzheimer disease and identification of lipoxidation targets. J. Biol. Chem. 2005, 280, 21522-21530. doi: 10.1074/jbc.M502255200. PMID: 15799962.
[160] Peng Y, Kim JM, Park HS, Yang A, Islam C, Lakatta EG, Lin L. AGE-RAGE signal generates a specific NF-κB RelA "barcode" that directs collagen I expression. Sci Rep. 2016 Jan 5;6:18822. doi: 10.1038/srep18822. PMID: 26729520; PMCID: PMC4700418.
[164] Heilbronn, L.K.; Ravussin, E. Calorie restriction and ageing: review of the literature and implications for studies in humans. Am. J. Clin. Nutr. 2003, 78, 361-369. doi: 10.1093/ajcn/78.3.361. PMID: 12936916.
[165] Fontana, L.; Meyer, T.E.; Klein, S.; Holloszy, J.O. Long-term calorie restriction is highly effective in reducing the risk for atherosclerosis in humans. Proc. Natl. Acad. Sci. USA 2004, 101, 6659-6663. doi:10.1073/pnas.0308291101. PMID: 15096581; PMCID: PMC404101.
[166] Radak, Z.; Chung, H.Y.; Goto, S. Exercise and hormesis: oxidative stress-related adaptation for successful ageing. Biogerontology 2005, 6, 71-75. doi: 10.1007/s10522-004-7386-7. PMID: 15834665.
replaced with
[53] Radak, Z.; Ishihara, K.; Tekus, E.; Varga, C.; Posa, A; Balogh, L.; Boldogh, I.; Koltai, E. Exercise, oxidants, and antioxidants change the shape of the bell-shaped hormesis curve. Redox Biol. 2017, 12, 285-290. doi: 10.1016/j.redox.2017.02.015. PMID: 28285189; PMCID: PMC5345970.
[55] Radak, Z.; Koltai, E.; Taylor, A.W.; Higuchi, M.; Kumagai, S.; Ohno, H.; Goto, S.; Boldogh, I. Redox-regulating sirtuins in aging, caloric restriction, and exercise. Free Radic. Biol. Med. 2013, 58, 87-97. doi: 10.1016/j.freeradbiomed.2013.01.004. PMID: 23339850.
[168] Ji, L.L.; Stratman, F.W.; Lardy, H.A. Antioxidant enzyme systems in rat liver and skeletal muscle. Influences of selenium deficiency, chronic training, and acute exercise. Arch. Biochem. Biophys. 1988, 263, 150-160. doi: 10.1016/0003-9861(88)90623-6. PMID: 3369860.
replaced with
[54] Ji, L.L.; Kang, C.; Zhang, Y. Exercise-induced hormesis and skeletal muscle health. Free Radic. Biol. Med. 2016, 98, 113-122. doi: 10.1016/j.freeradbiomed.2016.02.025. PMID: 26916558.
[169] Powers, S.K.; Criswell, D.; Lawler, J.; Martin, D.; Ji, L.L.; Herb, R.A.; Dudley, G. Regional training-induced alterations in diaphragmatic oxidative and antioxidant enzymes. Respir. Physiol. 1994, 95, 227-237. doi: 10.1016/0034-5687(94)90118-x. PMID: 8191043.
replaced with
[43] Powers, S.K.; Radak, Z.; Ji, L.L.; Jackson, M. Reactive oxygen species promote endurance exercise-induced adaptations in skeletal muscles. J. Sport Health Sci. 2024, 13, 780-792. doi: 10.1016/j.jshs.2024.05.001. PMID: 38719184; PMCID: PMC11336304.
[172] Pereira, B.; Costa Rosa, L.F.; Safi, D.A.; Medeiros, M.H.; Curi, R.; Bechara, E.J. Superoxide dismutase, catalase, and glutathione peroxidase activities in muscle and lymphoid organs of sedentary and exercise-trained rats. Physiol. Behav. 1994, 56, 1095-1099. doi: 10.1016/0031-9384(94)90349-2. PMID: 7824577.
replaced with
[248] Vasileiadou, O.; Nastos, G.G.; Chatzinikolaou, P.N.; Papoutsis, D.; Vrampa, D.I.; Methenitis, S.; Margaritelis, N.V. Redox profile of skeletal muscles: Implications for research design and interpretation. Antioxidants (Basel) 2023, 12, 1738. doi: 10.3390/antiox12091738. PMID: 37760040; PMCID: PMC10525275.
[173] Laughlin, M.H.; Simpson, T.; Sexton, W.L.; Brown, O.R.; Smith, J.K.; Korthuis, R.J. Skeletal muscle oxidative capacity, antioxidant enzymes, and exercise training. J. Appl. Physiol. (1985) 1990, 68, 2337-2343. doi: 10.1152/jappl.1990.68.6.2337. PMID: 2384414.
[186] Lillig, C.H.; Berndt, C.; Vergnolle, O.; Lönn, M.E.; Hudemann, C.; Bill, E.; Holmgren, A. Characterization of human glutaredoxin 2 as iron-sulfur protein: a possible role as redox sensor. Proc. Natl. Acad. Sci. USA 2005, 102, 8168-8173. doi: 10.1073/pnas.0500735102. PMID: 15917333; PMCID: PMC1149418.
replaced with
[258] Ouyang, Y.; Peng, Y.; Li. J.; Holmgren, A.; Lu, J. Modulation of thiol-dependent redox system by metal ions via thioredoxin and glutaredoxin systems. Metallomics 2018, 10, 218-228. doi: 10.1039/c7mt00327g. PMID: 29410996.
[191] Rodríguez-Manzaneque, M.T.; Tamarit, J.; Bellí, G.; Ros, J.; Herrero, E. Grx5 is a mitochondrial glutaredoxin required for the activity of iron/sulfur enzymes. Mol. Biol. Cell 2002, 13, 1109-1121. doi: 10.1091/mbc.01-10-0517. PMID: 11950925; PMCID: PMC102255.
replaced with
[269] Pandey, A.K.; Pain, J.; Singh, P.; Dancis, A.; Pain, D. Mitochondrial glutaredoxin Grx5 functions as a central hub for cellular iron-sulfur cluster assembly. J. Biol. Chem. 2025, 301, 108391. doi: 10.1016/j.jbc.2025.108391. PMID: 40074084; PMCID: PMC12004709.
[194] Mancuso, C.; Barone, E. The heme oxygenase/biliverdin reductase pathway in drug research and development. Curr. Drug Metab. 2009, 10, 579-594. doi: 10.2174/138920009789375405. PMID: 19702533.
replaced with
[270] Mancuso, C. Biliverdin as a disease-modifying agent: An integrated viewpoint. Free Radic. Biol. Med. 2023, 207, 133-143. doi: 10.1016/j.freeradbiomed.2023.07.015. PMID: 37459935.
[251] Buettner, G.R.; Jurkiewicz, B.A. Ascorbate radical: a valuable marker of oxidative stress. In: Analysis of Free Radicals in Biological Systems (Favier et al., eds.), 1985, Birkhäuser Verlag Basel/Switzerland.
replaced with
[327] Gęgotek, A.; Skrzydlewska, E. Ascorbic acid as antioxidant. Vitam. Horm. 2023, 121, 247-270. doi: 10.1016/bs.vh.2022.10.008. PMID: 36707136.
The DOI is correctly formatted but check if all digital object identifiers (DOIs) are included for other references where available.
All DOIs have been included whenever available.
The order of references should follow “1,2,3…..”
The reference format has been corrected.
Authors should be separated by “;” and font size should be standardized.
The requested changes have been made.
The use of journal name abbreviations should be consistent. If " J Gerontol A Biol Sci Med Sci" is abbreviated here, ensure all other journal names are treated similarly.
Consistency in the use of journal name abbreviations has been ensured.
Finally, the manuscript underwent a thorough linguistic review of English grammar, syntax, punctuation, hyphenation, and style.

Round 2
Reviewer 2 Report
Comments and Suggestions for Authors
no